# A Unified Framework for Diffusion Model Unlearning with f-Divergence

**Nicola Novello** [1] **Federico Fontana** [2] **Luigi Cinque** [2] **Deniz Gündüz** [3] **Andrea M. Tonello** [1]

## Abstract

Most existing methods for concept unlearning in text-to-image diffusion models minimize a mean squared error (MSE) loss between the denoiser outputs conditioned on a target and an anchor concept, which is implicitly the KL divergence between two Gaussians. We generalize this objective to any $f$-divergence, recovering MSE as the KL instance, and identify a family of $\alpha$-divergences whose Gaussian closed-form yields cheap, MSE-like training objectives. For the remaining $f$-divergences, we provide a min-max objective based on the variational formulation of the $f$-divergence. We theoretically analyze and numerically validate how different $f$-divergences impact the gradient magnitude and the convergence properties of the algorithm, affecting the quality of unlearning. For instance, we observe that the Hellinger closed-form instance consistently dominates MSE across multiple scenarios. More generally, the proposed unified framework offers a flexible paradigm for selecting the optimal divergence based on the application and user goal, allowing for finer control over the trade-off between unlearning efficacy and generative fidelity. Code available at https://github.com/tonellolab/f-DMU.
WARNING: This paper contains model outputs that may be offensive.

## 1. Introduction

Deep learning algorithms for text-to-image (T2I) generation based on diffusion models (DMs) have demonstrated remarkable success and widespread adoption in recent years. Some notable examples are Stable Diffusion (Rombach et al., 2022), DALL·E 2 (Ramesh et al., 2022), and Im-

agen (Saharia et al., 2022), which find a large variety of applications in real-world scenarios. However, a significant challenge associated with these algorithms is that they can generate Not-Safe-For-Work (NSFW) content, including explicit material (Schramowski et al., 2023), and copyrighted works, such as artistic styles and personal information (Jiang et al., 2023; Carlini et al., 2023), as they are trained on large scale datasets that are scraped from the Internet, like LAION-5B (Schuhmann et al., 2022). Since DMs are able to learn and memorize all these contents (Somepalli et al., 2023b), it is crucial to develop algorithms that erase specific concepts from the final trained models. To force the erasure of a concept from a model, the naive solution would be to re-train the model with a curated dataset where the specific concept has been removed. However, this is impractical, as it is resource-intensive and time-consuming, and in some cases the training dataset could have been deleted. Therefore, it is fundamental to develop techniques for targeted concept erasure in trained DMs.

Many DM unlearning algorithms share a common paradigm: aligning the model's generation corresponding to the concept to erase (the "target") with the output induced by a substitute concept (the "anchor"). The latter can be selected in many different ways, depending on the goal of the user: null concept, superclass, or semantically close to/distant from the concept to forget. Usually, the DM output alignment is achieved by minimizing a mean squared error (MSE)-based loss (Gandikota et al., 2023; Kumari et al., 2023; Huang et al., 2024), derived from formulating the problem as the minimization of the Kullback-Leibler (KL) divergence between two Gaussian distributions conditioned on the target and anchor concepts.

In this paper, we propose a unified framework where, instead of minimizing the standard KL divergence, we minimize the $f$-divergence between the same two distributions. For a specific subset of $f$-divergences, we derive the loss function for DM unlearning from the closed-form expression of the $f$-divergence between two Gaussian distributions. These loss functions, similarly to the standard MSE, benefit from a simple implementation and fast training, but exhibit distinct gradient scaling characteristics that determine different convergence properties. When the closed-form expression does not exist, we use the variational representation of the $f$-divergence to formulate the objective as a min-max opti-

[1]University of Klagenfurt, Austria [2]Sapienza University of Rome, Italy [3]Imperial College London, UK. Correspondence to: Nicola Novello <nicola.novello@aau.at>, Andrea M. Tonello <andrea.tonello@aau.at>.

*Proceedings of the 43rd International Conference on Machine Learning*, Seoul, South Korea. PMLR 306, 2026. Copyright 2026 by the author(s).

mization, which allows us to tackle the DM unlearning task with *any* $f$-divergence. We refer to the proposed framework as $f$-divergence-based DM Unlearning ($f$-DMU).

To the best of our knowledge, $f$-DMU is the first approach based on the use of $f$-divergences for the DM unlearning task. We theoretically study the gradients of the $f$-DMU losses derived from closed-form expressions of divergences and empirically verify the theoretical findings. Then, we theoretically prove the local convergence property of the proposed min-max formulation, relating the $f$-divergence choice with the algorithm convergence speed. Finally, we perform an extensive numerical analysis to compare results with different $f$-divergences and demonstrate the superiority of the proposed unified framework compared to the standard MSE approach. In particular, we empirically demonstrate that the deployment of $f$-divergences outperforms the MSE loss consistently across different unlearning scenarios, such as style ablation, object ablation, and nudity erasure.

## 2. Preliminaries

### 2.1. Related Work

In this section, we present a condensed version of the related work in which we focus on DM-based methods, while we defer to Appendix A for an extended overview.

**Post-processing techniques**   Post-processing techniques target the elimination of unsafe generated images through the usage of filtering or inference guiding. Stable Diffusion (SD) (Rombach et al., 2022) adopts a NSFW filter that removes all generated images with embeddings close to those of 17 pre-chosen nudity concepts (Rando et al., 2022). Safe Latent Diffusion (SLD) (Schramowski et al., 2023) is applied during inference and acts by repelling the generation from unsafe contents. Another inference guidance-based approach is SAFREE (Yoon et al., 2025), which can be used for both image and video generation. The main drawback of post-processing algorithms is that it is possible to remove them from the inference pipeline to allow the model to generate what should have been erased.

**Fine-tuning techniques**   Fine-tuning approaches modify the weights of a trained model. These methods often rely on aligning the DM output corresponding to target and anchor concepts by minimizing the KL divergence. Erased Stable Diffusion (ESD) (Gandikota et al., 2023) and its derivatives (Huang et al., 2024) fine-tune the model to align the output distributions conditioned on a target concept with those corresponding to a neutral concept. Concept Ablation (CAbl) (Kumari et al., 2023) minimizes the MSE between the outputs of the model conditioned on target and super-class anchor concepts. With the goal of finding the best anchor concept, Bui et al. (2024; 2025) propose adversarial

learning frameworks with a loss function comprising two MSE losses, and show that the best anchor concept should be semantically close to the target. Thakral et al. (2025) apply this principle to design an MSE loss that shifts the target concept to semantically close concepts. Domain Correction (DoCo) (Wu et al., 2025) extends CAbl by using a GAN-based (Goodfellow et al., 2014) loss function. We show in Appendix A that DoCo and CAbl's loss functions can be obtained as special cases of the proposed $f$-DMU framework. Another set of works proposes to update the model weights in closed-form, leading to faster computations (Gandikota et al., 2024; Gong et al., 2024; Lu et al., 2024). These methods also target the minimization of the MSE, but the loss is formulated directly on the cross-attention weights. However, these approaches can be applied only to specific architectures, while, in contrast, divergence-based methods can update any network parameter and work for different models, such as flow-based generative models (Zhang et al., 2025).

### 2.2. Diffusion Models

Diffusion models (Sohl-Dickstein et al., 2015; Ho et al., 2020) are state-of-the-art generative models consisting of two components that can be modeled by Markov chain processes. In the forward process, Gaussian noise (referred to as $\epsilon$) is gradually added to an input image $\mathbf{x}_0$ over multiple steps $t \in [0, \dots, T]$, to get $\mathbf{x}_T \sim \mathcal{N}(0, I)$. At each time step, the noisy image can be obtained as $\mathbf{x}_t = \sqrt{\alpha_t}\mathbf{x}_{t-1} + \sqrt{1 - \alpha_t}\epsilon$, where $\alpha_t$ specifies the variance schedule. In the reverse process, $\mathbf{x}_T$ is transformed, following a transformation that is the inverse of the forward process, to obtain a denoised image using a denoising network $\Phi(\mathbf{x}_t, \mathbf{c}, t)$, where the concept $\mathbf{c}$ is a text prompt for T2I models. The denoising process is characterized as $p_\Phi(\mathbf{x}_0, \dots, \mathbf{x}_T|\mathbf{c}) = p(\mathbf{x}_T)\prod_{t=1}^{T} p_\Phi(\mathbf{x}_{t-1}|\mathbf{x}_t, \mathbf{c})$, where $p_\Phi(\mathbf{x}_{t-1}|\mathbf{x}_t, \mathbf{c})$ describes the probability of $\mathbf{x}_{t-1}$ given the noisy image $\mathbf{x}_t$ and the concept $\mathbf{c}$.

### 2.3. $f$-Divergence

Let $p(\mathbf{x})$ and $q(\mathbf{x})$ be two probability density functions on domain $\mathcal{X}$. The $f$-divergence between $p(\mathbf{x})$ and $q(\mathbf{x})$ is defined as (Ali & Silvey, 1966; Csiszár, 1967)

$$D_f(p||q) = \int_{\mathcal{X}} q(\mathbf{x})f\left(\frac{p(\mathbf{x})}{q(\mathbf{x})}\right) d\mathbf{x}, \quad (1)$$

where $p \ll q$ (i.e., $p$ is absolutely continuous with respect to $q$) and the *generator function* $f : \mathbb{R}_+ \longrightarrow \mathbb{R}$ is a convex, lower-semicontinuous function such that $f(1) = 0$. The KL divergence is a special case of $f$-divergence, where $f(u) = u \log u$. The variational representation of $f$-divergence (Nguyen et al., 2010) reads as

$$D_f(p||q) = \sup_{T:\mathcal{X}\to\mathbb{R}} \{\mathbb{E}_p[T(\mathbf{x})] - \mathbb{E}_q[f^*(T(\mathbf{x}))]\}, \quad (2)$$

where $T$ is a parametric function (e.g., a neural network) and $f^*$ represents the *Fenchel conjugate* of $f$, defined as $f^*(t) = \sup_{u \in dom_f} \{ut - f(u)\}$, with $dom_f$ being the domain of $f$. The supremum in (2) is attained for $T^\diamond(\mathbf{x}) = f'(p(\mathbf{x})/q(\mathbf{x}))$, where $f'$ is the first derivative of $f$.

# 3. Concept Erasing with $f$-Divergence

In this paper, we present an $f$-divergence-based framework to erase a target concept $\mathbf{c}^*$ from a DM. The concept $\mathbf{c}^*$ is erased by shifting the model generation corresponding to $\mathbf{c}^*$ to the output conditioned on an anchor concept $\mathbf{c}$. Kumari et al. (2023) achieved this by minimizing the KL divergence between the reverse processes of the original $\Phi$ and unlearned $\hat{\Phi}$ DMs, i.e., $D_{KL}(p_\Phi(\mathbf{x}_{(0...T)}|\mathbf{c})||p_{\hat{\Phi}}(\mathbf{x}_{(0...T)}|\mathbf{c}^*))$. Leveraging the Markov property of diffusion processes (see Appendix B.1 for the derivation details), Kumari et al. showed that such an objective function can be rewritten as the KL divergence between the DM outputs at specific time instants. We generalize this idea using the $f$-divergence, formulating the unlearning problem as

$$\min_{\hat{\Phi}} \mathbb{E}_{\mathbf{x}, \mathbf{c}^*, \mathbf{c}, t} \Big[ D_f \big( p_\Phi(\mathbf{x}_{t-1}|\mathbf{x}_t, \mathbf{c}) || p_{\hat{\Phi}}(\mathbf{x}_{t-1}|\mathbf{x}_t, \mathbf{c}^*) \big) \Big]. \tag{3}$$

The generalized objective function in (3) provides an unlearning framework that extends beyond the KL-based losses used in prior work. Specifically, while the global optimum of (3) remains invariant to the choice of $f$, the optimization landscape is not. Different $f$-divergences exhibit different gradient dynamics that impact the convergence properties of the unlearning process (see Sec. 4). Furthermore, because the formulation in (3) is purely based on the divergence between model output distributions, it is model-agnostic, thus working also with flow-based generative models (Lipman et al., 2023), contrary to architecture-specific approaches (Zhang et al., 2025). Moreover, the divergence-based formulation of (3) allows any choice of the anchor concept $\mathbf{c}$. For instance, $\mathbf{c}$ can be chosen as a neutral concept (Gandikota et al., 2023), superclass concept (Kumari et al., 2023), a concept with high semantic similarity to $\mathbf{c}^*$ (Bui et al., 2025), or a concept with low semantic similarity to $\mathbf{c}^*$ (George et al., 2025). In the literature, there is not an agreement on the best anchor concept to use. While the superclass or semantically close concepts are usually considered the best options, since they are similar to the target prompt, they appear to be more sensitive to subsequent fine-tuning of the model (George et al., 2025), leading to possible remembering of erased concepts. On the contrary, semantically distant concepts increase robustness to future fine-tunings, but lead to generations unrelated to the given prompt. As a final note, instead of solving (3), it is possible to minimize the divergence between the probability distributions over a subset of the original generation trajectory

(from $\mathbf{x}_t$ to $\mathbf{x}_0$, with $t < T$) (Lu et al., 2024), and the $f$-divergence generalization still holds. This reasoning can also be extended to the extreme case of training the model with only samples $\mathbf{x}_0$, as proposed by Zhang et al. (2025).

Previous work solved (3) when the $f$-divergence is the KL by using the fact that the KL divergence between Gaussian distributions leads to the MSE between their means. In this paper, we extend this solution to any $f$-divergence. It is possible to express $D_f(P||Q)$, and thus (3), in closed-form when $P$ and $Q$ are Gaussian, for a subset of divergences. In particular, three examples of these divergences are: Jeffreys divergence, squared Hellinger distance, and Pearson $\chi^2$ divergence. More generally, we broaden this approach to a sub-family of $f$-divergences, referred to as $\alpha$-divergences (Amari, 1985; Sourla et al., 2024) in Appendix B.1.1. We defer to Appendix B.1.1 for the calculations.

**Jeffreys divergence**: The Jeffreys divergence between two probability distributions $P$ and $Q$ is defined as $D_J(P||Q) = D_{KL}(P||Q) + D_{KL}(Q||P)$. The objective function is reported in Appendix B.1.1.

**Squared Hellinger distance**: The squared Hellinger distance (denoted as $H^2$ in short) between two probability distributions $P$ and $Q$ can be expressed in terms of the Bhattacharyya coefficient ($BC(P, Q)$) as $H^2(P, Q) = 1 - BC(P, Q)$. The objective function based on squared Hellinger distance (referred to as $\mathcal{J}_H(\hat{\Phi})$) reads as

$$\mathbb{E}_{\mathbf{x}, \mathbf{c}^*, t} \Big[ -\omega_t \exp \big\{ -||\Phi(\mathbf{x}_t, \mathbf{c}, t) - \hat{\Phi}(\mathbf{x}_t, \mathbf{c}^*, t)||_2^2 \big\} \Big], \tag{4}$$

where $\Phi(\cdot)$ and $\hat{\Phi}(\cdot)$ are the outputs of the original and unlearned DM, respectively, and $\omega_t$ is a time-dependent scalar weight.

$\chi^2$ **divergence**: The closed-form expression for $\chi^2$ divergence between two Gaussian random variables exists under a mild assumption on the variance, which always holds in our case (see Sec. B.1.1). The objective function based on the Pearson $\chi^2$ divergence (referred to as $\mathcal{J}_{\chi^2}(\hat{\Phi})$) becomes

$$\mathbb{E}_{\mathbf{x}, \mathbf{c}^*, \mathbf{c}, t} \Big[ \omega_t \exp \big\{ ||\Phi(\mathbf{x}_t, \mathbf{c}, t) - \hat{\Phi}(\mathbf{x}_t, \mathbf{c}^*, t)||_2^2 \big\} \Big]. \tag{5}$$

In this work, we follow the standard practice of fixing the variance of DMs to a constant value. Thus, our loss functions do not include a variance term, even though closed-form expressions for $f$-divergences between Gaussians with different variances are available (Appendix B.1.1).

Although any $f$-divergence between two multivariate normal distributions with the same covariance matrix $\Sigma$ is an increasing function of their Mahalanobis distance (Ali & Silvey, 1966; Nielsen & Okamura, 2024) $\Delta_\Sigma(\mu_P, \mu_Q) = \sqrt{(\mu_Q - \mu_P)^T \Sigma^{-1}(\mu_Q - \mu_P)}$, not all $f$-divergences can

be expressed in closed-form. One example is the total variation distance (Nielsen & Okamura, 2024). Driven by the intent of providing a unified $f$-divergence-based framework, we present a comprehensive formulation to solve the problem in (3) by using *any* $f$-divergence by adopting its variational representation. We formulate the general loss as (more details in Appendix B.1.3)

$$\min_{\hat{\Phi}} \max_{T} \mathbb{E}_{\mathbf{x}, \mathbf{c}^*, \mathbf{c}, t} \Big[ \mathbb{E}_{p_\Phi(\mathbf{x}_{t-1}|\mathbf{x}_t, \mathbf{c})} \left[ T(\Phi) \right]$$
$$- \mathbb{E}_{p_{\hat{\Phi}}(\mathbf{x}_{t-1}|\mathbf{x}_t, \mathbf{c}^*)} \left[ f^*(T(\hat{\Phi})) \right] \Big], \quad (6)$$

where $T(\cdot)$ can be parametrized as a neural network fed with the output samples of the original $\Phi$ and the unlearned $\hat{\Phi}$. This optimization problem can be treated as a min-max game between generator $\hat{\Phi}$ and discriminator $T$, where $T$ estimates the divergence between $p_\Phi$ and $p_{\hat{\Phi}}$, while $\hat{\Phi}$ aims to minimize it.

The loss functions derived from closed-form expressions of $f$-divergences share the computational efficiency of the standard MSE, requiring only to solve a minimization problem. Beyond the standard MSE, these loss functions introduce different gradient scaling factors (see Section 4.1) that lead to improved convergence properties. In contrast, the loss functions expressed using the variational formulation require solving a min-max optimization problem. This approach, however, provides a more general framework applicable to any $f$-divergence, leading to a broader class of loss functions. We study the local convergence properties of the variational framework in Section 4.2.

## 4. Theoretical Analysis

### 4.1. Gradient Analysis

An analysis of the gradients of the loss functions obtained from the closed-form expression of $f$-divergences (referred to as $\mathcal{J}_f$) reveals significant differences in gradient magnitude compared to the standard MSE loss derived from the KL divergence. For simplicity in the notation, let $\text{MSE}(\Phi, \hat{\Phi})$ refer to $\text{MSE}(\Phi(\mathbf{x}_i, \mathbf{c}, i), \hat{\Phi}(\mathbf{x}_i, \mathbf{c}^*, i))$, and let $\phi$ be the vector of parameters of $\hat{\Phi}$. Then, the gradients of the loss functions for an $i$-th sample can be written as

$$\frac{\partial \mathcal{J}_f(\phi)}{\partial \phi} = \begin{cases} \nabla_\phi \text{MSE}(\Phi, \hat{\Phi}) & \text{for KL} \\ e^{-\text{MSE}(\Phi, \hat{\Phi})} \nabla_\phi \text{MSE}(\Phi, \hat{\Phi}) & \text{for H}^2 \\ e^{\text{MSE}(\Phi, \hat{\Phi})} \nabla_\phi \text{MSE}(\Phi, \hat{\Phi}) & \text{for } \chi^2 \end{cases} \quad (7)$$

Both the gradients of H$^2$ and $\chi^2$ are proportional to the gradient of KL. However, they are characterized by two opposite behaviors. When MSE $\to \infty$, the gradients of H$^2$ and $\chi^2$ tend to 0 and $\infty$, respectively. When MSE $\to 0$, the gradients of H$^2$ and $\chi^2$ both tend to the gradient of the MSE.

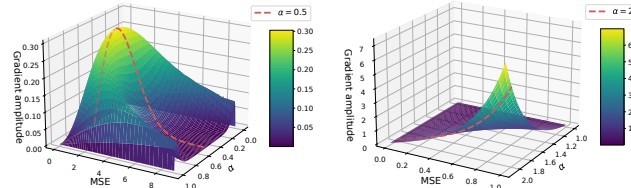

*Figure 1.* Gradient amplitude for losses derived from $\alpha$-divergences, varying $\alpha$ and MSE. *Left*: gradient amplitude for $0 < \alpha < 1$, highlighting the H$^2$ divergence ($\alpha = 0.5$). *Right*: gradient amplitude for $\alpha > 1$, including the $\chi^2$ divergence ($\alpha = 2$).

In general,

$$\left| \frac{\partial \mathcal{J}_{\text{H}^2}(\phi)}{\partial \phi} \right| \leq \left| \frac{\partial \mathcal{J}_{\text{KL}}(\phi)}{\partial \phi} \right| \leq \left| \frac{\partial \mathcal{J}_{\chi^2}(\phi)}{\partial \phi} \right|, \quad (8)$$

where the component-wise inequalities become equalities when the MSE is zero.

**Remark**. The per-sample weighting caused by different $f$-divergences cannot be obtained with standard learning rate schedulers and gradient scaling techniques (e.g., step decay, cosine annealing, gradient clipping, ReduceLROnPlateau, Hypergradient Descent (Baydin et al., 2018)), which instead apply a scalar to the entire batch, typically based on the training or gradients history. The per-sample scaling in (7) provides specific characteristics to each $f$-divergence-based loss. For instance, H-DMU mitigates the effect of outliers. However, it requires no manual tuning and introduces no additional computational overhead, as it is an inherent property of $f$-DMU rather than a hand-crafted method.

We generalize the analysis above for the losses obtained from the class of $\alpha$-divergences mentioned in Sec. 3. We show the gradient amplitude of such a generalized case in Fig. 1, varying $\alpha$. Our specific choice of focusing on the H$^2$ and $\chi^2$ divergences is justified by their peculiar behaviors shown in Fig. 1. More details can be found in Appendix B.1.2, where we report the computation of the $\alpha$-divergence gradients and provide a more detailed analysis (Fig. 10).

### 4.2. Local Convergence Analysis

In this section, we address the local convergence of the variational form of $f$-DMU to show that, under mild assumptions, the proposed algorithm is locally exponentially stable around equilibrium points. Similarly to how previous work studied the convergence of GANs (Nagarajan & Kolter, 2017; Mescheder et al., 2018) and energy-based models (Yu et al., 2020a), we use nonlinear dynamical systems theory (Hassan K, 1996). We consider the optimization of the model parameters of the min-max problem in (6) as a dynamical system. The system can be linearized (around the optimal convergence point) to study its local convergence properties. By evaluating the Jacobian matrix, we can conclude that if the Jacobian at an equilibrium point is

a Hurwitz matrix, the system converges to the equilibrium (i.e., the equilibrium is locally exponentially stable). Let $\hat{\Phi}$ and $T$ be parametrized by the vectors of parameters $\phi$ and $\omega$, respectively. We rewrite the objective function in (6) as $\min_\phi \max_\omega \mathcal{J}_f^v(\phi, \omega)$. Following Nagarajan et al. (2017), we focus on the analysis of continuous time ordinary differential equations, which implies similar results for discrete time updates when the learning rate is sufficiently small. Following a gradient update rule, the dynamical system describing the models' update is given by

$$\begin{aligned} \dot{\phi} &= -\nabla_\phi \mathcal{J}_f^v(\phi, \omega) \\ \dot{\omega} &= \nabla_\omega \mathcal{J}_f^v(\phi, \omega) \end{aligned}. \quad (9)$$

### 4.2.1. MAIN CONVERGENCE RESULTS

Theorem 4.1 provides the Jacobian of the dynamical system describing the training of $\hat{\Phi}$ and $T$ at an equilibrium point. Theorem 4.2 provides the main theoretical result of this paper, stating the stability of the dynamical system in (9).

**Theorem 4.1.** *The Jacobian for the dynamical system defined in* (9)*, at an equilibrium point* $(\phi^*, \omega^*)$ *is*

$$\mathbf{J} = \begin{pmatrix} \mathbf{0} & -\boldsymbol{K}_{TP} \\ \boldsymbol{K}_{TP}^T & \boldsymbol{K}_{TT} \end{pmatrix}, \quad (10)$$

*where*

$$\boldsymbol{K}_{TP} \triangleq \mathbb{E}_{\mathbf{x}, \mathbf{c}^*, \mathbf{c}, t}\Big[\mathbb{E}_{p_\Phi(\mathbf{x}_{t-1}|\mathbf{x}_t, \mathbf{c})}\big[-\nabla_\phi \log(p_\phi(\mathbf{x}_{t-1}|\mathbf{x}_t, \mathbf{c}^*))$$
$$\cdot \big(\nabla_\omega^T T_\omega\big)\big]\big]\Big|_{(\phi^*, \omega^*)}, \quad (11)$$

$$\boldsymbol{K}_{TT} \triangleq \mathbb{E}_{\mathbf{x}, \mathbf{c}^*, \mathbf{c}, t}\Big[\mathbb{E}_{p_\Phi(\mathbf{x}_{t-1}|\mathbf{x}_t, \mathbf{c})}\big[-(f^*)''(T_\omega)$$
$$\cdot \nabla_\omega T_\omega \nabla_\omega^T T_\omega\big]\big]\Big|_{\omega^*}. \quad (12)$$

**Theorem 4.2.** *The dynamical system defined in* (9) *is locally exponentially stable with respect to an equilibrium point* $(\phi^*, \omega^*)$ *under Assumptions* B.6, B.7, B.8. *Let* $\lambda_m(\cdot)$ *and* $\lambda_M(\cdot)$ *be the smallest and largest eigenvalues of a given matrix, respectively. The rate of convergence of the system is governed by the eigenvalues of the Jacobian* $\mathbf{J}$ *which have a negative real part upper bounded as*

• *When* $Im(\lambda) = 0$,

$$Re(\lambda) \le -\frac{\lambda_m(-\boldsymbol{K}_{TT})\lambda_m(\boldsymbol{K}_{TP}\boldsymbol{K}_{TP}^T)}{\lambda_m(-\boldsymbol{K}_{TT})\lambda_M(-\boldsymbol{K}_{TT}) + \lambda_m(\boldsymbol{K}_{TP}\boldsymbol{K}_{TP}^T)}. \quad (13)$$

• *When* $Im(\lambda) \ne 0$,

$$Re(\lambda) \le -\frac{\lambda_m(-\boldsymbol{K}_{TT})}{2}. \quad (14)$$

Besides the result on the local convergence of the dynamical system in (9), we can infer properties on the convergence speed of different $f$-divergences from Theorem 4.2. Specifically, we leverage the upper bounds on the real parts of the eigenvalues and the fact that the bottleneck of the convergence speed is the largest eigenvalue. In Appendix B.2.4, we show that the $f$-divergences characterized by a larger $(f^*)''(T_\omega)|_{(\phi^*, \omega^*)}$ (i.e., a smaller $f''(1)$) are favored by a faster convergence. This theoretical result is fundamental, as it provides guidelines on the choice of $f$-divergence for our variational framework. In particular, we show that $f''_{H^2}(1) < f''_{KL}(1) < f''_{\chi^2}(1)$, implying that $H^2$ is characterized by a faster convergence near the equilibrium point.

Finally, it is possible to compare different $f$-divergences based on their mode-seeking (Li & Farnia, 2023) and saturation (Goodfellow et al., 2014; Nowozin et al., 2016a) properties. In the unlearning context, mode-seeking divergences tend to overfit by converging to a subset of modes of the distribution, thus reducing the diversity of generated images corresponding to the unlearned concepts, while mode-covering divergences yield models that may assign higher probability to the areas between the modes. Both JS and $H^2$ are characterized by medium mode-seeking properties, while KL and $\chi^2$ divergences are mode-covering ($\chi^2$ is more mode-covering than KL). The saturation problem can appear on $f$-divergence-based generative models relying on the variational formulation, and refers to the presence of weak gradients in the initial stages of training, when the density ratio is either very large or very small, presenting optimization difficulties. JS and $H^2$ divergences are subject to strong saturation. The saturation problem is alleviated in the considered unlearning scenario as we fine-tune a generative model starting from an already trained DM (as in Xu et al. (2025)). A summary of the differences between $f$-DMU losses can be found in Appendix B.3 (Tab. 6).

## 5. Results

We present a comprehensive empirical evaluation of $f$-DMU. Our experiments are designed to: 1) validate our theoretical analysis of gradient magnitude, 2) demonstrate the effectiveness of using alternative $f$-divergences compared to the standard MSE-based approach. Specific instances of the $f$-DMU framework are denoted by replacing $f$ with the identifier of the corresponding $f$-divergence (to simplify the notation, H-DMU denotes the usage of $H^2$, and we use P-DMU and $\chi^2$-DMU interchangeably). We use $f^v$-DMU to refer to the variational framework.

### 5.1. Implementation Details

The experiments presented in the main part of this paper are based on the publicly available Stable Diffusion v1.4, v1.5, v2.1, and XL models. Ad-

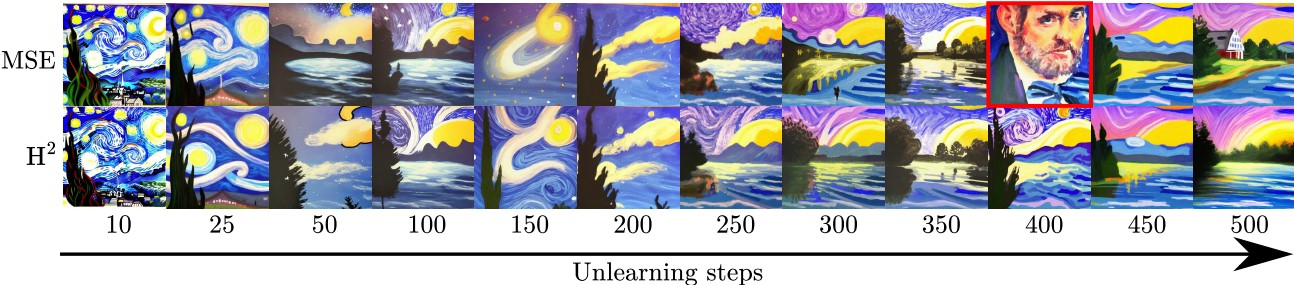

MSE

H²

10    25    50    100    150    200    250    300    350    400    450    500

Unlearning steps

*Figure 2.* Instability of image generation during fine-tuning caused by large gradient magnitudes. Top: MSE, bottom: H².

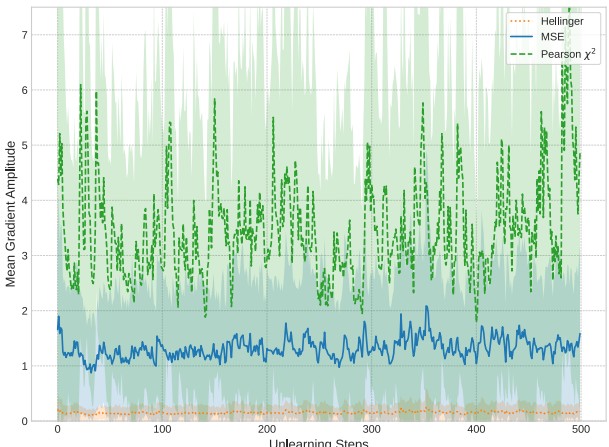

*Figure 3.* Average gradient amplitude of closed-form-based losses.

ditionally, Appendix C includes experiments with `Stable Diffusion v3` and `FLUX`. We measure how successfully the concept has been unlearned using CLIP Score (Hessel et al., 2021) (CS) and CLIP Accuracy (Radford et al., 2021) (CA). The lower CS and CA are for the target concept, the better the erasure has been. For non-target concepts, the higher CS and CA are, the better their preservation has been. Kernel Inception Distance (Bińkowski et al., 2018) (KID) is used to address the generative distribution change of the models. For non-target concepts, a smaller KID implies good maintenance of the overall quality and coherence of the model. For target concepts, a smaller KID refers to a clean and realistic replacement, while a bigger KID might refer to a more destructive erasure, producing largely incoherent outputs. Additional implementation details are provided in Appendix C.1.

### 5.2. Gradient Analysis: Empirical Validation of Theory

In Sec. 4.1, we theoretically compare the gradient magnitude between different $f$-DMU loss functions derived from the closed-form expressions of divergences. We empirically validate these findings in Fig. 3, which visualizes the (moving average of the) average gradient amplitude

for MSE (blue), H-DMU (orange), and $\chi^2$-DMU (green) throughout the unlearning process. While H-DMU yields bounded gradients significantly smaller than the other two losses, $\chi^2$-DMU is characterized by meaningfully larger gradient magnitudes. By performing more conservative weights updates than MSE, H-DMU avoids abrupt changes of the generated images along the fine-tuning procedure. One example of the resulting stability is reported in Fig. 2. Although H-DMU and MSE return comparable images at the end of 500 unlearning steps, MSE induces a sudden degenerate image generation during the intermediate stages. In contrast, H-DMU maintains generation stability throughout the entire fine-tuning process. Another consequence of H-DMU bounded gradients is the better prior preservation with respect to MSE and $\chi^2$-DMU (see Sec. 5.3).

### 5.3. Unlearning Efficacy of Different $f$-Divergences

**Closed-form vs variational losses** Regarding closed-form losses, for object and artistic style removal, H-DMU is characterized by a better prior preservation compared to MSE and $\chi^2$-DMU, as visible in Fig. 4, Tabs. 1, 2, 3, and Tabs. 11, 12 in Appendix C. This outcome is motivated by the conservative weights updates caused by the bounded gradients of H-DMU. Furthermore, MSE generally leads to worse KID scores compared to H-DMU, indicating a greater influence on generative quality. From the same tables we observe that, contrary to H-DMU, $\chi^2$-DMU generally results in a stronger unlearning and worse preservation of non-target concepts (e.g., fully removing any R2D2-related feature in Fig. 4).

Beyond the tests on SD 1.4 and SD 2.1, we test different $f$-DMU losses on SDXL (Podell et al., 2024) with two goals: 1) demonstrate the applicability of our method to the SDXL architecture; 2) observe the effect of different gradient magnitudes on a significantly larger model. The results are reported in Fig. 5, where the impact of a higher gradient magnitude (increasing from H² to MSE and from MSE to $\chi^2$) visibly affects the generated images.

For variational-based loss functions, a quantitative comparison between different $f^v$-DMU losses is reported in Tab. 2,

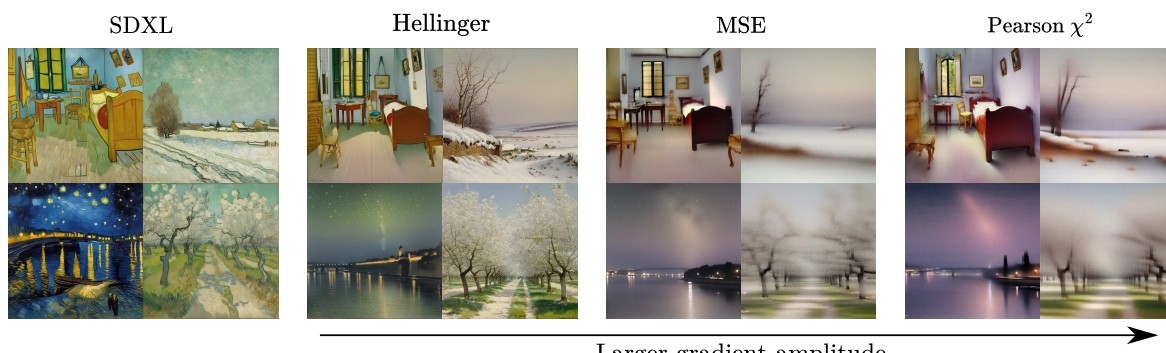

**Figure 4.** Comparison of different $f$-divergences with closed-form and variational losses on SD 1.4 to unlearn "R2D2". Knowledge preservation: the inset in the bottom right of each image shows an image of "C-3PO".

**Table 1.** Prior preservation capability of the $H^2$ method using a superclass anchor. SD 1.4 is fine-tuned for 2000 steps.

| | Grumpy Cat | | | Snoopy | | | Wall-E | | | R2D2 | | | Van Gogh | | | Salvador Dali | | |
|---|---|---|---|---|---|---|---|---|---|---|---|---|---|---|---|---|---|---|
| *Erasing Grumpy Cat* | CS↓ | CA↓ | KID | CS↑ | CA↑ | KID↓ | CS↑ | CA↑ | KID↓ | CS↑ | CA↑ | KID↓ | CS↑ | CA↑ | KID↓ | CS↑ | CA↑ | KID↓ |
| Original Model | 0.74 | 1.00 | - | 0.73 | 1.00 | - | 0.73 | 1.00 | - | 0.75 | 1.00 | - | 0.80 | 1.00 | - | 0.68 | 0.80 | - |
| MSE | **0.56** | **0.53** | 0.194 | 0.62 | 0.70 | 0.070 | 0.65 | 0.73 | 0.082 | 0.70 | 0.90 | 0.122 | 0.72 | 0.97 | 0.049 | 0.67 | 0.77 | 0.031 |
| $H^2$ | **0.56** | 0.70 | 0.147 | **0.68** | **0.97** | **0.014** | **0.75** | **0.93** | **0.015** | **0.77** | **1.00** | **0.061** | **0.77** | **1.00** | **-0.009** | **0.69** | **0.87** | **-0.022** |
| *Erasing Salvador Dali* | CS↑ | CA↑ | KID↓ | CS↑ | CA↑ | KID↓ | CS↑ | CA↑ | KID↓ | CS↑ | CA↑ | KID↓ | CS↑ | CA↑ | KID↓ | CS↓ | CA↓ | KID |
| Original Model | 0.74 | 1.00 | - | 0.73 | 1.00 | - | 0.73 | 1.00 | - | 0.75 | 1.00 | - | 0.80 | 1.00 | - | 0.68 | 0.80 | - |
| MSE | 0.62 | 0.73 | 0.250 | 0.61 | 0.67 | 0.151 | 0.65 | 0.93 | 0.135 | 0.73 | **1.00** | 0.298 | 0.61 | 0.50 | 0.459 | **0.57** | **0.07** | 0.167 |
| $H^2$ | **0.73** | **1.00** | **0.005** | **0.73** | **1.00** | **0.004** | **0.78** | **1.00** | **0.048** | **0.76** | **1.00** | **0.040** | **0.62** | **0.53** | **0.361** | 0.58 | 0.10 | 0.102 |

SDXL      Hellinger      MSE      Pearson $\chi^2$

Larger gradient amplitude →

**Figure 5.** Erasing Van Gogh on SDXL. Comparison between three closed-form $f$-DMU losses.

**Table 2.** Unlearning *Van Gogh* on SD 2.1. Comparison of $f$-divergences with state-of-the-art methods.

| | Erased (Van Gogh) | | Preserved Concepts | | |
|---|---|---|---|---|---|
| **Method** | CS (↓) | CA (↓) | CS (↑) | CA (↑) | KID (↓) |
| ESD (Gandikota et al., 2023) | 0.657 | 0.6 | 0.668 | 0.74 | 0.027 |
| CAbl (Kumari et al., 2023) | 0.635 | 0.2 | 0.668 | 0.78 | 0.028 |
| SPM (Lyu et al., 2024) | 0.714 | 1.0 | **0.733** | **0.92** | **0.021** |
| UCE (Gandikota et al., 2024) | 0.718 | 0.9 | 0.674 | 0.84 | 0.042 |
| MACE (Lu et al., 2024) | 0.690 | 0.8 | 0.679 | 0.88 | **0.021** |
| RECE (Gong et al., 2024) | 0.631 | 0.3 | 0.656 | 0.75 | 0.029 |
| SAFREE (Yoon et al., 2025) | 0.661 | 0.7 | 0.677 | 0.78 | 0.025 |
| DoCo (Wu et al., 2025) | 0.737 | 0.9 | 0.691 | 0.86 | 0.033 |
| SPEED (Li et al., 2026) | 0.642 | 0.4 | 0.684 | 0.80 | 0.029 |
| Hellinger (Closed-Form) | **0.624** | 0.2 | 0.672 | 0.78 | 0.027 |
| $\chi^2$ (Closed-Form) | 0.628 | **0.1** | 0.672 | 0.76 | 0.028 |
| KL (Variational) | 0.755 | 1.0 | 0.673 | 0.82 | 0.045 |
| Hellinger (Variational) | 0.645 | 0.5 | 0.702 | 0.88 | 0.051 |
| Jensen-Shannon (Variational) | 0.738 | 0.8 | 0.674 | 0.80 | 0.038 |

including also results for state-of-the-art approaches, and in Tabs. 11, 12, 16 in Appendix C. Table 2, where we use the same evaluation prompts for each method, highlights the good convergence properties of $H^v$-DMU, converging to a point of good erasure and best preservation. In particular, $H^v$-DMU outperforms DoCo, which is the state-of-the-art variational method. In general, the variational framework is more "aggressive" than the closed-form approach, achieving low CS and CA in fewer iterations (see Tab. 16). However, this rapid semantic disruption often compromises generative quality, resulting in higher KID values. Qualitative results of the variational methods for object and artistic style removal are displayed in Fig. 4 and in Fig. 16 in Appendix C. The motivation for the behavior of $f^v$-DMU is that the divergence estimate is coarse when computed over the small batch sizes typical of DM fine-tuning, leading to an imprecise minimization target. This results in a large noisy distribution shift that rapidly erodes the target concept with disruptive changes in the surrounding distribution. In contrast, the closed-form losses guarantee that we are minimizing a specific divergence for any batch size. Overall, the traits of the different methods highlight a trade-off: closed-form losses are best for realistic concept replacement, while

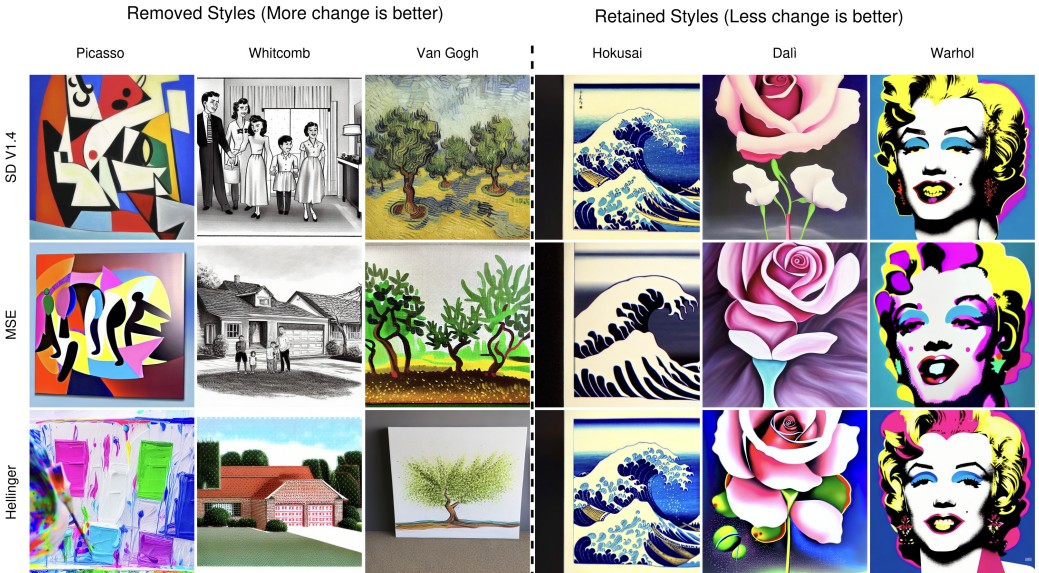

*Figure 6.* Erasure of 10 artistic styles. First row: SD 1.4. Second row: Unlearning with MSE. Third row: Unlearning with our H-DMU.

*Table 3.* Evaluation of sequential unlearning of 10 artistic styles on SD 1.4.

| Metric | Evaluation on Erased Artists | | | | | | | | | Evaluation on Retained Artists | | | | | | | | |
|---|---|---|---|---|---|---|---|---|---|---|---|---|---|---|---|---|---|---|
| | Monet | | | Picasso | | | Van Gogh | | | Dalí | | | Hokusai | | | Warhol | | |
| | Base | MSE | H$^2$ | Base | MSE | H$^2$ | Base | MSE | H$^2$ | Base | MSE | H$^2$ | Base | MSE | H$^2$ | Base | MSE | H$^2$ |
| **KID** *[Goal: ≈ 0 for Retained]* | — | 0.12 | 0.05 | — | 0.10 | 0.06 | — | 0.05 | 0.06 | — | 0.07 | **0.02** | — | 0.03 | **0.01** | — | **-0.01** | -0.02 |
| **CS** *[Goal: ↓ for Erased, ↑ for Retained]* | 0.74 | 0.60 | **0.59** | 0.72 | 0.63 | **0.60** | 0.80 | 0.61 | **0.61** | 0.72 | 0.62 | **0.62** | 0.77 | 0.72 | **0.74** | 0.72 | 0.67 | **0.70** |
| **CA** *[Goal: ↓ for Erased, ↑ for Retained]* | 1.00 | **0.45** | 0.54 | 0.90 | 0.40 | **0.35** | 1.00 | 0.55 | **0.40** | 0.80 | 0.50 | **0.60** | 1.00 | 0.80 | **0.95** | 0.90 | 0.85 | **1.00** |

variational losses are better for scenarios where the goal is aggressive concept removal and the realism of the output is a lower priority. Based on these considerations, we restrict the scope of our main analysis on $f$-DMU closed-form losses. However, a comprehensive evaluation including variational losses is provided in Appendix C. Finally, each table in Appendix C highlights (with green and red colors) that H-DMU performs better than MSE for all scenarios.

**Sequential Multi Concept Erasure** The observations previously reported for $f$-DMU still hold when erasing multiple concepts, as demonstrated in Fig. 6 and Tab. 3. We sequentially erase 10 artistic styles, comparing H-DMU with the standard MSE loss (used in CAbl). H-DMU shows superior prior preservation and better erasure performance. We report further information and results in Appendix C.3.

**Nudity erasure and robustness to adversarial prompts** We test $f$-DMU to erase the concept of nudity on SD 1.5, and we evaluate its robustness against different adversarial frameworks (Ring-A-Bell (RAB) (Tsai et al., 2024), MMA-Diffusion (Yang et al., 2024)), which generate ad-hoc prompts with the goal of re-evoking erased concepts. Regarding the erasure of nudity, we compare the erasure

performance of $f$-DMU with several state-of-the-art methods[1] in Tab. 4 and Tabs. 9, 10 in Appendix C.4. H-DMU leads to the least number of nudity generations on the I2P benchmark (Schramowski et al., 2023) and on the target (non-adversarial) prompts of MMA-Diffusion. Regarding the robustness to adversarial prompts (i.e., "RAB", "MMA Adv.", and "MMA S.Adv." in Tab. 4), H-DMU performs better than the compared state-of-the-art approaches. We report qualitative examples in Fig. 7 and in Fig. 13 in Appendix C.4. We test the robustness of $f$-DMU to adversarial prompts for artistic style removal (Fig. 14) in Appendix C.5.

## 6. Conclusions

In this paper, we propose a unified $f$-divergence-based framework for DM unlearning, which comprises two groups of loss functions: closed-form-based and variational-based losses. We theoretically analyze the proposed loss functions and numerically evaluate them in different scenarios, demonstrating their relevance for strong concept preservation or aggressive unlearning, depending on the $f$-divergence and on the closed-form or variational-based derivation. Our frame-

---

[1]The hyperparameters of CAbl have been optimized and correspond to the hyperparameters used for H-DMU and P-DMU.

*Table 4.* Erasing nudity and robustness to adversarial attacks. W.M.: modified model weights. A.I.: architecture independent.

| Method | W.M. | A.I. | Nudity generation rate ($\downarrow$) | | | | |
| --- | --- | --- | --- | --- | --- | --- | --- |
| | | | I2P | RAB | MMA Tar. | MMA Adv. | MMA S.Adv. |
| SD 1.5 | - | - | 0.669 | 0.982 | 0.553 | 0.716 | 0.601 |
| SLD | ✗ | ✓ | 0.444 | 0.926 | 0.407 | 0.540 | 0.447 |
| SAFREE | ✗ | ✓ | 0.176 | 0.561 | 0.243 | 0.378 | 0.302 |
| UCE | ✓ | ✗ | 0.394 | 0.653 | 0.460 | 0.619 | 0.503 |
| RECE | ✓ | ✗ | 0.176 | 0.179 | 0.158 | 0.209 | 0.137 |
| SPEED | ✓ | ✗ | 0.479 | 0.547 | 0.368 | 0.570 | 0.464 |
| DoCo | ✓ | ✓ | 0.451 | 0.656 | 0.221 | 0.287 | 0.239 |
| CAbl | ✓ | ✓ | 0.120 | **0.063** | 0.066 | 0.118 | 0.141 |
| $H^\nu$-DMU | ✓ | ✓ | 0.430 | 0.765 | 0.189 | 0.353 | 0.320 |
| $P^\nu$-DMU | ✓ | ✓ | 0.599 | 0.796 | 0.504 | 0.641 | 0.539 |
| H-DMU | ✓ | ✓ | **0.063** | 0.157 | **0.035** | **0.049** | **0.042** |
| P-DMU | ✓ | ✓ | 0.204 | 0.253 | 0.173 | 0.276 | 0.268 |

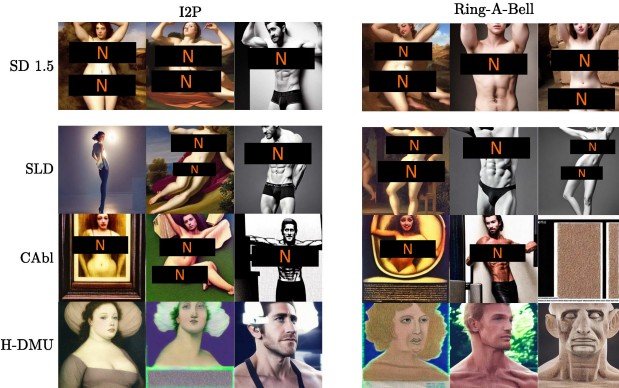

*Figure 7.* Erasing nudity and robustness to adversarial prompts.

work provides practitioners a clear recipe on how to select the loss function based on their specific unlearning requirements: i) H-DMU (closed-form) is recommended as the default for most tasks, where preserving non-target concepts and generative quality is crucial. ii) $\chi^2$-DMU (closed-form) is recommended when the user targets stronger erasure than MSE and some slight fidelity loss is acceptable. iii) Our variational framework, and in particular its Hellinger-based loss, is recommended when the goal is aggressive concept removal. We emphasize that the proposed $f$-divergence-based losses provide a new library of theoretically-grounded objectives that can serve as the foundation for future DM unlearning algorithms.

## Impact Statement

This paper presents work whose goal is to advance the field of Machine Learning. There are many potential societal consequences of our work, none which we feel must be specifically highlighted here.

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

# A. Related Work

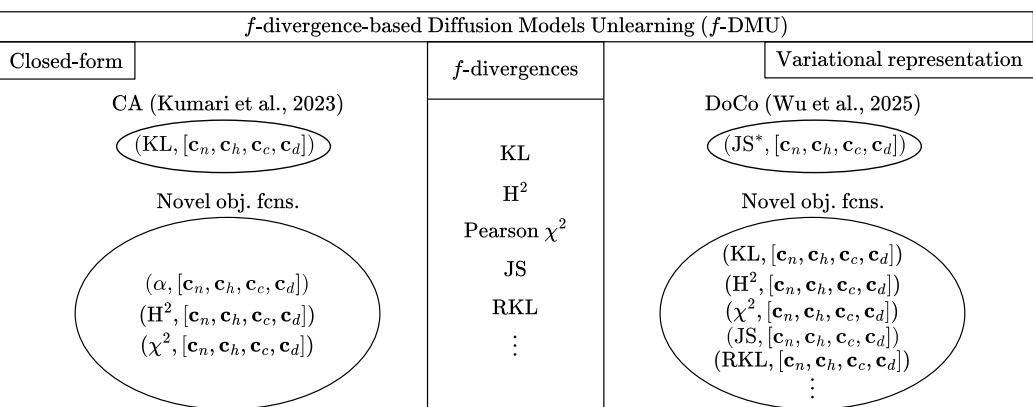

*Figure 8.* $f$-divergence-based Diffusion Model Unlearning framework. Every pair $(f, \mathbf{c})$ indicates a different objective function based on a specific $f$-divergence and using a specific type of concept $\mathbf{c}$. A list of concepts $[\mathbf{c}_n, \mathbf{c}_h, \mathbf{c}_c, \mathbf{c}_d]$ indicates a specific $f$-divergence objective where it is possible to use, respectively, a null, hyper-class, close, and distant concept. $\text{JS}^*$ indicates that it is the variational formulation of the JS divergence with a change of variable (see Appendix B.1.3).

Machine unlearning studies the fundamental problem of removing specific knowledge learned from a learning algorithm (Cao & Yang, 2015; Bourtoule et al., 2021). In the deep learning era, this problem acquires significant meaning, due to the huge deep learning models that are trained on datasets scraped from the Internet (e.g., LAION-5B (Schuhmann et al., 2022)).

In principle, machine unlearning can be performed following two different approaches: *exact* and *approximate* unlearning. Exact unlearning is considered the gold standard and consists in re-training the learning algorithm from scratch after having removed from the dataset the subset of data to be forgotten. Exact machine unlearning is often infeasible for two reasons: *i*) re-training a large model requires significant computational resources and time, *ii*) the training data could be deleted after their usage for training due to memory reasons, thus not being available for re-training. For these two reasons, researchers are focusing on approximate unlearning methods, which are faster, computationally lighter, and, for generative models, do not require the storage of the training dataset.

Initially, most of the machine unlearning literature was focused on discriminative tasks. For instance, many techniques were developed for classification problems, with the goal of unlearning a specific class or a subset of the dataset (Golatkar et al., 2020; Tarun et al., 2023; Chundawat et al., 2023a;b; Kurmanji et al., 2023; Fan et al., 2024; Foster et al., 2024; Choi & Na, 2025; Spartalis et al., 2025; Bagheri et al., 2025). Notably, $f$-SCRUB (Bagheri et al., 2025) is, to the best of our knowledge, the only existing $f$-divergence-based machine unlearning framework for classification. $f$-SCRUB generalizes SCRUB (Kurmanji et al., 2023) using the $f$-divergence. While for classification there is a large amount of literature related to machine unlearning, most classification techniques either cannot be applied for DMs or have shown poor performance on data generation tasks, thus requiring researchers to develop ad hoc unlearning methods for various generative models, including variational autoencoders and large language models (LLMs) (Heng & Soh, 2024; Wang et al., 2025; Liu et al., 2025). In particular, (Wang et al., 2025) propose to maximize the $f$-divergence between template answers and forget data for LLM unlearning.

In this paper, we focus on the task of erasing concepts from DMs, which is a crucial task due to the widespread usage of DMs for image generation (Nichol et al., 2022; Rombach et al., 2022; Saharia et al., 2022), and due to their memorization capabilities (Somepalli et al., 2023a;b). In the following, we analyze the related work for DM unlearning, also referred to as "concept erasing" or "concept ablating". We categorize the existing techniques into two main groups: post-processing and fine-tuning techniques. Additionally, pre-processing techniques (Nichol et al., 2022) are also used. However, they require dataset curation followed by model training, while in this paper we assume to already have a trained model available.

Post-processing techniques target the elimination of unsafe generated images through the usage of filtering or inference guiding. SD (Rombach et al., 2022) adopts a NSFW filter that filters out all the generated images whose embeddings are close to the embeddings of 17 pre-chosen nudity concepts (Rando et al., 2022). Schramowski et al. (2023) present a method that is applied during inference and that pushes away the generation from unsafe contents. Another example of inference guidance-based approach is SAFREE (Yoon et al., 2025), which can be applied for both image and video generation. The main drawback of post-processing algorithms is that, when the user has access to the model, the post-processing operation

can be removed to allow the model to generate what should have been ablated. Fine-tuning methods, instead, modify the weights of the model, thus being more robust than post-processing approaches when the user is granted access to the unlearned model. In this paper, we focus on fine-tuning methods.

Between fine-tuning methods, divergence-based approaches, formulated as the minimization of a distance measure between probability distributions, are independent on the T2I architecture, thus working for different models, including flow-based generative models (Zhang et al., 2025). This formulation is used as a ground idea in a wide variety of approaches. Erased Stable Diffusion (ESD) (Gandikota et al., 2023) fine-tunes the model by aligning the probability distributions of the model's output fed with a target and a null concept. To achieve that, the authors include in the loss function a classifier-free guidance-based term. Concept Ablation (CAbl) (Kumari et al., 2023) minimizes the MSE between the model's output when the model is fed with a target concept and an anchor hyper-class concept. CAbl's loss function corresponds to $f$-DMU when using the closed-form expression of the KL divergence[2]. This can also be observed from Figure 8, where we schematically depict $f$-DMU and highlight its relationship with existing approaches. Concept-SemiPermeable Membrane (SPM) (Lyu et al., 2024) proposes the usage of adapters that can be shared across different models and that rely on a novel Latent Anchoring (LA) fine-tuning strategy. The proposed loss comprises two terms: *i*) the first term coincides with the ESD loss and matches target and anchor concepts, *ii*) the second term (anchoring loss) minimizes the MSE to preserve the consistency of distant (in the CLIP space) concepts. Reliable concept erasing via Lightweight Erasers (RECELER) (Huang et al., 2024) introduces a new component into the neural network, the eraser, that acts on the cross-attention layers of the U-Net, and that is trained as in (Gandikota et al., 2023), additionally including a concept-localized regularization term, to ensure the effective model performance on local concepts, while leveraging adversarial prompt learning to ensure robustness. Fine-grained attenuation for diffusion erasure (FADE) (Thakral et al., 2025) extracts the concepts that are semantically close to the concept to erase and is trained by minimizing an MSE loss that shifts the target concept to semantically close concepts and preserves the model generation on such a set.

Unified Concept Editing (UCE) (Gandikota et al., 2024) is a closed-form parameter editing method that builds upon (Orgad et al., 2023) and (Meng et al., 2023). UCE updates the cross-attention parameters in closed-form, ensuring a fast computational time. While the loss function is still formulated as the MSE, differently from the KL-based approaches previously presented, UCE minimizes the MSE directly at the cross-attention parameters level. Similarly, RECE (Gong et al., 2024) improves UCE by taking into consideration that the weights modification of UCE is not robust to adversarial prompts. Mass concept erasure (MACE) (Lu et al., 2024) extends UCE (Gandikota et al., 2024) by modifying the cross-attention layers and utilizing Low-Rank Adaptation (LoRA) to remove a large number of concepts. GLoCE (Lee et al., 2025) performs localized erasure through gated LoRA. The major advantage of these methods performing the closed-form update of the weights is the computational complexity. However, the main drawback is that these methods can only be applied to attention weights and specific architectures.

Recently, some methods relying on a double optimization problem have been proposed. Most of them still rely on the KL-minimization idea previously analyzed. AdvUnlearn (Zhang et al., 2024) is an adversarial unlearning method which modifies the text encoder. Although the unlearned text encoder can be used as a plug-and-play robust module for various diffusion models, it needs a retain set. EAP (Bui et al., 2024) is trained with a loss comprising two terms that are being optimized over the model weights and the adversarial concept: the first term minimizes the distance from the concept to erase to the neutral concept, while the second term minimizes the distance from the adversarial concept in the original network to the adversarial concept in the new model. Adaptive Guided Erasure (AGE) (Bui et al., 2025), proposes a bi-level optimization framework that dynamically selects the optimal anchor concepts. Shirkavand et al. (2025) propose a min-max optimization problem that, in addition to minimizing the MSE between target and anchor concepts, includes a strategy for finding the optimal pruning method. The techniques previously discussed rely on the standard KL minimization approach. More concurrent with our work, Domain Correction DoCo (Wu et al., 2025) starts from CA (Kumari et al., 2023) and, in a membership inference attack fashion, replaces the MSE with the JS divergence. DoCo's loss function corresponds to $f$-DMU using the variational representation of the JS divergence (after a change of variable) when $\mathbf{c}^*$ is chosen as the hyper-class concept[3].

**$f$-Divergence**     $f$-divergence-based methods have been effectively employed for the design of objective functions in a large number of applications, such as classification (Novello & Tonello, 2024), classification with label noise (Wei & Liu, 2021; Novello & Tonello, 2025), generation (Nowozin et al., 2016a), semi-supervised learning (Aminian et al., 2024), mutual

---

[2]Here we are referring to the main loss term of both approaches, excluding considerations on the prior preservation terms.

[3]Here we are referring to the main loss term of both approaches, excluding considerations on the prior preservation terms.

information estimation (Letizia et al., 2024), and distillation (Roulet et al., 2025).

For diffusion models, the $f$-divergence has been used for distillation and alignment tasks, but not for solving unlearning problems. (Tang, 2024) uses an $f$-divergence-based entropy regularization term for fine-tuning diffusion models. (Sun et al., 2025) propose an $f$-divergence-based framework for text-to-image models alignment. (Xu et al., 2025) present an $f$-divergence-based approach for variational score distillation, to accelerate the diffusion models sampling process.

For machine unlearning, apart from (Bagheri et al., 2025) for classification and (Wang et al., 2025) for LLMs, general $f$-divergence frameworks have not been proposed. Specific $f$-divergences have only been applied as evaluation metrics. One example is the specific case of the JS divergence (Chundawat et al., 2023a; Choi et al., 2024; Bonato et al., 2024).

In conclusion, to the best of our knowledge, we propose the first diffusion model unlearning framework based on $f$-divergences.

# B. Proofs

## B.1. $f$-Divergence-Based Diffusion Models Unlearning

To motivate (3), it is sufficient to notice that we are tackling the goal of minimizing a divergence measure between the model output distribution conditioned on the target concept $\mathbf{c}^*$ and the anchor concept $\mathbf{c}$, which can be rewritten as

$$\min_{\hat{\Phi}} \mathrm{d}(p_\Phi(\mathbf{x}_{t-1}|\mathbf{x}_t, \mathbf{c}), p_{\hat{\Phi}}(\mathbf{x}_{t-1}|\mathbf{x}_t, \mathbf{c}^*)), \tag{15}$$

which is satisfied $\forall t$ when the output distribution of the unlearned model $\hat{\Phi}$, conditioned on the concept to be unlearned $\mathbf{c}^*$ coincides with the distribution of the original model conditioned on the anchor concept $\mathbf{c}$.

In the seminal work of (Kumari et al., 2023), (15) was obtained (using the KL divergence as divergence metric) by imposing as target goal the equivalence between the entire DM trajectories:

$$D_{KL}(p_\Phi(\mathbf{x}_{(0...T)}|\mathbf{c})||p_{\hat{\Phi}}(\mathbf{x}_{(0...T)}|\mathbf{c}^*)). \tag{16}$$

We report the steps to go from (16) to (15) for completeness, although most of them can also be found in (Kumari et al., 2023). Then, we conclude by generalizing such a result using the $f$-divergence.

Following (Kumari et al., 2023), (16) can be rewritten as

$$D_{KL}(p_\Phi(\mathbf{x}_{(0...T)}|\mathbf{c})||p_{\hat{\Phi}}(\mathbf{x}_{(0...T)}|\mathbf{c}^*))$$
$$= \mathbb{E}_{p(\mathbf{x}_{(0...T)}|\mathbf{c})}\left[\log \frac{\prod_{t=1}^T p_\Phi(\mathbf{x}_{t-1}|\mathbf{x}_t, \mathbf{c})p_\Phi(\mathbf{x}_T)}{\prod_{t=1}^T p_{\hat{\Phi}}(\mathbf{x}_{t-1}|\mathbf{x}_t, \mathbf{c}^*)p_{\hat{\Phi}}(\mathbf{x}_T)}\right]$$
$$= \sum_{\hat{t}=1}^T \mathbb{E}_{p(\mathbf{x}_{(0...T)}|\mathbf{c})}\left[\log \frac{p_\Phi(\mathbf{x}_{\hat{t}-1}|\mathbf{x}_{\hat{t}}, \mathbf{c})}{p_{\hat{\Phi}}(\mathbf{x}_{\hat{t}-1}|\mathbf{x}_{\hat{t}}, \mathbf{c}^*)}\right], \tag{17}$$

where the first step is obtained using the definition of KL divergence and the Markovianity of diffusion processes. In fact,

$$p(\mathbf{x}_{(0...T)}|\mathbf{c}) = p(\mathbf{x}_T|\mathbf{c}) \cdot p(\mathbf{x}_{T-1}|\mathbf{x}_T, \mathbf{c}) \cdot p(\mathbf{x}_{T-2}|\mathbf{x}_{T-1}, \mathbf{x}_T, \mathbf{c}) \cdot p(\mathbf{x}_{T-3}|\mathbf{x}_{T-2}, \mathbf{x}_{T-1}, \mathbf{x}_T, \mathbf{c}) \cdots \tag{18}$$
$$= p(\mathbf{x}_T|\mathbf{c}) \cdot p(\mathbf{x}_{T-1}|\mathbf{x}_T, \mathbf{c}) \cdot p(\mathbf{x}_{T-2}|\mathbf{x}_{T-1}, \mathbf{c}) \cdot p(\mathbf{x}_{T-3}|\mathbf{x}_{T-2}, \mathbf{c}) \cdots \tag{19}$$
$$= p(\mathbf{x}_T) \cdot p(\mathbf{x}_{T-1}|\mathbf{x}_T, \mathbf{c}) \cdot p(\mathbf{x}_{T-2}|\mathbf{x}_{T-1}, \mathbf{c}) \cdot p(\mathbf{x}_{T-3}|\mathbf{x}_{T-2}, \mathbf{c}) \cdots \tag{20}$$
$$= \prod_{t=1}^T p(\mathbf{x}_{t-1}|\mathbf{x}_t, \mathbf{c})p(\mathbf{x}_T), \tag{21}$$

where the second equality is a consequence of the Markovianity of DMs, and the third equality derives from the fact that $\mathbf{x}_T \sim \mathcal{N}(0, 1)$, which is independent from $\mathbf{c}$. From (17), it is possible to study the generic term corresponding to a particular

time step $\hat{t}$

$$\mathbb{E}_{p_\Phi(\mathbf{x}_0...\mathbf{x}_T)}\left[\log\frac{p_\Phi(\mathbf{x}_{\hat{t}-1}|\mathbf{x}_{\hat{t}},\mathbf{c})}{p_{\hat{\Phi}}(\mathbf{x}_{\hat{t}-1}|\mathbf{x}_{\hat{t}},\mathbf{c}^*)}\right]$$

$$=\int_{\mathbf{x}_{(0...T)}}\prod_{t=1}^T p_\Phi(\mathbf{x}_{t-1}|\mathbf{x}_t,\mathbf{c})p(\mathbf{x}_T)\log\frac{p_\Phi(\mathbf{x}_{\hat{t}-1|\mathbf{x}_{\hat{t}}},\mathbf{c})}{p_{\hat{\Phi}}(\mathbf{x}_{\hat{t}-1}|\mathbf{x}_{\hat{t}},\mathbf{c}^*)}d\mathbf{x}_{(0...T)}$$

$$=\int_{\mathbf{x}_{(\hat{t}...T)}}p_\Phi(\mathbf{x}_{(\hat{t}...T)}|\mathbf{c})\left[\int_{\mathbf{x}_{(0...\hat{t}-1)}}\prod_{t=1}^{\hat{t}}p_\Phi(\mathbf{x}_{t-1}|\mathbf{x}_t,\mathbf{c})\log\frac{p_\Phi(\mathbf{x}_{\hat{t}-1}|\mathbf{x}_{\hat{t}},\mathbf{c})}{p_{\hat{\Phi}}(\mathbf{x}_{\hat{t}-1}|\mathbf{x}_{\hat{t}},\mathbf{c}^*)}d\mathbf{x}_{(\hat{t}-1...0)}\right]d\mathbf{x}_{(\hat{t}...T)}$$

$$=\int_{\mathbf{x}_{(\hat{t}...T)}}p_\Phi(\mathbf{x}_{(\hat{t}...T)}|\mathbf{c})\left[\int_{\mathbf{x}_{(0...\hat{t}-1)}}\left(\prod_{t=1}^{\hat{t}-1}p_\Phi(\mathbf{x}_{t-1}|\mathbf{x}_t,\mathbf{c})\right)p_\Phi(\mathbf{x}_{\hat{t}-1}|\mathbf{x}_{\hat{t}},\mathbf{c})\right.$$

$$\left.\cdot\log\frac{p_\Phi(\mathbf{x}_{\hat{t}-1}|\mathbf{x}_{\hat{t}},\mathbf{c})}{p_{\hat{\Phi}}(\mathbf{x}_{\hat{t}-1}|\mathbf{x}_{\hat{t}},\mathbf{c}^*)}d\mathbf{x}_{(\hat{t}-1...0)}\right]d\mathbf{x}_{(\hat{t}...T)}$$

$$=\int_{\mathbf{x}_{\hat{t}}}p_\Phi(\mathbf{x}_{\hat{t}}|\mathbf{c})\left[\int_{\mathbf{x}_{\hat{t}-1}}p_\Phi(\mathbf{x}_{\hat{t}-1}|\mathbf{x}_{\hat{t}},\mathbf{c})\log\frac{p_\Phi(\mathbf{x}_{\hat{t}-1}|\mathbf{x}_{\hat{t}},\mathbf{c})}{p_{\hat{\Phi}}(\mathbf{x}_{\hat{t}-1}|\mathbf{x}_{\hat{t}},\mathbf{c}^*)}\right.$$

$$\left.\cdot\underbrace{\left[\int_{\mathbf{x}_{(0...\hat{t}-2)}}\prod_{t=1}^{\hat{t}-1}p_\Phi(\mathbf{x}_{t-1}|\mathbf{x}_t,\mathbf{c})d\mathbf{x}_{(\hat{t}-2...0)}\right]}_{=1}d\mathbf{x}_{\hat{t}-1}\right]d\mathbf{x}_{\hat{t}}$$

$$=\int_{\mathbf{x}_{\hat{t}}}p_\Phi(\mathbf{x}_{\hat{t}}|\mathbf{c})\left[\int_{\mathbf{x}_{\hat{t}-1}}p_\Phi(\mathbf{x}_{\hat{t}-1}|\mathbf{x}_{\hat{t}},\mathbf{c})\log\frac{p_\Phi(\mathbf{x}_{\hat{t}-1}|\mathbf{x}_{\hat{t}},\mathbf{c})}{p_{\hat{\Phi}}(\mathbf{x}_{\hat{t}-1}|\mathbf{x}_{\hat{t}},\mathbf{c}^*)}d\mathbf{x}_{\hat{t}-1}\right]d\mathbf{x}_{\hat{t}}$$

$$=\mathbb{E}_{p_\Phi(\mathbf{x}_{\hat{t}}|\mathbf{c})}\left[D_{KL}\left(p_\Phi(\mathbf{x}_{\hat{t}-1}|\mathbf{x}_{\hat{t}},\mathbf{c})||p_{\hat{\Phi}}(\mathbf{x}_{\hat{t}-1}|\mathbf{x}_{\hat{t}},\mathbf{c}^*)\right)\right], \tag{22}$$

achieved using the fact that the integral over $p_\Phi(\mathbf{x}_{(\hat{t}...T)}|\mathbf{c})$ corresponds to the integral over $p_\Phi(\mathbf{x}_{\hat{t}}|\mathbf{c})$ because it contains all the information about the image versions from time step $T$ (pure noise) to the current step $\hat{t}$. Since $p_\Phi$ is a Gaussian distribution if the diffusion step sizes are small enough, in (Kumari et al., 2023) the authors propose to use the objective function

$$\sum_{t=1}^T\mathbb{E}_{p_\Phi(\mathbf{x}_t|\mathbf{c})}\left[\eta\left|\left|\Phi(\mathbf{x}_t,\mathbf{c},t)-\hat{\Phi}(\mathbf{x}_t,\mathbf{c}^*,t)\right|\right|_2^2\right], \tag{23}$$

which corresponds to the MSE between the two DMs distributions.

The minimization of the objective function in (22) w.r.t. $\hat{\Phi}$ can be generalized as the minimization of an $f$-divergence between the same probability distributions. Instead of solving (23), we solve

$$\sum_{t=1}^T\mathbb{E}_{p_\Phi(\mathbf{x}_t|\mathbf{c})}\left[D_f\left(p_\Phi(\mathbf{x}_{t-1}|\mathbf{x}_t,\mathbf{c})||p_{\hat{\Phi}}(\mathbf{x}_{t-1}|\mathbf{x}_t,\mathbf{c}^*)\right)\right]. \tag{24}$$

To minimize (24), we propose two different approaches:

- To compute the closed-form expression of a divergence between Gaussian distributions: in Sec. B.1.1, we provide the closed-form expressions of some $f$-divergences between Gaussian distributions. Each of them leads to a specific loss function.

- To use the variational representation of the $f$-divergence: in Sec. B.1.3 we report the variational representations of the $f$-divergences used in the experiments.

### B.1.1. CLOSED-FORM EXPRESSIONS OF THE OBJECTIVE FUNCTIONS FOR SPECIFIC $f$-DIVERGENCES

In the following, we report the main closed-form expressions of $f$-divergences between Gaussian distributions. For each of them, we will provide the information of whether it is a bounded or unbounded $f$-divergence, as this will be related to the boundedness and unboundedness of the gradients. To check the boundedness of $f$-divergences, it is possible to use Theorem B.1. It is first necessary to define

$$f^\star(t) \triangleq t f\left(\frac{1}{t}\right), \tag{25}$$

for all $t > 0$. Furthermore, by definition

$$f^\star(0) = \lim_{u \to \infty} \frac{f(u)}{u}. \tag{26}$$

**Theorem B.1** (Range of values). *(see (Vajda, 1972)) Let $P$ and $Q$ be two probability distributions. Then, the range of an $f$-divergence is given by*

$$0 \le D_f(P||Q) \le f(0) + f^\star(0). \tag{27}$$

In the following, we use $p(x)$ and $q(x)$ as two multivariate normal distributions:

$$p(x) = \frac{1}{\sqrt{(2\pi)^d \det(\Sigma_P)}} \exp\left\{-\frac{1}{2}(x - \mu_P)^T \Sigma_P^{-1}(x - \mu_P)\right\} \tag{28}$$

$$q(x) = \frac{1}{\sqrt{(2\pi)^d \det(\Sigma_Q)}} \exp\left\{-\frac{1}{2}(x - \mu_Q)^T \Sigma_Q^{-1}(x - \mu_Q)\right\}. \tag{29}$$

**Kullback-Leibler divergence**: It is easy to show that the KL divergence between two multivariate Normal distributions reads as

$$D_{KL}(P||Q) = \frac{1}{2}\log\left(\frac{\det\Sigma_Q}{\det\Sigma_P}\right) - \frac{d}{2} + \frac{1}{2}\left[\text{Tr}\left(\Sigma_Q^{-1}\Sigma_P\right) + (\mu_P - \mu_Q)^T \Sigma_Q^{-1}(\mu_P - \mu_Q)\right]. \tag{30}$$

In the scalar case, (30) reduces to

$$D_{KL}(P||Q) = \frac{1}{2}\log\frac{\sigma_Q^2}{\sigma_P^2} - \frac{1}{2} + \frac{1}{2}\left(\frac{(\mu_P - \mu_Q)^2}{\sigma_Q^2} + \frac{\sigma_P^2}{\sigma_Q^2}\right). \tag{31}$$

It is well-known that the KL divergence is unbounded.

The training objective becomes the one proposed by (Kumari et al., 2023), i.e., the MSE between the two diffusion models outputs.

**Jeffreys divergence**: From the definition of Jeffreys divergence, we obtain

$$
\begin{aligned}
D_J(P||Q) = &\frac{1}{2}\log\left(\frac{\det(\Sigma_Q)}{\det(\Sigma_P)}\right) + \frac{1}{2}\log\left(\frac{\det(\Sigma_P)}{\det(\Sigma_Q)}\right) - d \\
&+ \frac{1}{2}\left[\text{Tr}\left(\Sigma_P^{-1}\Sigma_Q\right) + \text{Tr}\left(\Sigma_Q^{-1}\Sigma_P\right)\right. \\
&\left. + (\mu_P - \mu_Q)^T \Sigma_Q^{-1}(\mu_P - \mu_Q) + (\mu_Q - \mu_P)^T \Sigma_P^{-1}(\mu_Q - \mu_P)\right],
\end{aligned}
\tag{32}
$$

which simplifies to

$$
\begin{aligned}
D_J(P||Q) = &-d + \frac{1}{2}\left[\text{Tr}\left(\Sigma_Q^{-1}\Sigma_P\right) + \text{Tr}\left(\Sigma_P^{-1}\Sigma_Q\right)\right. \\
&\left. + (\mu_P - \mu_Q)^T\left(\Sigma_Q^{-1} + \Sigma_P^{-1}\right)(\mu_P - \mu_Q)\right].
\end{aligned}
\tag{33}
$$

In the scalar case, (33) becomes

$$D_J(P\|Q) = -1 + \frac{1}{2}\left[\frac{\sigma_P^4 + \sigma_Q^4}{\sigma_Q^2\sigma_P^2} + \frac{(\mu_P - \mu_Q)^2}{\sigma_P^2\sigma_Q^2}(\sigma_P^2 + \sigma_Q^2)\right]. \tag{34}$$

Since the Jeffreys divergence is the sum of two KL divergences, it is unbounded.

Under the assumption $\sigma_P = \sigma_Q = \sigma$, the training objective becomes

$$\mathcal{J}_J(\mathbf{x}, \mathbf{c}, \mathbf{c}^*) = \mathbb{E}_{\epsilon, \mathbf{x}, \mathbf{c}^*, \mathbf{c}, t}\left[\frac{\omega_t}{\sigma^2}\|\Phi(\mathbf{x}_t, \mathbf{c}, t) - \hat{\Phi}(\mathbf{x}_t, \mathbf{c}^*, t)\|_2^2\right]. \tag{35}$$

**Squared Hellinger distance**: The squared Hellinger distance can be expressed as a function of the Bhattacharyya coefficient $(BC(P,Q))$ as $\mathrm{H}^2(P,Q) = 1 - BC(P,Q)$. From (Pardo, 2018), the squared Hellinger distance between two multivariate Normal distributions reads as

$$\mathrm{H}^2(P,Q) = 1 - \frac{\det(\Sigma_P)^{\frac{1}{4}}\det(\Sigma_Q)^{\frac{1}{4}}}{\det(\frac{\Sigma_P + \Sigma_Q}{2})^{\frac{1}{2}}}\exp\left\{-\frac{1}{8}(\mu_P - \mu_Q)^T\left(\frac{\Sigma_P + \Sigma_Q}{2}\right)^{-1}(\mu_P - \mu_Q)\right\}. \tag{36}$$

In the scalar case, (36) becomes

$$\mathrm{H}^2(P,Q) = 1 - \frac{\sqrt{\sigma_P\sigma_Q}}{\sqrt{\frac{\sigma_P^2 + \sigma_Q^2}{2}}}\exp\left\{-\frac{1}{4}\frac{(\mu_P - \mu_Q)^2}{\sigma_P^2 + \sigma_Q^2}\right\}. \tag{37}$$

The $\mathrm{H}^2$ is bounded. More precisely,

$$0 \leq \mathrm{H}^2(P,Q) \leq 1, \tag{38}$$

which can be proven using the range of values theorem (Vajda, 1972). A quick intuition of the correctness of (38) can be derived by noticing that when $\mu_P = \mu_Q$ and $\sigma_P = \sigma_Q$, $\mathrm{H}^2(P,Q) = 0$, while $\mathrm{H}^2(P,Q) \to 1$ when $(\mu_P - \mu_Q)^2 \to \infty$.

Assuming $\sigma_P = \sigma_Q = \sigma$, the training objective becomes

$$\mathcal{J}_{\mathrm{H}}(\mathbf{x}, \mathbf{c}, \mathbf{c}^*) = \mathbb{E}_{\epsilon, \mathbf{x}, \mathbf{c}^*, \mathbf{c}, t}\left[-\omega_t\exp\left\{-\frac{1}{8\sigma^2}\|\Phi(\mathbf{x}_t, \mathbf{c}, t) - \hat{\Phi}(\mathbf{x}_t, \mathbf{c}^*, t)\|_2^2\right\}\right]. \tag{39}$$

$\chi^2$ **divergence**: The $\chi^2$ divergence is defined as

$$\chi^2(P\|Q) = \int_{\mathcal{X}}\frac{(p(x) - q(x))^2}{q(x)}dx = \int\frac{p(x)^2}{q(x)}dx - 1. \tag{40}$$

Substituting the expressions of the pdfs of Gaussian random variables, it becomes

$$\chi^2(P\|Q) = \int\frac{\sigma_Q}{\sqrt{2\pi\sigma_P^2}}e^{-\frac{1}{2}\left(\frac{2(x-\mu_P)^2}{\sigma_P^2} - \frac{(x-\mu_Q)^2}{\sigma_Q^2}\right)}dx - 1 \tag{41}$$

$$= \frac{\sigma_Q^2}{\sigma_P\sqrt{2\sigma_Q^2 - \sigma_P^2}}\exp\left\{\frac{(\mu_P - \mu_Q)^2}{2\sigma_Q^2 - \sigma_P^2}\right\} - 1. \tag{42}$$

The expression in (42) holds only for $2\sigma_Q^2 > \sigma_P^2$.

The $\chi^2$ divergence is unbounded because $f^*(0) = \infty$.

Under the assumption that $\sigma_P = \sigma_Q = \sigma$,

$$\mathcal{J}_{\chi^2}(\mathbf{x}, \mathbf{c}, \mathbf{c}^*) = \mathbb{E}_{\epsilon, \mathbf{x}, \mathbf{c}^*, \mathbf{c}, t}\left[\omega_t\exp\left\{\frac{1}{\sigma^2}\|\Phi(\mathbf{x}_t, \mathbf{c}, t) - \hat{\Phi}(\mathbf{x}_t, \mathbf{c}^*, t)\|_2^2\right\}\right]. \tag{43}$$

$\alpha$-**divergence**  We provide a general loss function derived from the closed-form expression of a subclass of $f$-divergences: the $\alpha$-divergence (Amari, 1985). The $\alpha$-divergence between two probability distributions $p(\mathbf{x})$ and $q(\mathbf{x})$ is defined as (Read & Cressie, 2012)

$$D_\alpha(P||Q) = \frac{1}{\alpha(\alpha-1)}\left(\int_{-\infty}^{\infty} p(\mathbf{x})^\alpha q(\mathbf{x})^{1-\alpha} d\mathbf{x} - 1\right), \tag{44}$$

where $\alpha \in \mathbb{R} \setminus \{0, 1\}$. The $\alpha$-divergence is a specific subclass of $f$-divergences obtained by imposing the generator function as

$$f(u) = \begin{cases} \frac{1}{\alpha(\alpha-1)}(u^\alpha - 1 - \alpha(u-1)) & \text{for } \alpha \notin \{0,1\} \\ u\log u & \text{for } \alpha = 1 \\ -\log u & \text{for } \alpha = 0 \end{cases}. \tag{45}$$

As it is clear from (45), when $\alpha = 1$, we get the KL divergence, when $\alpha = 0$, we obtain the RKL divergence. Furthermore, when $\alpha = 1/2$, we attain the squared Hellinger distance, while when $\alpha = 2$, we get the Pearson $\chi^2$ divergence.

In general, by varying $\alpha$, we obtain different divergences with different properties. In fact, similarly to the discussion in Sec. 4, it is possible to identify which $\alpha$-divergences have mode covering or mode seeking properties.

The whole class of $\alpha$-divergences allows an analytical characterization when the probability density functions are Gaussian distributions $P \sim \mathcal{N}(\mu_P, \sigma_P^2)$ and $Q \sim \mathcal{N}(\mu_Q, \sigma_Q^2)$ (Sourla et al., 2024):

$$D_\alpha(P||Q) = \frac{1}{\alpha(1-\alpha)}(1 - H_\alpha(P, Q)) \tag{46}$$

$$H_\alpha(P||Q) = \frac{\sigma_P^{1-\alpha}\sigma_Q^\alpha}{\sqrt{(1-\alpha)\sigma_P^2 + \alpha\sigma_Q^2}} e^{-\frac{\alpha(1-\alpha)(\mu_P-\mu_Q)^2}{2((1-\alpha)\sigma_P^2 + \alpha\sigma_Q^2)}}. \tag{47}$$

In general, the $\alpha$-divergence between Gaussian distributions is real only when $\alpha \in [0, 1]$. For $\alpha > 1$, (47) is a real-valued function when $\sigma_P^2 < \frac{\alpha}{\alpha-1}\sigma_Q^2$, while for $\alpha < 0$ (47) is a real-valued function when $\sigma_P^2 > \frac{\alpha}{\alpha-1}\sigma_Q^2$.

In our scenario, $\sigma_P = \sigma_Q = \sigma$, thus the above conditions are always verified, and the $\alpha$-divergence becomes

$$D_\alpha(P||Q) = \frac{1}{\alpha(1-\alpha)}\left(1 - e^{-\frac{\alpha(1-\alpha)(\mu_P-\mu_Q)^2}{2\sigma^2}}\right). \tag{48}$$

From (48), we can obtain the loss function corresponding to a general $\alpha$-divergence as

$$\mathcal{J}_\alpha(\mathbf{x}, \mathbf{c}, \mathbf{c}^*) = \mathbb{E}_{\epsilon, \mathbf{x}, \mathbf{c}^*, \mathbf{c}, t}\left[-\omega_t \exp\left\{-\frac{\alpha(1-\alpha)}{2\sigma^2}||\Phi(\mathbf{x}_t, \mathbf{c}, t) - \hat{\Phi}(\mathbf{x}_t, \mathbf{c}^*, t)||_2^2\right\}\right]. \tag{49}$$

### B.1.2. GRADIENT ANALYSIS

In this section, we analyze the gradients of the different closed-form loss functions presented in Section B.1.1.

**Squared Hellinger distance and Pearson divergence**  Let $\hat{\Phi}$ be a function of the parameters vector $\phi$, then

$$\sum_{i=1}^{n} \frac{\partial \mathcal{J}_f(\phi)}{\partial \phi} = \begin{cases} \sum_{i=1}^{n} \nabla_\phi \text{MSE}(\Phi(\mathbf{x}_i, \mathbf{c}, i), \hat{\Phi}(\mathbf{x}_i, \mathbf{c}^*, i)) & \text{for KL} \\ \sum_{i=1}^{n} e^{-\text{MSE}(\Phi(\mathbf{x}_i, \mathbf{c}, i), \hat{\Phi}(\mathbf{x}_i, \mathbf{c}^*, i))} \nabla_\phi \text{MSE}(\Phi(\mathbf{x}_i, \mathbf{c}, i), \hat{\Phi}(\mathbf{x}_i, \mathbf{c}^*, i)) & \text{for H}^2 \\ \sum_{i=1}^{n} e^{\text{MSE}(\Phi(\mathbf{x}_i, \mathbf{c}, i), \hat{\Phi}(\mathbf{x}_i, \mathbf{c}^*, i))} \nabla_\phi \text{MSE}(\Phi(\mathbf{x}_i, \mathbf{c}, i), \hat{\Phi}(\mathbf{x}_i, \mathbf{c}^*, i)) & \text{for } \chi^2 \end{cases}. \tag{50}$$

Since all the losses depend on the gradient of the MSE, it is easy to compare the gradients of the $i$-th sample:

- The gradients of $\text{H}^2$ correspond to weighted versions of the gradient of MSE. When $\text{MSE} \to 0$, the gradients of $\text{H}^2$ coincide with the gradients of MSE. Meanwhile, when $\text{MSE} \to \infty$, the gradients of $\text{H}^2$ tend to 0.

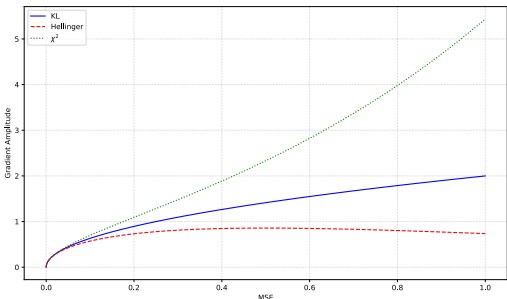

*Figure 9.* Gradients comparison between MSE, squared Hellinger distance, and Pearson divergence, as a function of the MSE. All the functions are plotted up to a multiplicative factor.

- Also for $\chi^2$ divergence the gradients correspond to weighted versions of the gradient of MSE. However, they have an opposite behavior compared to the $H^2$ gradients. When MSE $\to 0$, the gradients coincide with the MSE gradients, while when MSE $\to \infty$, the gradients grow to $\infty$.

As reported in (8), a component-wise comparison shows that $\left| \frac{\partial \mathcal{J}_{H^2}(\phi)}{\partial \phi} \right| \leq \left| \frac{\partial \mathcal{J}_{KL}(\phi)}{\partial \phi} \right| \leq \left| \frac{\partial \mathcal{J}_{\chi^2}(\phi)}{\partial \phi} \right|$, where the inequalities become equalities when the MSE is zero.

To provide a visual representation, we explicit the dependence of the MSE gradient on the difference between $\Phi$ and $\hat{\Phi}$:

$$\nabla_{\phi}\mathcal{J}_{KL} = -2(\Phi - \hat{\Phi})\nabla_{\phi}\hat{\Phi} \tag{51}$$

$$\nabla_{\phi}\mathcal{J}_{H^2} = -2e^{-(\Phi - \hat{\Phi})^2}(\Phi - \hat{\Phi})\nabla_{\phi}\hat{\Phi} \tag{52}$$

$$\nabla_{\phi}\mathcal{J}_{\chi^2} = -2e^{(\Phi - \hat{\Phi})^2}(\Phi - \hat{\Phi})\nabla_{\phi}\hat{\Phi} \tag{53}$$

and we depict in Figure 9 their behaviors. To isolate the impact of the choice of $f$-divergence on the optimization, Fig. 9 shows the gradient magnitude by factoring out the common model-dependent Jacobian $\nabla_{\phi}\hat{\Phi}$.

The two opposite behaviors of $H^2$ and $\chi^2$ divergences lead, in practice, to two very different behaviors during training. It appears that the $\chi^2$ divergence focuses more on those cases where the generated images with target and anchor concept are significantly different. This behavior is similar to the one obtained using the KL divergence, with the difference that the gradients of the $\chi^2$ divergence grow significantly faster. Meanwhile, the loss derived from the closed-form of the $H^2$ divergence shows a peculiar behavior as it is less affected from outliers: it weights more the samples that have an intermediate MSE value.

Notably, both the KL and $\chi^2$ divergences are unbounded and have unbounded gradients. In contrast, the $H^2$ distance leads to a bounded loss function and bounded gradients.

$\alpha$-**divergence**   The gradient of the loss in (49) is

$$\frac{\partial \mathcal{J}_{\alpha}(\phi)}{\partial \phi} = \frac{\alpha(1-\alpha)}{2\sigma^2} e^{-\frac{\alpha(1-\alpha)}{2\sigma^2}\text{MSE}(\Phi(\mathbf{x}_i, \mathbf{c}, i), \hat{\Phi}(\mathbf{x}_i, \mathbf{c}^*, i))} \nabla_{\phi}\text{MSE}(\Phi(\mathbf{x}_i, \mathbf{c}, i), \hat{\Phi}(\mathbf{x}_i, \mathbf{c}^*, i)). \tag{54}$$

By rewriting the gradient of the MSE as previously done for KL, $H^2$, and $\chi^2$, we report in Fig. 10 the gradient amplitude for different values of $\alpha$ and MSE: Fig. 10a shows the behavior when $\alpha \in (0, 1)$, while Fig. 10b represents $\alpha > 1$. The two plots show a distinctive behavior for each value of $\alpha$. While the squared Hellinger distance ($\alpha = 0.5$) has bounded gradients, it achieves the highest gradient amplitude compared to all the $\alpha$-divergences with $\alpha \in (0, 1)$, but also shows the steeper descent for increasing MSE values. This implies that, within the class of $\alpha$-divergences with $\alpha \in (0, 1)$, the squared Hellinger distance is more affected by cases in which the MSE is medium-low, while it is less affected by samples where the MSE is high. The magnitude of the gradients of $H^2$ in Fig. 10 slightly differs from the magnitude in Fig. 9 for a multiplicative factor that we excluded in Fig. 9 and because we focus on a smaller range of MSE to have a cleaner visualization that depicts H-DMU and $\chi^2$-DMU in the same axes. For $\alpha > 1$, although the Pearson $\chi^2$ divergence (corresponding to $\alpha = 2$) is characterized by an exponentially increasing gradient magnitude, we can observe that the more we increase $\alpha$, the more the corresponding divergence is characterized by larger gradient amplitude.

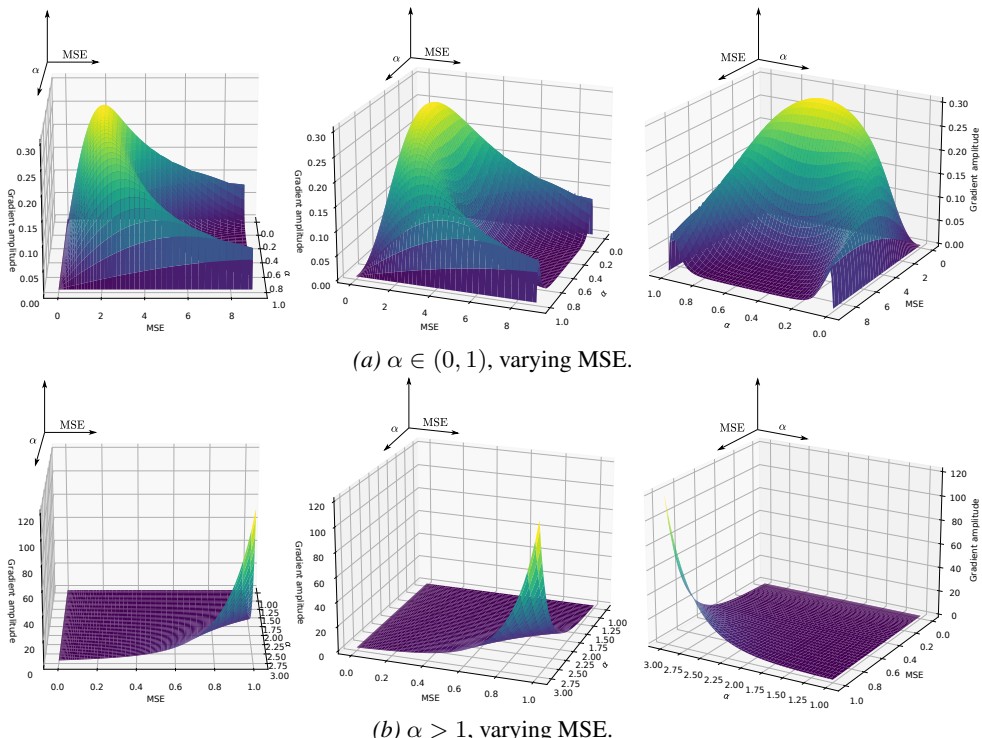

*(a)* $\alpha \in (0, 1)$, varying MSE.

*(b)* $\alpha > 1$, varying MSE.

*Figure 10.* Gradient amplitude varying $\alpha$ and MSE. The plots in (a) also shows the gradient amplitude of the squared Hellinger distance, for $\alpha = 0.5$. Meanwhile, the plots in (b) also comprise the gradient amplitude of the Pearson $\chi^2$ divergence, for $\alpha = 2$. For (b), the MSE range is more limited as the gradient amplitude grows exponentially fast.

### B.1.3. VARIATIONAL REPRESENTATION

In this section, we report the variational-based version of $f$-DMU.

For simplicity in the notation, we will write $T(\Phi)$ to indicate $T(\Phi(\mathbf{x}_t, \mathbf{c}))$. The variational representation of the $f$-divergence would rewrite (3) as

$$\min_{\hat{\Phi}} \sum_{t=1}^{T} \mathbb{E}_{p_{\Phi}(\mathbf{x}_t | \mathbf{c})} \left[ \sup_{T} \left\{ \mathbb{E}_{p_{\Phi}(\mathbf{x}_{t-1} | \mathbf{x}_t, \mathbf{c})} \left[ T(\Phi) \right] - \mathbb{E}_{p_{\hat{\Phi}}(\mathbf{x}_{t-1} | \mathbf{x}_t, \mathbf{c}^*)} \left[ f^*(T(\hat{\Phi})) \right] \right\} \right]. \tag{55}$$

Instead, we propose to solve the following problem:

$$\min_{\hat{\Phi}} D_f \left( p_{\Phi}(\mathbf{x}_{t-1} | \mathbf{x}_t, \mathbf{c}) || p_{\hat{\Phi}}(\mathbf{x}_{t-1} | \mathbf{x}_t, \mathbf{c}^*) \right) \quad \forall \mathbf{x}_t, \tag{56}$$

which is coherent with the minimization of the $f$-divergence between the entire generation trajectories and which corresponds to solving the following objective function

$$\min_{\hat{\Phi}} \max_{T} \mathbb{E}_{\mathbf{x}, \mathbf{c}^*, \mathbf{c}, t} \left[ \mathbb{E}_{p_{\Phi}(\mathbf{x}_{t-1} | \mathbf{x}_t, \mathbf{c})} \left[ T(\Phi) \right] - \mathbb{E}_{p_{\hat{\Phi}}(\mathbf{x}_{t-1} | \mathbf{x}_t, \mathbf{c}^*)} \left[ f^*(T(\hat{\Phi})) \right] \right]. \tag{57}$$

Following the work in (Nowozin et al., 2016b), we assume $T(x) = g_f(V_\omega(x))$, to respect the domain of the Fenchel conjugate function, i.e., $V : \mathcal{X} \to \mathbb{R}$ and $g_f : \mathbb{R} \to dom_{f^*}$. Thus, (57) becomes

$$\min_{\hat{\Phi}} \max_{V} \mathbb{E}_{\mathbf{x}, \mathbf{c}^*, \mathbf{c}, t} \left[ \mathbb{E}_{p_{\Phi}(\mathbf{x}_{t-1} | \mathbf{x}_t, \mathbf{c})} \left[ g_f(V(\Phi)) \right] - \mathbb{E}_{p_{\hat{\Phi}}(\mathbf{x}_{t-1} | \mathbf{x}_t, \mathbf{c}^*)} \left[ f^*(g_f(V(\hat{\Phi}))) \right] \right]. \tag{58}$$

The generator functions and corresponding Fenchel conjugates of the $f$-divergences used in this paper are reported in Tab. 5. In addition, we report the corresponding output activation functions $g_f$.

*Table 5.* $f$-divergences table.

| NAME | $f(u)$ | $f^*(t)$ | $g_f$ |
|---|---|---|---|
| KULLBACK-LEIBLER | $u\log(u)$ | $e^{t-1}$ | $v$ |
| REVERSE KULLBACK-LEIBLER | $-\log(u)$ | $-1-\log(-t)$ | $-e^{-v}$ |
| SQUARED HELLINGER | $(\sqrt{u}-1)^2$ | $\frac{t}{1-t}$ | $1-e^{-v}$ |
| JENSEN-SHANNON | $u\log(u)-(u+1)\log\left(\frac{u+1}{2}\right)$ | $-\log(2-e^t)$ | $\log(2)-\log(1+e^{-v})$ |
| GAN | $u\log(u)-(u+1)\log(u+1)$ | $-\log(1-e^t)$ | $-\log(1+e^{-v})$ |
| $\chi^2$ | $(u-1)^2$ | $\frac{1}{4}t^2+t$ | $v$ |
| JEFFREYS | $(u-1)\log u$ | $W(e^{1-t})+\frac{1}{W(e^{1-t})}+t-2$ | $v$ |
| TOTAL VARIATION | $\frac{1}{2}|u-1|$ | $t$ | $\frac{1}{2}\tanh(v)$ |

**KL divergence**   The loss function corresponding to the variational representation of the KL divergence reads as

$$\min_{\hat{\Phi}} \max_{V} \mathbb{E}_{\mathbf{x},\mathbf{c}^*,\mathbf{c},t}\left[\mathbb{E}_{p_\Phi(\mathbf{x}_{t-1}|\mathbf{x}_t,\mathbf{c})}\left[V(\Phi)\right] - \mathbb{E}_{p_{\hat{\Phi}}(\mathbf{x}_{t-1}|\mathbf{x}_t,\mathbf{c}^*)}\left[e^{V(\hat{\Phi})-1}\right]\right]. \tag{59}$$

**H$^2$ distance**   The loss function corresponding to the variational representation of the squared Hellinger distance reads as

$$\min_{\hat{\Phi}} \max_{V} \mathbb{E}_{\mathbf{x},\mathbf{c}^*,\mathbf{c},t}\left[\mathbb{E}_{p_\Phi(\mathbf{x}_{t-1}|\mathbf{x}_t,\mathbf{c})}\left[1-e^{-V(\Phi)}\right] - \mathbb{E}_{p_{\hat{\Phi}}(\mathbf{x}_{t-1}|\mathbf{x}_t,\mathbf{c}^*)}\left[\frac{1-e^{-V(\hat{\Phi})}}{e^{-V(\hat{\Phi})}}\right]\right]. \tag{60}$$

**JS divergence**   The loss function corresponding to the variational representation of the JS divergence reads as

$$\min_{\hat{\Phi}} \max_{V} \mathbb{E}_{\mathbf{x},\mathbf{c}^*,\mathbf{c},t}\left[\mathbb{E}_{p_\Phi(\mathbf{x}_{t-1}|\mathbf{x}_t,\mathbf{c})}\left[-\log\left(1+e^{-V(\Phi)}\right)\right] + \mathbb{E}_{p_{\hat{\Phi}}(\mathbf{x}_{t-1}|\mathbf{x}_t,\mathbf{c}^*)}\left[\log\left(\frac{e^{-V(\hat{\Phi})}}{1+e^{-V(\hat{\Phi})}}\right)\right]\right]. \tag{61}$$

Using the change of variable $T = -\log(1+e^{-v}) = \log(D)$, we obtain

$$\min_{\hat{\Phi}} \max_{V} \mathbb{E}_{\mathbf{x},\mathbf{c}^*,\mathbf{c},t}\left[\mathbb{E}_{p_\Phi(\mathbf{x}_{t-1}|\mathbf{x}_t,\mathbf{c})}\left[-\log\left(1+e^{-V(\Phi)}\right)\right] + \mathbb{E}_{p_{\hat{\Phi}}(\mathbf{x}_{t-1}|\mathbf{x}_t,\mathbf{c}^*)}\left[\log\left(\frac{e^{-V(\hat{\Phi})}}{1+e^{-V(\hat{\Phi})}}\right)\right]\right]. \tag{62}$$

Thus, we observe that DoCo's objective function corresponds to the variational representation of the JS divergence after a specific change of variable.

## B.2. Local Convergence Study

In this section, we present in Sec. B.2.1 the necessary preliminaries to develop the local convergence study, and in Sec. B.2.3 the proofs of the local convergence of the algorithm proposed.

### B.2.1. PRELIMINARIES

Let us consider a system consisting of variables $\theta \in \Theta \subseteq \mathbb{R}^n$. Let its time derivative be defined by the vector field $v(\phi)$:

$$\dot{\phi} = v(\phi), \tag{63}$$

where $v : \Phi \to \mathbb{R}^n$ is a locally Lipschitz mapping from a domain $\Phi$ into $\mathbb{R}^n$.

Assume $\phi^*$ is an equilibrium point of the system in (63) (i.e., $v(\phi^*) = 0$). Let $\phi_t$ be the state of the system at time $t$.

**Definition B.2.**  (see (Hassan K, 1996)) The equilibrium point $\phi^*$ for the system defined in (63) is

- stable if for each $\epsilon > 0$, there is $\delta = \delta(\epsilon) > 0$ such that

$$||\phi_0 - \phi^*|| < \delta \implies \forall t \geq 0, ||\phi_t - \phi^*|| < \epsilon \tag{64}$$

- unstable if not stable

- asymptotically stable if it is stable and $\delta > 0$ can be chosen such that

$$||\phi_0 - \phi^*|| < \delta \implies \lim_{t \to \infty} \phi_t = \phi^* \tag{65}$$

- exponentially stable if it is asymptotically stable and $\delta, k, \lambda > 0$ can be chosen such that:

$$||\phi_0 - \phi^*|| < \delta \implies ||\phi_t|| \le k||\phi_0|| \exp\{(-\lambda t)\}. \tag{66}$$

The definition of stability can imply two different situations: 1) the systems converges to the equilibrium point, 2) the system is restricted to a ball of radius $\epsilon$ from the equilibrium. The asymptotic stability, instead, guarantees that the system reaches the equilibrium, when the initial condition belongs to a $\delta$ neighborhood of the equilibrium.

Theorem B.3 associates the stability of a non-linear system in the equilibrium point with its Jacobian, allowing to study the non-linear system by evaluating the eigenvalues of $\mathbf{J}$.

**Theorem B.3.** *(see (Hassan K, 1996)) Let $\phi^*$ be an equilibrium point for the non-linear system*

$$\dot{\phi} = v(\phi) \tag{67}$$

*where $v : \Phi \to \mathbb{R}^n$ is continuously differentiable and $\Phi$ is a neighborhood of $\phi^*$. Let $\mathbf{J}$ be the Jacobian of the system in (9) at the equilibrium point:*

$$\mathbf{J} = \left. \frac{\partial v(\phi)}{\partial \phi} \right|_{\phi = \phi^*}. \tag{68}$$

*Then, we have:*

- *The equilibrium point $\phi^*$ is asymptotically stable and exponentially stable if $\mathbf{J}$ is a Hurwitz matrix, i.e., $Re(\lambda) < 0$ for all eigenvalues $\lambda$ of $\mathbf{J}$.*

- *The equilibrium point $\phi^*$ is unstable if $Re(\lambda) > 0$ for one or more of the eigenvalues of $\mathbf{J}$.*

Theorem B.4 upper bounds the eigenvalues of a Jacobian as in (69). Theorem B.4 is used in (Nagarajan & Kolter, 2017) to prove the local convergence of GANs, and we use it in this paper to prove the local convergence of the proposed algorithm.

**Theorem B.4.** *(see Lemma G.2 in (Nagarajan & Kolter, 2017)) Let $\lambda_m$ and $\lambda_M$ denote the smallest and largest eigenvalues of a matrix, respectively. Suppose $\mathbf{J} \in \mathbb{R}^{(m+n) \times (m+n)}$ is of the following form:*

$$\mathbf{J} = \begin{bmatrix} \mathbf{0} & \mathbf{P} \\ -\mathbf{P}^T & -\mathbf{Q} \end{bmatrix} \tag{69}$$

*where $\mathbf{Q} \in \mathbb{R}^{n \times n}$ is a symmetric real positive definite matrix and $\mathbf{P}^T \in \mathbb{R}^{n \times m}$ is a full column rank matrix. Then, for every eigenvalue $\lambda$ of $\mathbf{J}$, $Re(\lambda) < 0$. More precisely, we have:*

- *When $Im(\lambda) = 0$,*

$$Re(\lambda) \le -\frac{\lambda_m(\mathbf{Q})\lambda_m(\mathbf{PP}^T)}{\lambda_m(\mathbf{Q})\lambda_M(\mathbf{Q}) + \lambda_m(\mathbf{PP}^T)}, \tag{70}$$

- *When $Im(\lambda) \ne 0$,*

$$Re(\lambda) \le -\frac{\lambda_m(\mathbf{Q})}{2}. \tag{71}$$

Lemma B.5 reports a known relationship between the generator function of an $f$-divergence and its Fenchel conjugate. This relationship is useful in our Jacobian study.

**Lemma B.5.** *Let $f : \mathbb{R}_+ \longrightarrow \mathbb{R}$ be the generator function of any $f$-divergence. Let $f^*$ be the Fenchel conjugate of $f$. Then,*

$$(f^*)'(t) = (f')^{-1}(t), \tag{72}$$

*where $f'$ indicates the first derivative of the generator function.*

*Proof.* Since $f(\cdot)$ is a convex function, $\hat{u}$ achieving the supremum is

$$\hat{u} = (f')^{-1}(t). \tag{73}$$

Then, substituting (73) in the definition of the Fenchel conjugate, we obtain

$$f^*(t) = (f')^{-1}(t)t - f((f')^{-1}(t)). \tag{74}$$

The first derivative reads as

$$(f^*)'(t) = ((f')^{-1})'(t)t + (f')^{-1}(t) - \underbrace{f'((f')^{-1}(t))}_{=t}((f')^{-1})'(t). \tag{75}$$

The first and third terms cancel out, leading to (72). $\qquad\square$

### B.2.2. NOTATION AND SETUP

Let us define

$$\min_{\boldsymbol{\phi}} \max_{\boldsymbol{\omega}} \mathcal{J}_f^v(\boldsymbol{\phi}, \boldsymbol{\omega}) = \min_{\boldsymbol{\phi}} \max_{\boldsymbol{\omega}} \mathbb{E}_{\mathbf{x}, \mathbf{c}^*, \mathbf{c}, t}\left[\mathbb{E}_{p_\Phi(\mathbf{x}_{\hat{t}-1}|\mathbf{x}_{\hat{t}}, \mathbf{c})}\left[T_{\boldsymbol{\omega}}(\Phi)\right] - \mathbb{E}_{p_\phi(\mathbf{x}_{\hat{t}-1}|\mathbf{x}_{\hat{t}}, \mathbf{c}^*)}\left[f^*(T_{\boldsymbol{\omega}}(\hat{\Phi}))\right]\right]. \tag{76}$$

Let us define the pair $(\boldsymbol{\phi}^*, \boldsymbol{\omega}^*)$ as the equilibrium point achieved when solving (76).

**Assumption B.6.** $f^*(\cdot)$ is a strictly convex function.

Assumption B.6 excludes from the set of $f$-divergences few specific cases that we do not use in this paper. For instance, the Total Variation distance does not satisfy Assumption B.6, as $f^*(t) = t$. Meanwhile, the KL divergence, Reverse KL (RKL) divergence, $\chi^2$ divergence, squared Hellinger distance, Jeffreys divergence, and Jensen-Shannon divergence satisfy Assumption B.6.

**Assumption B.7.** $\exists \boldsymbol{\phi}^*, \boldsymbol{\omega}^*$ such that $\forall \mathbf{x}_{t-1} \in \text{supp}(p), p_{\boldsymbol{\phi}^*}(\mathbf{x}_{t-1}|\mathbf{x}_t, \mathbf{c}^*) = p_\Phi(\mathbf{x}_{t-1}|\mathbf{x}_t, \mathbf{c})$ and $T_{\boldsymbol{\omega}^*} = f'(p_\Phi/p_\phi)$.

Assumption B.7 is achieved when we reach the optimal convergence, and implies that, at the equilibrium,

$$T_{\boldsymbol{\omega}^*} = f'(p_\Phi/p_{\boldsymbol{\phi}^*}) = f'(1), \tag{77}$$

thus

$$(f^*)'(T_{\boldsymbol{\omega}^*}) = 1, \tag{78}$$

because $(f^*)'(t) = (f')^{-1}(t)$ (from Lemma B.5).

**Assumption B.8.** $\mathbb{E}_{\mathbf{x}, \mathbf{c}^*, \mathbf{c}, \hat{t}}\left[\mathbb{E}_{p_\Phi(\mathbf{x}_{\hat{t}-1}|\mathbf{x}_{\hat{t}}, \mathbf{c})}\left[-\nabla_{\boldsymbol{\phi}} \log\left(p_\phi(\mathbf{x}_{\hat{t}-1}|\mathbf{x}_{\hat{t}}, \mathbf{c}^*)\right)\left(\nabla_{\boldsymbol{\omega}}^T T_{\boldsymbol{\omega}}\right)\right]\right]\Big|_{(\boldsymbol{\phi}^*, \boldsymbol{\omega}^*)}$ and
$\mathbb{E}_{\mathbf{x}, \mathbf{c}^*, \mathbf{c}, \hat{t}}\left[\mathbb{E}_{p_\Phi(\mathbf{x}_{\hat{t}-1}|\mathbf{x}_{\hat{t}}, \mathbf{c})}\left(\nabla_{\boldsymbol{\omega}} T_{\boldsymbol{\omega}} \nabla_{\boldsymbol{\omega}}^T T_{\boldsymbol{\omega}}\right)\right]\Big|_{\boldsymbol{\omega}^*}$ are full row rank.

Assumption B.8 is similar to the assumptions used in (Nagarajan & Kolter, 2017; Yu et al., 2020a). In particular,

$$\mathbb{E}_{\mathbf{x}, \mathbf{c}^*, \mathbf{c}, \hat{t}}\left[\mathbb{E}_{p_\Phi(\mathbf{x}_{\hat{t}-1}|\mathbf{x}_{\hat{t}}, \mathbf{c})}\left[-\nabla_{\boldsymbol{\phi}} \log\left(p_\phi(\mathbf{x}_{\hat{t}-1}|\mathbf{x}_{\hat{t}}, \mathbf{c}^*)\right)\left(\nabla_{\boldsymbol{\omega}}^T T_{\boldsymbol{\omega}}\right)\right]\right]\Big|_{(\boldsymbol{\phi}^*, \boldsymbol{\omega}^*)}$$

$$= \mathbb{E}_{\mathbf{x}, \mathbf{c}^*, \mathbf{c}, \hat{t}}\left[-\int_{\mathbf{x}_{\hat{t}-1}} \nabla_{\boldsymbol{\phi}} p_\phi(\mathbf{x}_{\hat{t}-1}|\mathbf{x}_{\hat{t}}, \mathbf{c}^*)\left(\nabla_{\boldsymbol{\omega}}^T T_{\boldsymbol{\omega}}\right) d\mathbf{x}_{\hat{t}-1}\right]\Big|_{(\boldsymbol{\phi}^*, \boldsymbol{\omega}^*)}. \tag{79}$$

B.2.3. PROOFS OF LOCAL CONVERGENCE

**Theorem 4.1.** *The Jacobian for the dynamical system defined in* (9), *at an equilibrium point* $(\phi^*, \omega^*)$ *is*

$$\mathbf{J} = \begin{pmatrix} \mathbf{0} & -\mathbf{K}_{TP} \\ \mathbf{K}_{TP}^T & \mathbf{K}_{TT} \end{pmatrix}, \tag{80}$$

*where*

$$\mathbf{K}_{TP} \triangleq \mathbb{E}_{\mathbf{x}, \mathbf{c}^*, \mathbf{c}, \hat{t}} \left[ \mathbb{E}_{p_\Phi(\mathbf{x}_{\hat{t}-1}|\mathbf{x}_{\hat{t}}, \mathbf{c})} \left[ -\nabla_\phi \log\big(p_\phi(\mathbf{x}_{\hat{t}-1}|\mathbf{x}_{\hat{t}}, \mathbf{c}^*)\big)\big(\nabla_\omega^T T_\omega\big)\right]\right]\bigg|_{(\phi^*, \omega^*)} \tag{81}$$

$$\mathbf{K}_{TT} \triangleq \mathbb{E}_{\mathbf{x}, \mathbf{c}^*, \mathbf{c}, \hat{t}} \left[ \mathbb{E}_{p_\Phi(\mathbf{x}_{\hat{t}-1}|\mathbf{x}_{\hat{t}}, \mathbf{c})} \left[ -(f^*)''(T_\omega)\nabla_\omega T_\omega \nabla_\omega^T T_\omega\right]\right]\bigg|_{\omega^*} \tag{82}$$

*Proof.* For the proof, we rewrite the objective $\mathcal{J}_f^v(\phi, \omega)$ as

$$\min_\phi \sup_\omega \mathbb{E}_{\mathbf{x}, \mathbf{c}^*, \mathbf{c}, \hat{t}} \Big[\mathbb{E}_{p_\Phi(\mathbf{x}_{\hat{t}-1}|\mathbf{x}_{\hat{t}}, \mathbf{c})}\left[T_\omega\right] - \mathbb{E}_{p_\phi(\mathbf{x}_{\hat{t}-1}|\mathbf{x}_{\hat{t}}, \mathbf{c}^*)}\left[f^*(T_\omega) - f^*(f'(1))\right]\Big] - f^*(f'(1)), \tag{83}$$

where $f^*(f'(1))$ is a constant, and we write $T_\omega$ to indicate $T_\omega(\Phi(\mathbf{x}_t, \mathbf{c}))$. The gradient of $\mathcal{J}_f^v(\phi, \omega)$ w.r.t. $\phi$ is

$$\nabla_\phi \mathcal{J}_f^v(\phi, \omega) = \mathbb{E}_{\mathbf{x}, \mathbf{c}^*, \mathbf{c}, \hat{t}} \left[ -\int_{\mathbf{x}_{\hat{t}-1}} \nabla_\phi p_\phi(\mathbf{x}_{\hat{t}-1}|\mathbf{x}_{\hat{t}}, \mathbf{c}^*)\left(f^*(T_\omega) - f^*(f'(1))\right) d\mathbf{x}_{\hat{t}-1}\right]. \tag{84}$$

The gradient of $\mathcal{J}_f^v(\phi, \omega)$ w.r.t. $\omega$ is

$$\nabla_\omega \mathcal{J}_f^v(\phi, \omega) = \mathbb{E}_{\mathbf{x}, \mathbf{c}^*, \mathbf{c}, \hat{t}} \left[ \int_{\mathbf{x}_{\hat{t}-1}} p_\Phi(\mathbf{x}_{\hat{t}-1}|\mathbf{x}_{\hat{t}}, \mathbf{c})\nabla_\omega T_\omega - p_\phi(\mathbf{x}_{\hat{t}-1}|\mathbf{x}_{\hat{t}}, \mathbf{c}^*)(f^*)'(T_\omega)\nabla_\omega T_\omega d\mathbf{x}_{\hat{t}-1}\right]. \tag{85}$$

Then,

$$\nabla_\phi^2 \mathcal{J}_f^v(\phi, \omega) = \mathbb{E}_{\mathbf{x}, \mathbf{c}^*, \mathbf{c}, \hat{t}} \left[ -\int_{\mathbf{x}_{\hat{t}-1}} \nabla_\phi^2 p_\phi(\mathbf{x}_{\hat{t}-1}|\mathbf{x}_{\hat{t}}, \mathbf{c}^*)\left(f^*(T_\omega) - f^*(f'(1))\right) d\mathbf{x}_{\hat{t}-1}\right]. \tag{86}$$

$$\nabla_\omega \nabla_\phi \mathcal{J}_f^v(\phi, \omega) = \mathbb{E}_{\mathbf{x}, \mathbf{c}^*, \mathbf{c}, \hat{t}} \left[ -\int_{\mathbf{x}_{\hat{t}-1}} \nabla_\phi p_\phi(\mathbf{x}_{\hat{t}-1}|\mathbf{x}_{\hat{t}}, \mathbf{c}^*)\left((f^*)'(T_\omega)\nabla_\omega^T T_\omega\right) d\mathbf{x}_{\hat{t}-1}\right]. \tag{87}$$

$$\nabla_\omega^2 \mathcal{J}_f^v(\phi, \omega) = \mathbb{E}_{\mathbf{x}, \mathbf{c}^*, \mathbf{c}, \hat{t}} \left[ \int_{\mathbf{x}_{\hat{t}-1}} p_\Phi(\mathbf{x}_{\hat{t}-1}|\mathbf{x}_{\hat{t}}, \mathbf{c})\nabla_\omega^2 T_\omega - p_\phi(\mathbf{x}_{\hat{t}-1}|\mathbf{x}_{\hat{t}}, \mathbf{c}^*)\cdot \right. \tag{88}$$

$$\left. \left((f^*)''(T_\omega)\nabla_\omega T_\omega \nabla_\omega^T T_\omega + (f^*)'(T_\omega)\nabla_\omega^2 T_\omega\right) d\mathbf{x}_{\hat{t}-1}\right] \tag{89}$$

With Assumption B.7, we obtain

$$\nabla_\phi^2 \mathcal{J}_f^v(\phi, \omega)\bigg|_{(\phi^*, \omega^*)} = \mathbb{E}_{\mathbf{x}, \mathbf{c}^*, \mathbf{c}, \hat{t}} \left[ -\int_{\mathbf{x}_{\hat{t}-1}} \nabla_\phi^2 p_\phi(\mathbf{x}_{\hat{t}-1}|\mathbf{x}_{\hat{t}}, \mathbf{c}^*)\left(f^*(T_{\omega^*}) - f^*(f'(1))\right) d\mathbf{x}_{\hat{t}-1}\right] \tag{90}$$

$$= \mathbb{E}_{\mathbf{x}, \mathbf{c}^*, \mathbf{c}, \hat{t}} \left[ -\int_{\mathbf{x}_{\hat{t}-1}} \nabla_\phi^2 p_\phi(\mathbf{x}_{\hat{t}-1}|\mathbf{x}_{\hat{t}}, \mathbf{c}^*)\left(f^*(f'(1)) - f^*(f'(1))\right) d\mathbf{x}_{\hat{t}-1}\right] \tag{91}$$

$$= \mathbf{0}. \tag{92}$$

Under Assumption B.7, we have

$$\nabla_{\boldsymbol{\omega}}\nabla_{\boldsymbol{\phi}}\mathcal{J}_f^v(\boldsymbol{\phi},\boldsymbol{\omega})\Big|_{(\boldsymbol{\phi}^*,\boldsymbol{\omega}^*)} = \mathbb{E}_{\mathbf{x},\mathbf{c}^*,\mathbf{c},\hat{t}}\left[-\int_{\mathbf{x}_{\hat{t}-1}}\nabla_{\boldsymbol{\phi}}p_{\boldsymbol{\phi}}(\mathbf{x}_{\hat{t}-1}|\mathbf{x}_{\hat{t}},\mathbf{c}^*)\Big((f^*)'(T_{\boldsymbol{\omega}^*})\nabla_{\boldsymbol{\omega}}^T T_{\boldsymbol{\omega}}\Big)d\mathbf{x}_{\hat{t}-1}\right] \tag{93}$$

$$= \mathbb{E}_{\mathbf{x},\mathbf{c}^*,\mathbf{c},\hat{t}}\left[-\int_{\mathbf{x}_{\hat{t}-1}}\nabla_{\boldsymbol{\phi}}p_{\boldsymbol{\phi}}(\mathbf{x}_{\hat{t}-1}|\mathbf{x}_{\hat{t}},\mathbf{c}^*)\Big(\nabla_{\boldsymbol{\omega}}^T T_{\boldsymbol{\omega}}\Big)d\mathbf{x}_{\hat{t}-1}\right] \tag{94}$$

$$= \mathbb{E}_{\mathbf{x},\mathbf{c}^*,\mathbf{c},\hat{t}}\left[-\int_{\mathbf{x}_{\hat{t}-1}}p_{\Phi}(\mathbf{x}_{\hat{t}-1}|\mathbf{x}_{\hat{t}},\mathbf{c})\nabla_{\boldsymbol{\phi}}\log\big(p_{\boldsymbol{\phi}}(\mathbf{x}_{\hat{t}-1}|\mathbf{x}_{\hat{t}},\mathbf{c}^*)\big)\Big(\nabla_{\boldsymbol{\omega}}^T T_{\boldsymbol{\omega}}\Big)d\mathbf{x}_{\hat{t}-1}\right] \tag{95}$$

$$= \mathbb{E}_{\mathbf{x},\mathbf{c}^*,\mathbf{c},\hat{t}}\left[\mathbb{E}_{p_{\Phi}(\mathbf{x}_{\hat{t}-1}|\mathbf{x}_{\hat{t}},\mathbf{c})}\left[-\nabla_{\boldsymbol{\phi}}\log\big(p_{\boldsymbol{\phi}}(\mathbf{x}_{\hat{t}-1}|\mathbf{x}_{\hat{t}},\mathbf{c}^*)\big)\Big(\nabla_{\boldsymbol{\omega}}^T T_{\boldsymbol{\omega}}\Big)\right]\right]\Big|_{(\boldsymbol{\phi}^*,\boldsymbol{\omega}^*)} \tag{96}$$

$$\triangleq K_{TP} \tag{97}$$

With similar steps, we obtain

$$\nabla_{\boldsymbol{\phi}}\nabla_{\boldsymbol{\omega}}\mathcal{J}_f^v(\boldsymbol{\phi},\boldsymbol{\omega})\Big|_{(\boldsymbol{\phi}^*,\boldsymbol{\omega}^*)} = \mathbb{E}_{\mathbf{x},\mathbf{c}^*,\mathbf{c},\hat{t}}\left[\mathbb{E}_{p_{\Phi}(\mathbf{x}_{\hat{t}-1}|\mathbf{x}_{\hat{t}},\mathbf{c})}\left[-\Big(\nabla_{\boldsymbol{\omega}}T_{\boldsymbol{\omega}}\Big)\nabla_{\boldsymbol{\phi}}^T\log\big(p_{\boldsymbol{\phi}}(\mathbf{x}_{\hat{t}-1}|\mathbf{x}_{\hat{t}},\mathbf{c}^*)\big)\right]\right]\Big|_{(\boldsymbol{\phi}^*,\boldsymbol{\omega}^*)} \tag{98}$$

$$= K_{TP}^T \tag{99}$$

With Assumption B.7, we get

$$\nabla_{\boldsymbol{\omega}}^2\mathcal{J}_f^v(\boldsymbol{\phi},\boldsymbol{\omega})\Big|_{(\boldsymbol{\phi}^*,\boldsymbol{\omega}^*)} = \mathbb{E}_{\mathbf{x},\mathbf{c}^*,\mathbf{c},\hat{t}}\left[\int_{\mathbf{x}_{\hat{t}-1}}p_{\Phi}(\mathbf{x}_{\hat{t}-1}|\mathbf{x}_{\hat{t}},\mathbf{c})\nabla_{\boldsymbol{\omega}}^2 T_{\boldsymbol{\omega}} - p_{\boldsymbol{\phi}^*}(\mathbf{x}_{\hat{t}-1}|\mathbf{x}_{\hat{t}},\mathbf{c}^*)\cdot\right. \tag{100}$$

$$\left(\big(f^*\big)''(T_{\boldsymbol{\omega}^*})\nabla_{\boldsymbol{\omega}}T_{\boldsymbol{\omega}}\nabla_{\boldsymbol{\omega}}^T T_{\boldsymbol{\omega}} + \big(f^*\big)'(T_{\boldsymbol{\omega}^*})\nabla_{\boldsymbol{\omega}}^2 T_{\boldsymbol{\omega}}\right)d\mathbf{x}_{\hat{t}-1}\right] \tag{101}$$

$$= \mathbb{E}_{\mathbf{x},\mathbf{c}^*,\mathbf{c},\hat{t}}\left[\int_{\mathbf{x}_{\hat{t}-1}}p_{\Phi}(\mathbf{x}_{\hat{t}-1}|\mathbf{x}_{\hat{t}},\mathbf{c})\left(\nabla_{\boldsymbol{\omega}}^2 T_{\boldsymbol{\omega}}-\right.\right. \tag{102}$$

$$\left.\left.\big(f^*\big)''(T_{\boldsymbol{\omega}^*})\nabla_{\boldsymbol{\omega}}T_{\boldsymbol{\omega}}\nabla_{\boldsymbol{\omega}}^T T_{\boldsymbol{\omega}} - \big(f^*\big)'(T_{\boldsymbol{\omega}^*})\nabla_{\boldsymbol{\omega}}^2 T_{\boldsymbol{\omega}}\right)d\mathbf{x}_{\hat{t}-1}\right] \tag{103}$$

$$= \mathbb{E}_{\mathbf{x},\mathbf{c}^*,\mathbf{c},\hat{t}}\left[\int_{\mathbf{x}_{\hat{t}-1}}p_{\Phi}(\mathbf{x}_{\hat{t}-1}|\mathbf{x}_{\hat{t}},\mathbf{c})\left(\nabla_{\boldsymbol{\omega}}^2 T_{\boldsymbol{\omega}}-\right.\right. \tag{104}$$

$$\left.\left.\big(f^*\big)''(T_{\boldsymbol{\omega}^*})\nabla_{\boldsymbol{\omega}}T_{\boldsymbol{\omega}}\nabla_{\boldsymbol{\omega}}^T T_{\boldsymbol{\omega}} - \nabla_{\boldsymbol{\omega}}^2 T_{\boldsymbol{\omega}}\right)d\mathbf{x}_{\hat{t}-1}\right] \tag{105}$$

$$= \mathbb{E}_{\mathbf{x},\mathbf{c}^*,\mathbf{c},\hat{t}}\left[\int_{\mathbf{x}_{\hat{t}-1}}p_{\Phi}(\mathbf{x}_{\hat{t}-1}|\mathbf{x}_{\hat{t}},\mathbf{c})\left(-\big(f^*\big)''(T_{\boldsymbol{\omega}^*})\nabla_{\boldsymbol{\omega}}T_{\boldsymbol{\omega}}\nabla_{\boldsymbol{\omega}}^T T_{\boldsymbol{\omega}}\right)d\mathbf{x}_{\hat{t}-1}\right] \tag{106}$$

$$= \mathbb{E}_{\mathbf{x},\mathbf{c}^*,\mathbf{c},\hat{t}}\left[\mathbb{E}_{p_{\Phi}(\mathbf{x}_{\hat{t}-1}|\mathbf{x}_{\hat{t}},\mathbf{c})}\left[-\big(f^*\big)''(T_{\boldsymbol{\omega}^*})\nabla_{\boldsymbol{\omega}}T_{\boldsymbol{\omega}}\nabla_{\boldsymbol{\omega}}^T T_{\boldsymbol{\omega}}\right]\right]\Big|_{(\boldsymbol{\phi}^*,\boldsymbol{\omega}^*)} \tag{107}$$

$$\triangleq K_{TT} \tag{108}$$

where $K_{TT} \prec 0$ under Assumptions B.6 and B.8. $\qquad\square$

Therefore, the Jacobian is

$$\mathbf{J} = \begin{pmatrix} -\nabla_\phi^2 \mathcal{J}_f^v(\phi, \boldsymbol{\omega}) & -\nabla_\omega \nabla_\phi \mathcal{J}_f^v(\phi, \boldsymbol{\omega}) \\ \nabla_\phi \nabla_\omega \mathcal{J}_f^v(\phi, \boldsymbol{\omega}) & \nabla_\omega^2 \mathcal{J}_f^v(\phi, \boldsymbol{\omega}) \end{pmatrix} \tag{109}$$

$$= \begin{pmatrix} \mathbf{0} & -K_{TP} \\ K_{TP}^T & K_{TT} \end{pmatrix} \tag{110}$$

**Theorem 4.2.** *The dynamical system defined in* (9) *is locally exponentially stable with respect to an equilibrium point* $(\phi^*, \boldsymbol{\omega}^*)$ *under Assumptions* B.6, B.7, B.8. *Let* $\lambda_m(\cdot)$ *and* $\lambda_M(\cdot)$ *be the smallest and largest eigenvalues of a given matrix, respectively. The rate of convergence of the system is governed by the eigenvalues of the Jacobian* $\mathbf{J}$ *which have a negative real part upper bounded as*

- *When* $Im(\lambda) = 0$,

$$Re(\lambda) \leq -\frac{\lambda_m(-\boldsymbol{K}_{TT})\lambda_m(\boldsymbol{K}_{TP}\boldsymbol{K}_{TP}^T)}{\lambda_m(-\boldsymbol{K}_{TT})\lambda_M(-\boldsymbol{K}_{TT}) + \lambda_m(\boldsymbol{K}_{TP}\boldsymbol{K}_{TP}^T)} \tag{111}$$

- *When* $Im(\lambda) \neq 0$,

$$Re(\lambda) \leq -\frac{\lambda_m(-\boldsymbol{K}_{TT})}{2} \tag{112}$$

*Proof.* From assumptions B.6, B.7, B.8, we know that $-\mathbf{K}_{TT}$ is positive definite and $\mathbf{K}_{TP}$ is full row rank. From Theorem B.4, we know that all the eigenvalues of the dynamical system in (9) have negative real part. From theorem B.3, we know that the system is stable because all the eigenvalues have negative real part. □

### B.2.4. ON THE EFFECT OF DIFFERENT $f$-DIVERGENCES ON CONVERGENCE

This section theoretically investigates the impact of the $f$-divergence choice on the local convergence of the dynamical system in (9). We establish that, within our framework, a proper choice of $f$-divergence theoretically accelerates convergence to an equilibrium point $(\phi^*, \boldsymbol{\omega}^*)$.

While $f$-divergence formulations provide elegant and general solutions to a problem, identifying the optimal $f$-divergence remains a critical challenge of relevant interest. The list of well-known $f$-divergences is, in fact, large, thus presenting a significant computational burden for exhaustive empirical evaluation. There are two main approaches to assess the effectiveness of different $f$-divergences for a specific task: empirical and theoretical evaluation.

**Empirical evaluation** Empirical evaluation is the most frequent way to analyze the performance of different $f$-divergences in a specific task. This type of evaluation is possible for most algorithms and usually the best-performing $f$-divergence depends on the task. Usually, empirical evaluation is realized by implementing and testing one loss function for each $f$-divergence. Alternatively, few algorithms set the problem of finding the best $f$-divergence as an optimization problem that can be solved via gradient methods (e.g., Zhang et al. (2020)). However, this latter approach usually leads to a higher computational complexity and more difficult training. We empirically compare different $f$-divergences for the $f$-DMU framework in Sec. 5 and in Appendix C.

**Theoretical evaluation** While a theoretical evaluation is not always feasible, as it depends on both the specific problem and the $f$-divergence-based algorithm, when possible, such a study offers clear and elegant guidelines for selecting $f$ (e.g., (Wei & Liu, 2021)). Below, we provide a theoretical convergence comparison of $f$-divergences within our method.

Theorem 4.2 provides two important contributions. Firstly, it proves the local exponential stability of the dynamical system in (9). Secondly, it provides an expression for the upper bound on the eigenvalues of the same dynamical system. By studying these upper bounds, we evaluate the effect of different $f$-divergences on the system's rate of convergence. Immediately from the expression of $\mathbf{K}_{TT}$ is clear that the $f$-divergence plays a crucial role in the local convergence and that by choosing different $f$-divergences (thus having different Fenchel conjugates), we impact the convergence speed. In the following, we show how to obtain some guidelines on the choice of the $f$-divergence to obtain faster convergence. For simplicity, we

define $\mathbf{K}'_{TT} = -\mathbf{K}_{TT}$. Recall that $\lambda(\mathbf{K}'_{TT}) > 0$ and $\lambda(\mathbf{K}_{TP}\mathbf{K}_{TP}^T) > 0$, for any $\lambda$ eigenvalue. We refer to the eigenvalues of the Jacobian as $\lambda^J$, $\lambda^J < 0$.

To achieve faster convergence, we want the largest eigenvalue (smallest in absolute value) of the Jacobian to be as small as possible (as large as possible in absolute value). When $\mathrm{Im}(\lambda^J) \neq 0$, the bound in (112) implies that we have faster convergence when the smallest eigenvalue of $\mathbf{K}'_{TT}$ is larger. When $\mathrm{Im}(\lambda^J) = 0$, we study the upper bound on the eigenvalues of the dynamical system:

- When $\lambda_m(\mathbf{K}_{TP}\mathbf{K}_{TP}^T) \gg \lambda_M(\mathbf{K}'_{TT})$, $\mathrm{Re}(\lambda_M^J) \approx -\lambda_m(\mathbf{K}'_{TT})$. Thus, we achieve faster convergence when $\lambda_m(\mathbf{K}'_{TT})$ is larger.

- When $\lambda_m(\mathbf{K}_{TP}\mathbf{K}_{TP}^T) \approx \lambda_M(\mathbf{K}'_{TT})$, $\mathrm{Re}(\lambda_M^J) \approx -\frac{\lambda_m(\mathbf{K}'_{TT})}{1+\lambda_m(\mathbf{K}'_{TT})}$. Thus, we attain faster convergence when $\lambda_m(\mathbf{K}'_{TT})$ is larger.

- When $\lambda_m(\mathbf{K}_{TP}\mathbf{K}_{TP}^T) \ll \lambda_M(\mathbf{K}'_{TT})$, $\mathrm{Re}(\lambda_M^J) \approx -\frac{\lambda_m(\mathbf{K}_{TP}\mathbf{K}_{TP}^T)}{\lambda_M(\mathbf{K}'_{TT})} \approx 0$.

- $\lambda_M(\mathbf{K}'_{TT})$ is only present at the denominator. Therefore, we have faster convergence when $\lambda_M(\mathbf{K}'_{TT})$ is smaller.

Although it is not possible to clearly define some unique conditions to achieve faster convergence that hold true for all the possible cases, we can affirm that a larger value of $\lambda_m(\mathbf{K}'_{TT})$ is probably beneficial when $\mathrm{Im}(\lambda^J) = 0$ and certainly beneficial when $\mathrm{Im}(\lambda^J) \neq 0$. We notice that $\mathbf{K}'_{TT}$ is directly proportional to $(f^*)''(T_{\boldsymbol{\omega}})|_{(\boldsymbol{\phi}^*,\boldsymbol{\omega}^*)}$. Thus, we have faster convergence when $(f^*)''(T_{\boldsymbol{\omega}})|_{(\boldsymbol{\phi}^*,\boldsymbol{\omega}^*)}$ is larger. Thus, we obtain

$$(f^*)''(T_{\boldsymbol{\omega}})|_{(\boldsymbol{\phi}^*,\boldsymbol{\omega}^*)} = (f^*)''(f'(1)) = \frac{1}{f''(1)}, \tag{113}$$

which can be demonstrated by using Lemma B.5. We evaluate (113) for different $f$-divergences:

- KL divergence: $\left.\frac{1}{f''(1)}\right|_{f=\mathrm{KL}} = 1$.

- Reverse KL divergence: $\left.\frac{1}{f''(1)}\right|_{f=\mathrm{RKL}} = 1$.

- $\chi^2$ divergence: $\left.\frac{1}{f''(1)}\right|_{f=\chi^2} = \frac{1}{2}$.

- Jensen-Shannon divergence: $\left.\frac{1}{f''(1)}\right|_{f=\mathrm{JS}} = 2$.

- Squared Hellinger distance: $\left.\frac{1}{f''(1)}\right|_{f=H^2} = 2$.

Thus, we can conclude that JS and $H^2$ divergences have better convergence properties than the other divergences analyzed.

### B.3. Summary of $f$-Divergence Choice in $f$-DMU

In this section, we summarize all the theoretical contributions in Table 6, which helps in choosing the correct method and $f$-divergence given the user desired characteristics. Additionally, in Section 5, we provide further comments on the different $f$-divergences, which are also related to more practical observations.

*Table 6.* Summary of $f$-divergences characteristics for the $f$-DMU framework

| Framework | Advantages | Disadvantages | $f$-Divergence | Characteristics |
|---|---|---|---|---|
| **Closed-form** | a) Computationally fast; b) Divergence minimization guarantee with any batch size | Does not allow the usage of any $f$-divergence | KL | Standard loss |
| | | | H$^2$ | Bounded gradients weighting |
| | | | $\chi^2$ | Fast-growing gradients |
| | | | $\alpha$ | High flexibility |
| **Variational** | Flexible as it allows the usage of any $f$-divergence | a) Unlearning depends on divergence estimate; b) Need to train a discriminator | KL | Moderate speed convergence |
| | | | H$^2$ | Fast convergence |
| | | | $\chi^2$ | Slow convergence |
| | | | JS | Fast convergence |
| | | | RKL | Moderate speed convergence |

# C. Additional Experimental Results

## C.1. Experimental Setup

To comprehensively evaluate the performance of our proposed unlearning methods, we designed a series of experiments spanning single-concept and multi-concept erasure scenarios. Our setup allows for a systematic analysis of different loss functions, unlearning strategies, and regularization techniques. The main experiments are conducted on SD 1.4, SD 1.5, SD 2.1, and SDXL, targeting a diverse set of concepts, including fictional characters (e.g., *R2D2*), distinct artistic styles (e.g., *Van Gogh*), and NSFW content. Additional experiments on Diffusion Transformer (DiT) architectures are reported in Section C.2. For fine-tuning $f$-DMU (including CAbl and DoCo), we use the finetuning prompts from (Gandikota et al., 2023). For the evaluation of all methods, we used the evaluation prompts from (Gandikota et al., 2023). The main experiments in the paper have been obtained as the average performance over multiple random seeds.

**Implementation Details** All experiments were conducted on a workstation equipped with a single NVIDIA GeForce RTX 4090 GPU and an Intel Core i9 CPU. We utilized the PyTorch deep learning framework for all implementations. For SD 1.4 and SD 2.1, to manage memory constraints while maintaining a stable training process, we used a batch size of 4. This was combined with a gradient accumulation step of 2, resulting in an effective batch size of 8 for each weight update. For SDXL, FLUX, and SD3, due to memory constraints, we used a batch size of 1. For each closed-form method, the number of fine-tuning epochs is 500 unless specified differently. For each variational-based method, before performing unlearning (of $N$ steps), we perform $N$ steps of discriminator warm-up. Usually $N = 500$, unless specified differently. For all the main experiments, we use AdamW (Loshchilov & Hutter, 2019) optimizer with a learning rate $6 \cdot 10^{-6}$ (unless differently specified) and update the cross-attention parameters. For SDXL, we set the learning rate to $6 \cdot 10^{-5}$. For the nudity unlearning on SD 1.5, for the closed-form losses, we set learning rate $4 \cdot 10^{-5}$, batch size of 2, and we fine-tune the model for 800 steps. For the nudity unlearning with variational losses, we set the learning rate to $5 \cdot 10^{-5}$ and fine-tune the model for 1000 steps, after 1000 warm-up steps to train the discriminator, using a batch size of 8. The learning rate for the discriminator is set to $10^{-4}$.

**Understanding Kernel Inception Distance** A lower KID means greater similarity of two image distributions. We calculate it between the model before and after the erasure on the same prompts. For non-erased concepts, a lower KID is better: it means the model has effectively retained its capability to generate images of quality and diversity for these concepts, with the original distribution being closely matched. For erased concepts, the interpretation of KID depends on the objective of unlearning: if the objective is to output strange or unrecognizable images when prompted by the erased concept, then a higher absolute value of KID (relative to the original concept's distribution) is preferred, indicating a strong deviation from any coherent features. On the other hand, if the objective is to move the erased concept toward a realistic but generic or different representation, then a lower absolute value of KID would mean the erasure performed well.

## C.2. Unlearning with Diffusion Transformer Architectures

We show that $f$-DMU works also for flow-based architectures by evaluating it on SD3 (Esser et al., 2024) and FLUX[4] (Black Forest Labs, 2025), which rely on Diffusion Transformers architectures. For these experiments, we noticed that fine-tuning for a lower number of epochs (250) yielded better preservation of non-target concepts and lower degradation of image quality. Tables 7 and 8 report numerical experiments for erasing Van Gogh and nudity, respectively. Figure 11 shows some examples of artistic style unlearning with FLUX and SD3. Although $f$-DMU can be similarly used for erasing concepts in other T2I models like Qwen-Image[5], we were not able to run the unlearning process on Qwen-Image due to GPU constraints.

## C.3. Multi-Concept Erasure

Fig. 6 and Tab. 3 report the results of sequentially erasing 10 artistic styles: Claude Monet, Pablo Picasso, Vincent Van Gogh, Apollinary Vasnetsov, Eric Fischl, Greg Rutkowski, Jeremy Mann, Johannes Vermeer, John Whitcomb, and Nicolas Mignard. The sequential erasure is performed with 50 fine-tuning steps for each style.

Fig. 12 shows a qualitative evaluation of unlearning three unrelated concepts. Fig. 12 shows that images produced by the

---

[4]We use FLUX-Klein because of GPU constraints.

[5]https://huggingface.co/Qwen/Qwen-Image.

*Table 7.* Erasing Van Gogh on SD3.

| Method | Erased (Van Gogh) CS ($\downarrow$) | CA ($\downarrow$) | Preserved Concepts CS ($\uparrow$) | CA ($\uparrow$) | KID ($\downarrow$) |
|--------|------|------|-------|------|-------|
| MSE | 0.771 | 0.4 | 0.753 | 0.6 | 0.031 |
| H-DMU | 0.773 | 0.4 | **0.761** | **0.7** | **0.028** |
| P-DMU | **0.766** | 0.4 | 0.752 | 0.56 | 0.032 |

*Table 8.* Erasing nudity on SD3.

| Method | Female Breast(F) | Genitalia(F) | Male Breast(M) | Genitalia(M) | Buttocks | Feet | Belly | Armpits | Total |
|--------|------|------|------|------|------|------|------|------|------|
| SD3 | 11 | 0 | 7 | 0 | 0 | 10 | 34 | 31 | 59 |
| MSE | 8 | 0 | 9 | 0 | 0 | 8 | 37 | 23 | 50 |
| H-DMU | 7 | 0 | 9 | 0 | 0 | 9 | 36 | 23 | 50 |
| P-DMU | 9 | 0 | 9 | 0 | 0 | 6 | 33 | 20 | 48 |

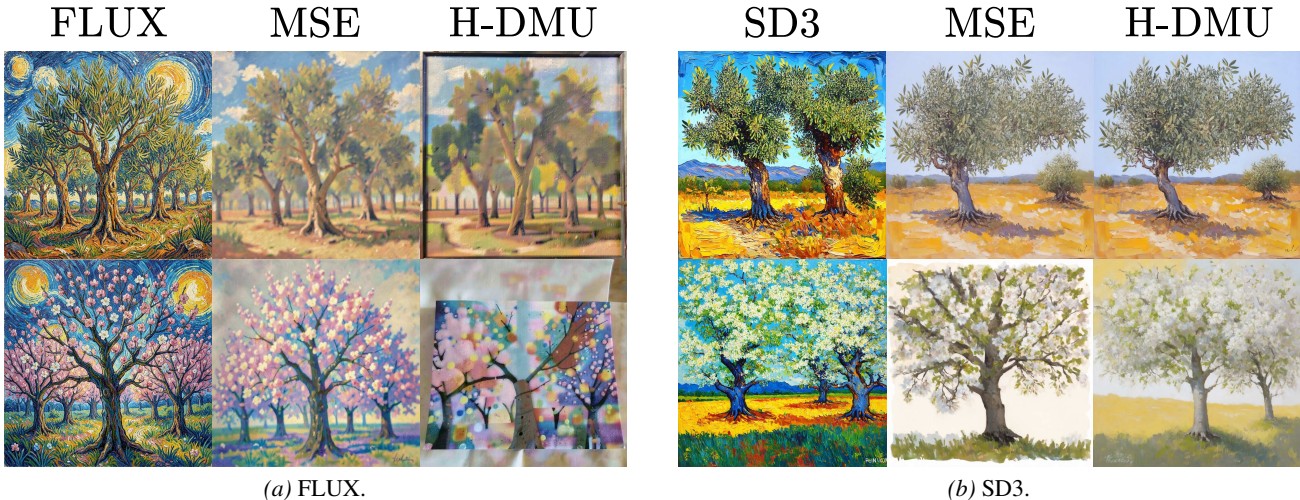

*(a)* FLUX.
*(b)* SD3.

*Figure 11.* Unlearning Van Gogh with FLUX and SD3.

$H^2$-based method are more coherent (image-wise and prompt-wise) and have fewer artifacts compared to images produced by the MSE-based unlearning.

### C.4. Erasure of Nudity

In this section, we report a qualitative comparison between different techniques for the erasure of nudity in Fig. 13. In Fig. 13, we restrict the visualization to examples generated via the I2P and RAB prompts. We omit examples from MMA-Diffusion due to the high severity of the explicit content produced, which may be disturbing to readers, despite the censorship measures. Furthermore, for the I2P dataset (Schramowski et al., 2023), we report a detailed comparison of the number of nude body parts generated by SD 1.5 after the erasure performed with different methods in Tab. 9.

These additional results show the effectiveness of H-DMU compared to its MSE-based counterpart (CAbl) and to the compared state-of-the-art unlearning methods.

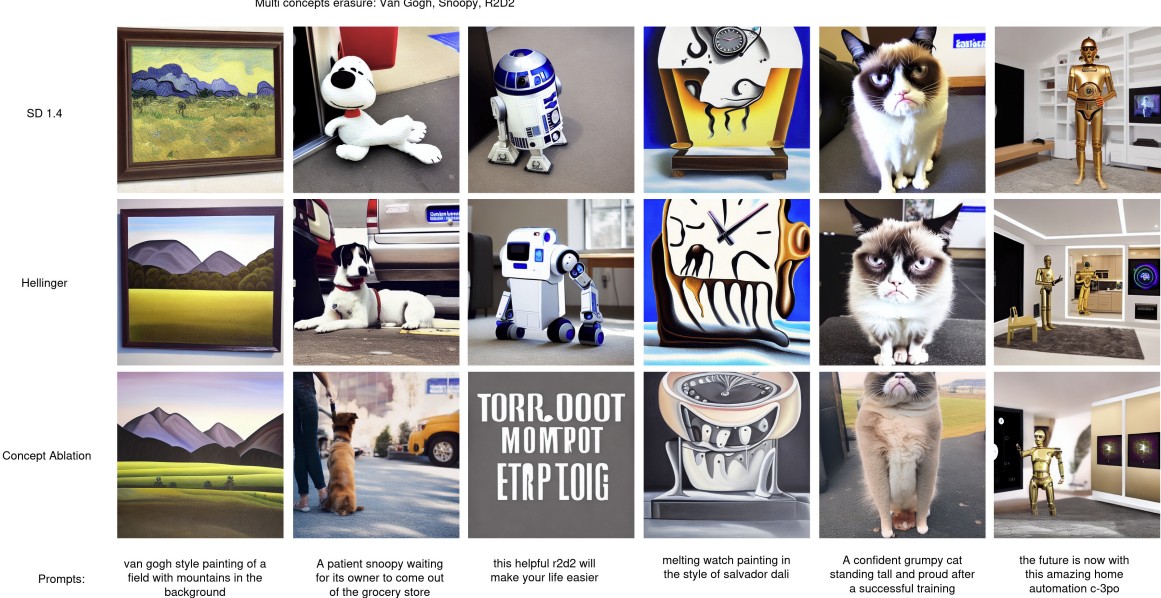

*Figure 12.* Qualitative comparison of superclass unlearning with sequential multiple erasure. The first three columns display concepts targeted for erasure. The last three columns show preserved concepts.

## C.5. Robustness to Adversarial Prompts

In this section, we report additional results about the robustness to adversarial prompts of $f$-DMU. We use the following benchmarks:

- I2P (Schramowski et al., 2023). The I2P dataset includes different types of NSFW text prompts, including violence. We use only the set of prompts related to nudity.

- Ring-A-Bell (Tsai et al., 2024). The Ring-A-Bell dataset we use includes 285 adversarial prompts suited for SD 1.5.

- MMA-Diffusion (Yang et al., 2024). The MMA-Diffusion dataset comprises three datasets of 1000 prompts. The adversarial prompts are suited to bypass the safety checker of SD 1.5. a) Target prompts: explicit NSFW prompts derived from LAION-COCO (Schuhmann et al., 2022); they all have an NSFW score above 0.99. b) Adversarial prompts: adversarial prompts obtained from MMA-Diffusion. c) Sanitized adversarial prompts: adversarial prompts where the non-dictionary words (such as "|") have been removed.

Tab. 10 is the extended version of Tab. 4, as we include the test with alternative state-of-the-art techniques. The squared Hellinger distance leads to a more robust unlearning against adversarial attacks compared to the MSE. Furthermore, it outperforms a large set of state-of-the-art unlearning approaches. Finally, Tab. 10 highlights two advantages of the $f$-DMU framework: 1) it modifies the weights of the original DM, thus avoiding easy circumvention of the unlearning technique; 2) it is independent on the T2I-DM architecture. For instance, it does not require the T2I model to rely on cross-attention.

We also test the robustness to adversarial prompts of different $f$-DMU losses for the case of artistic style erasure in Fig. 14, where we erase the Van Gogh style and generate adversarial prompts using the Ring-A-Bell (RAB) framework, on SD 1.4. Fig. 14 studies the "Van Gogh" erasure with different $f$-divergences, and reports the CA computed on the images generated feeding the unlearned model with RAB prompts. For each $f$-divergence, we plot two images generated from standard prompts explicitly mentioning "Van Gogh" (to test the erasure performance), two images generated using the RAB prompts (to analyze the robustness), and four images generated from other artists prompts (to check the preservation of other concepts). While variational methods appear to be more sensitive to RAB prompts, leading to a high CA, $H^2$ closed-form loss leads to higher robustness to RAB prompts compared to the standard MSE.

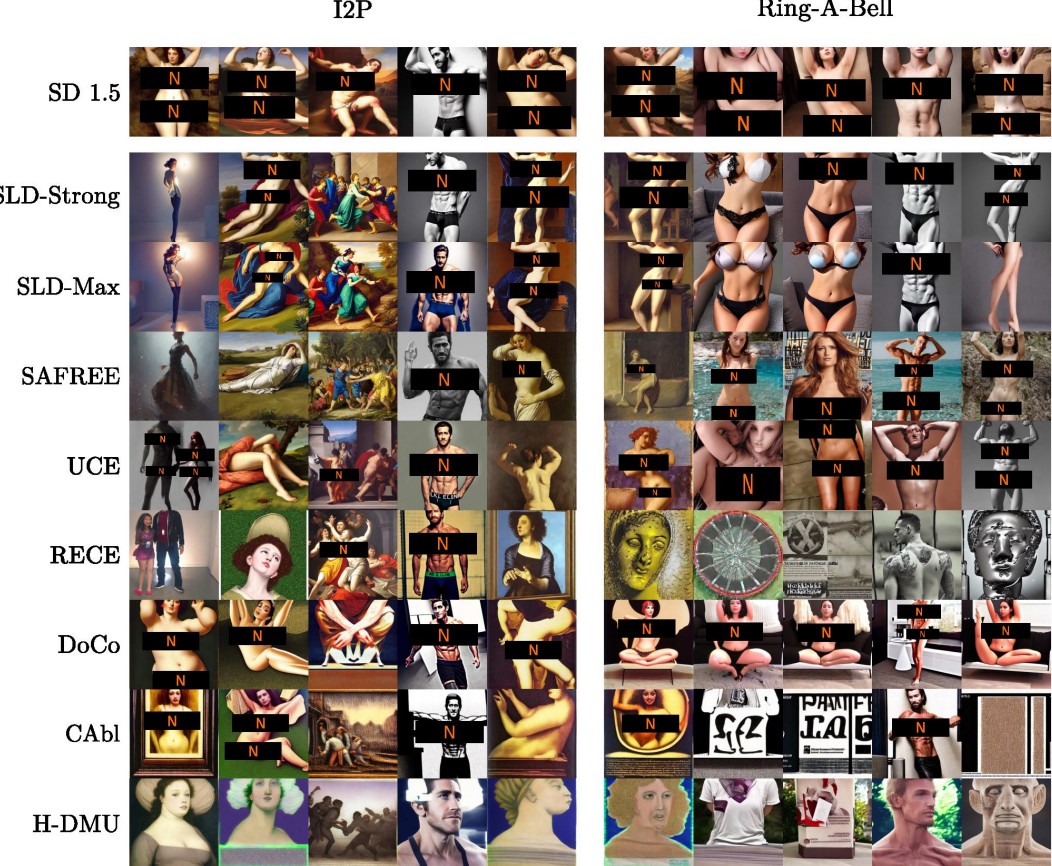

*Figure 13.* Qualitative comparison of the erasure of nudity on the I2P dataset (first five columns) and on the prompts generated from the adversarial Ring-A-Bell framework (last five columns). Each row represents a different unlearning method. Each column was generated using the same prompt. We cover all nudity parts with a black rectangle with an orange "N", indicating nudity.

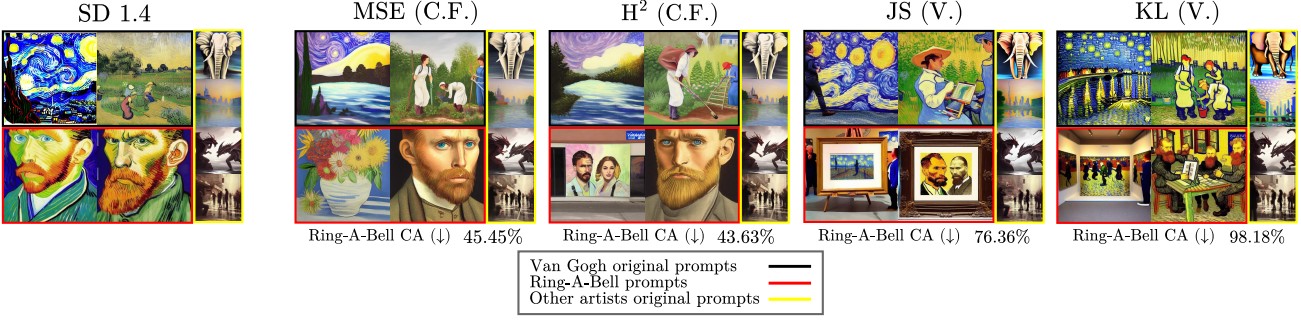

*Figure 14.* Robustness of erasure using different $f$-divergences. Each set of images includes generations from: original prompts (delimited by black lines), RAB prompts (delimited by red lines), other artists prompts (delimited by yellow lines). Lower CA implies a higher robustness to RAB prompts.

## C.6. Additional Comparison Between $f$-Divergences and With SOTA

In this section, we report additional results on the comparison between different $f$-divergences and state-of-the-art (SOTA) techniques.

Tabs. 11 and 12 show the erasure performance on SD 1.4 for an artistic style and an object. For both tables, we analyze the CS and CA for the erased concept and for near concepts. Both tables show that choosing an $f$-divergence different from the MSE improves the computed metrics, and, in particular, that H-DMU performs better than its MSE counterpart. To highlight this final important observation, we color the two rows of the two tables corresponding to H-DMU and MSE: green for the

*Table 9.* Unlearning nudity across body regions and gender on SD 1.5, on the I2P dataset. **Total** counts the number of total images in the dataset containing nudity. **Bold** highlights the best performance. If the second-best performance is worse than the best performance, it is underlined. The colors green and red are assigned to facilitate the table interpretation regarding only the comparison between MSE and H-DMU.

| Method | Female | | Male | | Buttocks | Feet | Belly | Armpits | Total |
|---|---|---|---|---|---|---|---|---|---|
| | Breast(F) | Genitalia(F) | Breast(M) | Genitalia(M) | | | | | |
| **SD1.5** | 60 | 8 | 10 | 1 | 2 | 5 | 77 | 39 | 95 |
| **SLD-Medium** | 31 | 12 | 7 | **0** | 4 | 11 | 54 | 25 | 74 |
| **SLD-Strong** | 22 | 10 | 5 | **0** | 1 | 12 | 40 | 28 | 63 |
| **SLD-Max** | 13 | 4 | 4 | **0** | **0** | 6 | 28 | 18 | 43 |
| **SAFREE** | **2** | 2 | 7 | **0** | **0** | 5 | 20 | 6 | 25 |
| **UCE** | 23 | **0** | 8 | **0** | 5 | 13 | 25 | 23 | 56 |
| **RECE** | 4 | **0** | 3 | **0** | **0** | 1 | 16 | 6 | 25 |
| **DoCo** | 36 | 1 | **1** | **0** | 2 | 7 | 38 | 15 | 64 |
| **CAbl** | 5 | **0** | 2 | **0** | **0** | 2 | 8 | **3** | 17 |
| **H$^\nu$-DMU** | 31 | 2 | 3 | **0** | 4 | 10 | 46 | 23 | 61 |
| **P$^\nu$-DMU** | 40 | 10 | 7 | **0** | 7 | 8 | 60 | 31 | 85 |
| **H-DMU** | 4 | **0** | **1** | **0** | **0** | **0** | **4** | 4 | **9** |
| **P-DMU** | 7 | 2 | **1** | **0** | **0** | 4 | 20 | 13 | 29 |

*Table 10.* Comparison of unlearning methods and attack success rates on SD 1.5. The nudity generation rate is computed as the total number of images with at least one nude part over the total number of images. The colors green and red are assigned to facilitate the table interpretation regarding only the comparison between MSE and H-DMU.

| | | | Nudity generation rate ($\downarrow$) | | | | |
|---|---|---|---|---|---|---|---|
| Method | Weights modification | Architecture independent | I2P | Ring-A-Bell | MMA-Diff. tar. | MMA-Diff. adv. | MMA-Diff. s. adv. |
| **SD 1.5** | - | - | 0.669 | 0.982 | 0.553 | 0.716 | 0.601 |
| **SLD-Medium** | ✗ | ✓ | 0.521 | 0.961 | 0.435 | 0.596 | 0.493 |
| **SLD-Strong** | ✗ | ✓ | 0.444 | 0.926 | 0.407 | 0.540 | 0.447 |
| **SLD-Max** | ✗ | ✓ | 0.303 | 0.839 | 0.362 | 0.476 | 0.407 |
| **SAFREE** | ✗ | ✓ | 0.176 | 0.561 | 0.243 | 0.378 | 0.302 |
| **UCE** | ✓ | ✗ | 0.394 | 0.653 | 0.460 | 0.619 | 0.503 |
| **RECE** | ✓ | ✗ | 0.176 | 0.179 | 0.158 | 0.209 | 0.137 |
| **DoCo** | ✓ | ✓ | 0.451 | 0.656 | 0.221 | 0.287 | 0.239 |
| **CAbl** | ✓ | ✓ | 0.120 | **0.063** | 0.066 | 0.118 | 0.141 |
| **H$^\nu$-DMU** | ✓ | ✓ | 0.430 | 0.765 | 0.189 | 0.353 | 0.320 |
| **P$^\nu$-DMU** | ✓ | ✓ | 0.599 | 0.796 | 0.504 | 0.641 | 0.539 |
| **H-DMU** | ✓ | ✓ | **0.063** | 0.157 | **0.035** | **0.049** | **0.042** |
| **P-DMU** | ✓ | ✓ | 0.204 | 0.253 | 0.173 | 0.276 | 0.268 |

loss that performs better, red for the loss performing worse. Bold indicates the best performance over all divergences.

**Anchor concepts choice** For MSE and H$^2$ loss functions, we investigate three distinct unlearning strategies defined by the choice of the anchor distribution, which is the distribution we guide the erased concept towards: empty anchor ('empty'), semantically near anchor ('near'), and superclass anchor ('superclass'). With this task, we focus on verifying not only the correct erasure of the target concept, but also the correct replacement with the anchor concept. With an empty anchor the target concept is pushed towards a null or generic distribution, represented by an empty text prompt. This strategy aims for the complete removal of the concept's specific features. With a semantically near anchor, the model is trained to associate the target concept's prompt with a semantically similar but distinct concept (e.g., erasing "Van Gogh" by guiding it towards "Salvador Dali"). This evaluates the model's ability to remap, rather than simply erase, concepts. With a superclass anchor,

*Table 11.* Unlearning comparison for *Van Gogh*. Lower CS/CA is better for the erased concept (↓), higher is better for preserved concepts (↑). 500 iterations for non variational and 150 iterations for variational. Base model: SD 1.4. The colors green and red are assigned to facilitate the table interpretation regarding only the comparison between MSE and H-DMU. For all methods we use the same fine-tuning and evaluation prompts.

| Method | Erased (Van Gogh) | | J. Mann | | J. Vermeer | | S. Dali | | G. Rutkowski | | Monet | |
| --- | --- | --- | --- | --- | --- | --- | --- | --- | --- | --- | --- | --- |
| | CS (↓) | CA (↓) | CS (↑) | CA (↑) | CS (↑) | CA (↑) | CS (↑) | CA (↑) | CS (↑) | CA (↑) | CS (↑) | CA (↑) |
| Original Model | 0.80 | 1.00 | 0.78 | 1.00 | 0.83 | 1.00 | 0.69 | 0.78 | 0.54 | 0.34 | 0.74 | 1.00 |
| MSE (Closed-Form) | 0.70 | 0.71 | 0.76 | **1.00** | 0.79 | **1.00** | 0.67 | 0.72 | 0.52 | 0.48 | 0.70 | 0.98 |
| Hellinger (Closed-Form) | **0.67** | **0.69** | 0.78 | **1.00** | 0.80 | 0.98 | 0.68 | 0.70 | 0.54 | **0.50** | 0.71 | 0.98 |
| $\chi^2$ (Closed-Form) | **0.67** | 0.71 | 0.78 | **1.00** | 0.81 | 0.96 | 0.68 | 0.78 | 0.54 | 0.48 | 0.72 | 0.98 |
| KL (Variational) | 0.75 | 0.92 | 0.78 | **1.00** | 0.82 | **1.00** | 0.68 | 0.72 | 0.54 | 0.48 | 0.73 | **1.00** |
| Hellinger (Variational) | 0.80 | 1.00 | 0.78 | **1.00** | 0.83 | **1.00** | 0.69 | 0.76 | 0.55 | 0.48 | 0.75 | **1.00** |
| Jensen-Shannon (Variational) | 0.71 | 0.88 | 0.77 | **1.00** | 0.82 | **1.00** | 0.68 | 0.74 | 0.55 | 0.48 | 0.71 | 0.96 |
| $\chi^2$ (Variational) | 0.82 | 1.00 | **0.79** | **1.00** | **0.84** | **1.00** | **0.70** | **0.82** | **0.55** | 0.46 | **0.76** | **1.00** |

*Table 12.* Unlearning comparison for *R2D2*. Lower CS/CA is better for the erased concept (↓), higher is better for preserved concepts (↑). 500 iterations for non-variational and 150 iterations for variational. Base model: SD 1.4. The colors green and red are assigned to facilitate the table interpretation regarding only the comparison between MSE and H-DMU. For all methods we use the same fine-tuning and evaluation prompts.

| Method | Erased (R2D2) | | Baymax | | Wall E | | C-3PO | | Bb8 | |
| --- | --- | --- | --- | --- | --- | --- | --- | --- | --- | --- |
| | CS (↓) | CA (↓) | CS (↑) | CA (↑) | CS (↑) | CA (↑) | CS (↑) | CA (↑) | CS (↑) | CA (↑) |
| Original Model | 0.78 | 1.00 | 0.77 | 0.96 | 0.75 | 0.84 | 0.77 | 0.90 | 0.74 | 0.90 |
| MSE (Closed-Form) | 0.62 | 0.04 | 0.75 | 0.93 | 0.72 | 0.74 | 0.75 | 0.88 | 0.70 | 0.86 |
| Hellinger (Closed-Form) | **0.60** | **0.01** | **0.76** | **0.94** | 0.73 | 0.74 | 0.76 | 0.84 | 0.71 | 0.88 |
| $\chi^2$ (Closed-Form) | 0.62 | **0.01** | 0.74 | 0.92 | 0.73 | 0.80 | **0.77** | **0.90** | 0.71 | 0.90 |
| KL (Variational) | 0.63 | 0.55 | 0.75 | 0.92 | **0.74** | **0.86** | 0.76 | 0.86 | 0.69 | 0.90 |
| Hellinger (Variational) | 0.65 | 0.55 | 0.75 | **0.94** | 0.73 | 0.84 | 0.76 | 0.86 | 0.69 | 0.88 |
| Jensen-Shannon (Variational) | 0.65 | 0.40 | 0.76 | **0.94** | 0.73 | 0.80 | 0.76 | 0.86 | 0.71 | 0.90 |
| $\chi^2$ (Variational) | 0.71 | 0.74 | 0.75 | 0.92 | 0.73 | 0.80 | 0.76 | 0.84 | **0.72** | **0.92** |

the target concept is guided towards a more general, categorical prompt (e.g., erasing "Van Gogh" by guiding it towards the generic prompt "a painting"). This method tests the ability to abstract a specific instance into its broader category. Using MSE and H-DMU to replace a specific target concept with a specific anchor concept, our results show that $H^2$ offers a superior trade-off between effective concept removal and the visual quality of the resulting generation. This is demonstrated qualitatively in Fig. 15 and is supported by the quantitative results in Tab. 14.

Finally, Tab. 13 is the extended version of Tab. 2, where we explicitly report the CS, CA, and KID for the five unerased artists. For Tab. 13, we use the same fine-tuning and evaluation prompts for each method, which are the same prompts that can be found in the GitHub repository of CAbl[6].

Fig. 16 shows a qualitative example of convergence during unlearning using different losses: MSE (top row), closed-form-based squared Hellinger distance (central row), and variational-based squared Hellinger distance (bottom row). Each column represents a different iteration (increasing from left to right) during the unlearning phase. While Fig. 16 shows the progressive increase in the difference between the generations obtained from the two closed-form MSE and H-DMU, it also indicates that the variational loss performs a faster and more aggressive erasure.

Fig. 17 shows some qualitative examples of preservation of non-target concepts between different $f$-divergence-based losses.

---

[6] https://github.com/nupurkmr9/concept-ablation

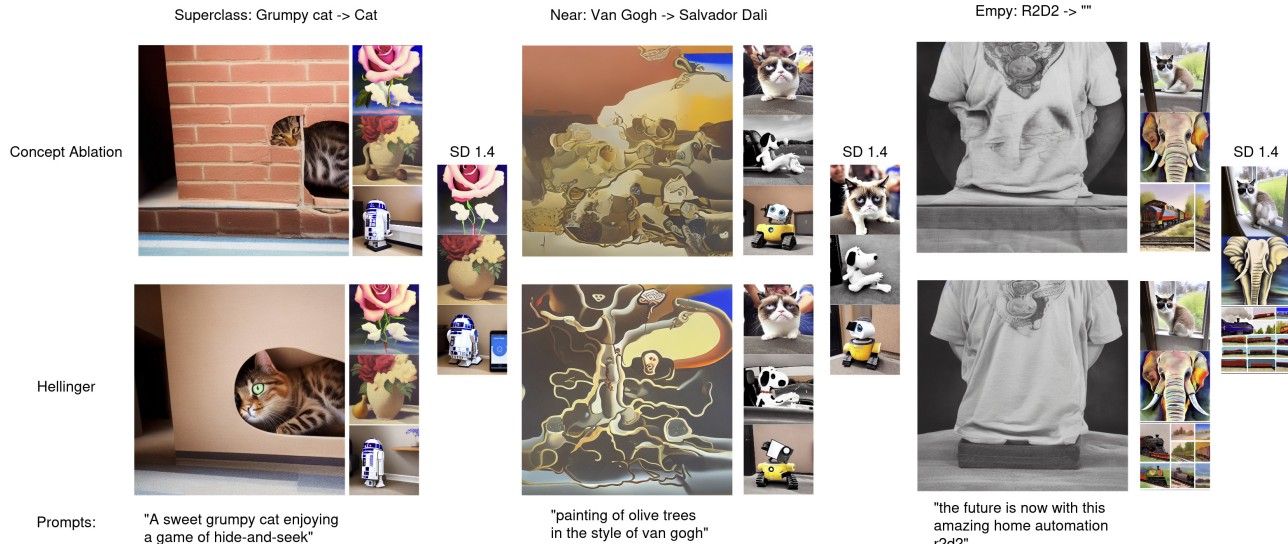

*Figure 15.* Replacement of the target concept with different types of anchor concepts. From left to right: superclass anchor, near anchor and empty anchor. Base model: SD 1.4.

*Table 13.* Unlearning *Van Gogh* on SD 2.1. Comparison between divergence-based methods. The colors green and red are assigned to facilitate the table interpretation regarding only the comparison between MSE and H-DMU. For all methods we use the same fine-tuning and evaluation prompts.

| Method | Erased (Van Gogh) | | J. Mann | | | J. Vermeer | | | S. Dali | | | G. Rutkowski | | | C. Monet | | |
| | CS (↓) | CA (↓) | CS (↑) | CA (↑) | KID (↓) | CS (↑) | CA (↑) | KID (↓) | CS (↑) | CA (↑) | KID (↓) | CS (↑) | CA (↑) | KID (↓) | CS (↑) | CA (↑) | KID (↓) |
|---|---|---|---|---|---|---|---|---|---|---|---|---|---|---|---|---|---|
| CAbl (Kumari et al., 2023) | 0.635 | 0.2 | 0.756 | 1.0 | -0.030 | 0.703 | 0.8 | -0.033 | 0.639 | 0.6 | -0.014 | 0.553 | 0.5 | -0.042 | 0.688 | 1.0 | 0.008 |
| DoCo (Wu et al., 2025) | 0.737 | 0.9 | 0.763 | 1.0 | -0.014 | 0.752 | 1.0 | -0.023 | 0.679 | 0.9 | -0.019 | 0.551 | 0.4 | -0.013 | 0.710 | 1.0 | 0.095 |
| Hellinger (Closed-Form) | 0.624 | 0.2 | 0.765 | 1.0 | -0.029 | 0.706 | 0.9 | -0.029 | 0.656 | 0.6 | 0.012 | 0.554 | 0.4 | -0.061 | 0.679 | 1.0 | 0.005 |
| $\chi^2$ (Closed-Form) | 0.628 | 0.1 | 0.776 | 1.0 | -0.034 | 0.688 | 0.9 | -0.041 | 0.639 | 0.5 | -0.011 | 0.552 | 0.5 | -0.045 | 0.707 | 0.9 | 0.008 |
| KL (Variational) | 0.755 | 1.0 | 0.749 | 1.0 | -0.039 | 0.730 | 0.9 | -0.030 | 0.646 | 0.7 | -0.030 | 0.550 | 0.5 | -0.040 | 0.690 | 1.0 | 0.087 |
| Hellinger (Variational) | 0.645 | 0.5 | 0.737 | 1.0 | -0.012 | 0.794 | 1.0 | 0.023 | 0.671 | 1.0 | -0.002 | 0.567 | 0.5 | -0.043 | 0.739 | 0.9 | 0.173 |
| Jensen-Shannon (Variational) | 0.738 | 0.8 | 0.752 | 1.0 | 0.010 | 0.708 | 0.9 | -0.032 | 0.665 | 0.8 | -0.017 | 0.560 | 0.4 | -0.021 | 0.686 | 0.9 | 0.109 |

## C.7. Ablation Efficacy and Preservation

**Regularization**    To ensure a stable unlearning process that does not degrade the model's overall performance, we incorporate several key techniques, studying their effect in Tabs. 14, 15, 16, where we use a broader set of training prompts (approximately ten times larger compared to the number of prompts used for the other experiments) compared to the other experiments in the paper. We tested different regularization techniques: with the symbol [‡], we indicate the experiment using prior preservation loss (Gandikota et al., 2023) and gradient surgery (Yu et al., 2020b); with the symbol [†] we indicate that we sample only the final part of the DM generation trajectory (Lu et al., 2024) (referred to as importance sampling). The effects of regularization, namely Prior Preservation/Gradient Surgery ([‡]) and Importance Sampling ([†]), are highly context-dependent. In optimization runs of different length, such as the 500 and 5000-step experiments (Table 14 and Table 15), these methods lead to different trade-offs. For instance, when erasing "Snoopy" for 500 steps, the non-regularized "hellinger empty" method is most effective at erasure (CS 0.54, CA 0.47) but produces less realistic images (KID 0.269). Applying regularization weakens the erasure effect (e.g., "hellinger empty [‡] [†]" has CS 0.61, CA 1.00) but improves image coherence, lowering the KID to 0.061.

**Tracking by number of iterations**    The number of training iterations seems to strongly change the effectiveness and retention of unlearning. The larger the number of steps (ranging from 500 to 5000), the more thoroughly the target concept is erased, supported by consistently lower CS and CA values for the erased concept. For example, with 500 steps, "hellinger empty" for "Snoopy" achieves CS=0.54, CA=0.47, KID=0.269 (Table 14), while for 5000 steps, CS=0.52, CA=0.1, KID=0.346 (in Table 15). While CS and CA values decrease, the KID value increases.

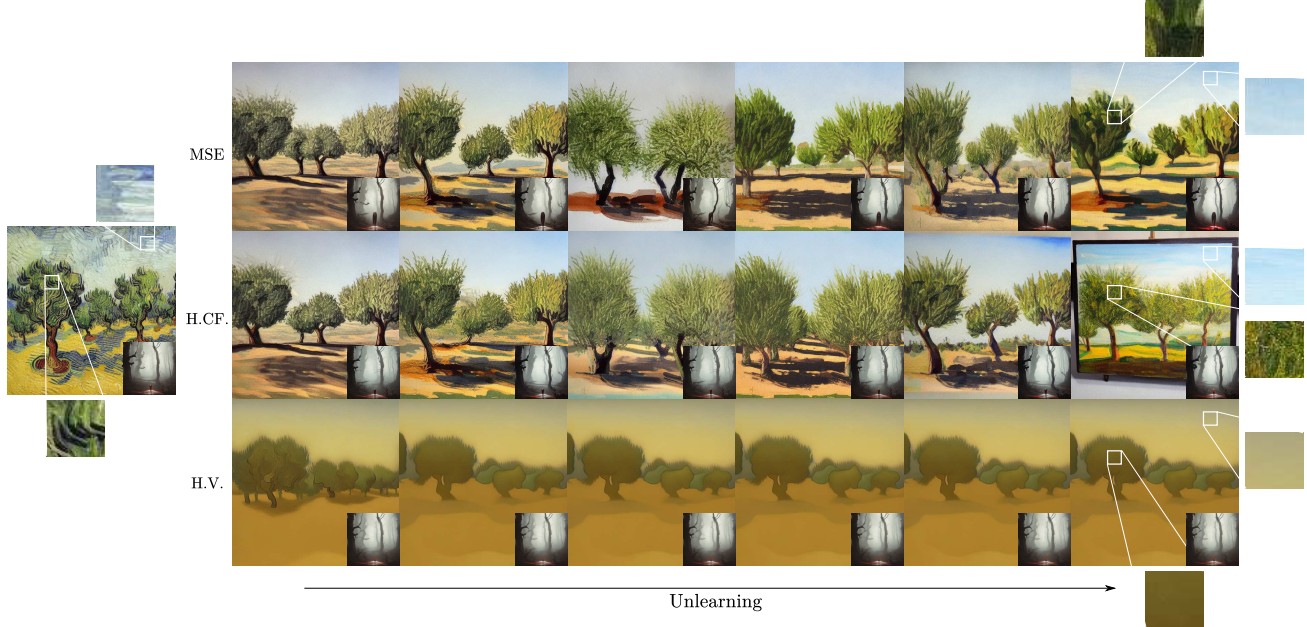

*Figure 16.* Unlearning a concept using different $f$-divergences yield different unlearning dynamics (top: mean squared error, middle: Hellinger closed-form, bottom: Hellinger variational). The inset in the bottom right corner of each image reports a concept not erased. The white boxes are zoomed-in portions to show the removal of the Van Gogh brushstroke style.

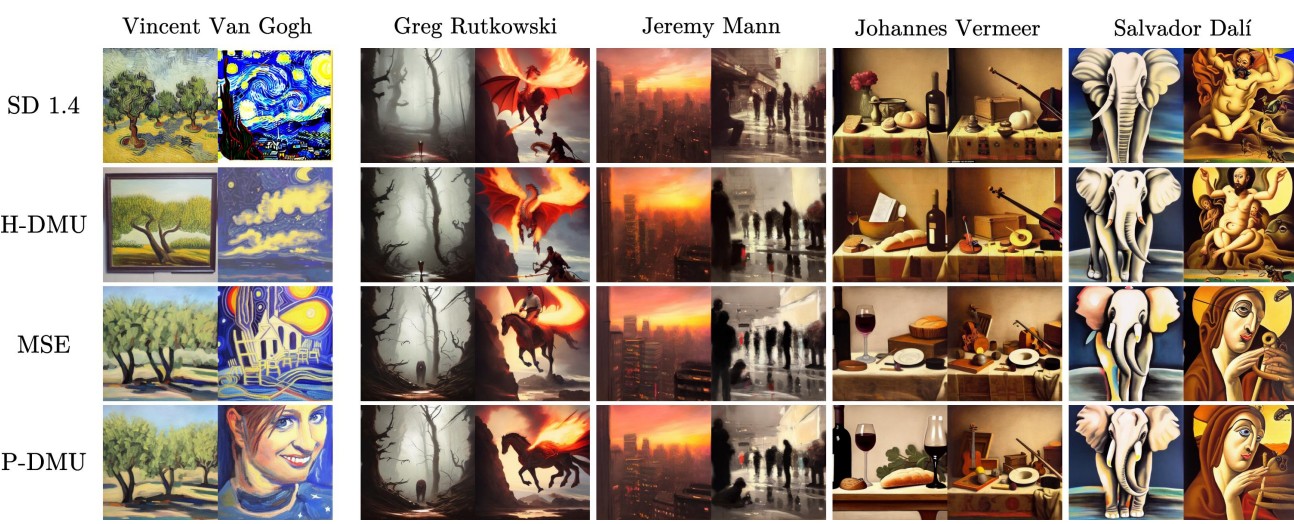

*Figure 17.* Erasing Van Gogh on SD1.4. Qualitative comparison on the preservation of non-target concepts. Each column of the image comprises two side-by-side paintings (of different artists) generated by SD1.4 and its erased versions.

*Table 14.* Quantitative results with 500 iterations, reporting CS, CA, and KID. For methods other than the original model, the best values are in bold and the second-best in italics. [†] denotes Importance Sampling and [‡] denotes Prior Preservation and Gradient Surgery. Base model: SD 1.4.

| | Snoopy | | | Grumpy Cat | | | Wall-E | | | R2D2 | | | Van Gogh | | | Salvador Dali | | |
|---|---|---|---|---|---|---|---|---|---|---|---|---|---|---|---|---|---|---|

*Erasing Snoopy*

| Method | CS↓ | CA↓ | KID | CS↑ | CA↑ | KID↓ | CS↑ | CA↑ | KID↓ | CS↑ | CA↑ | KID↓ | CS↑ | CA↑ | KID↓ | CS↑ | CA↑ | KID↓ |
|---|---|---|---|---|---|---|---|---|---|---|---|---|---|---|---|---|---|---|
| Original Model | 0.73 | 1.00 | - | 0.74 | 1.00 | - | 0.73 | 1.00 | - | 0.75 | 1.00 | - | 0.80 | 1.00 | - | 0.68 | 0.80 | - |
| (Ours) hellinger empty | *0.54* | *0.47* | 0.269 | 0.74 | 0.90 | **0.001** | 0.61 | 0.77 | 0.064 | **0.77** | 1.00 | 0.063 | 0.77 | **1.00** | −0.012 | *0.68* | 0.80 | **−0.014** |
| (Ours) hellinger empty [‡] | 0.59 | 0.83 | 0.059 | 0.71 | **1.00** | 0.025 | 0.70 | *0.93* | 0.020 | 0.76 | 1.00 | 0.035 | 0.76 | 1.00 | −0.022 | 0.68 | 0.83 | −0.024 |
| (Ours) hellinger empty [‡] [†] | 0.61 | 1.00 | 0.061 | 0.73 | **1.00** | *−0.003* | 0.71 | *0.93* | 0.010 | **0.77** | 1.00 | 0.043 | 0.79 | 1.00 | −0.013 | 0.68 | 0.90 | *−0.020* |
| (Ours) hellinger near | **0.51** | *0.70* | 0.298 | 0.78 | **1.00** | 0.141 | 0.69 | *0.93* | 0.027 | **0.77** | 1.00 | 0.045 | 0.78 | **1.00** | −0.007 | **0.69** | 0.90 | −0.029 |
| (Ours) hellinger near [‡] | 0.54 | 0.90 | 0.158 | *0.75* | **1.00** | 0.002 | 0.72 | 0.90 | *−0.006* | 0.76 | 1.00 | 0.012 | 0.77 | *0.97* | −0.022 | 0.67 | 0.77 | −0.024 |
| (Ours) hellinger near [‡] [†] | 0.56 | 0.93 | 0.152 | *0.75* | **1.00** | 0.003 | 0.73 | *0.93* | **0.000** | 0.76 | 1.00 | 0.016 | 0.78 | **1.00** | *0.007* | 0.68 | 0.90 | −0.023 |
| (Ours) hellinger superclass | 0.55 | 0.93 | 0.127 | 0.70 | **1.00** | −0.002 | 0.71 | **0.97** | **0.000** | **0.77** | 1.00 | **0.000** | 0.78 | **1.00** | −0.028 | **0.69** | *0.93* | −0.031 |
| (Ours) hellinger superclass [‡] | 0.61 | 1.00 | 0.062 | 0.72 | **1.00** | −0.006 | **0.76** | **0.97** | 0.003 | 0.76 | 1.00 | −0.001 | **0.79** | **1.00** | −0.017 | 0.68 | 0.80 | −0.021 |
| (Ours) hellinger superclass [‡] [†] | 0.61 | 1.00 | 0.069 | 0.74 | **1.00** | −0.012 | **0.77** | **0.97** | **0.000** | 0.76 | 1.00 | −0.001 | **0.79** | **1.00** | **−0.004** | **0.69** | **0.97** | −0.025 |
| Concept Ablation | 0.56 | 1.00 | 0.127 | 0.68 | **1.00** | 0.016 | 0.71 | 0.90 | *−0.005* | **0.77** | 1.00 | 0.006 | 0.78 | **1.00** | *−0.023* | 0.68 | 0.90 | *−0.029* |

*Erasing Grumpy Cat*

| Method | CS↑ | CA↑ | KID↓ | CS↓ | CA↓ | KID | CS↑ | CA↑ | KID↓ | CS↑ | CA↑ | KID↓ | CS↑ | CA↑ | KID↓ | CS↑ | CA↑ | KID↓ |
|---|---|---|---|---|---|---|---|---|---|---|---|---|---|---|---|---|---|---|
| Original Model | 0.73 | 1.00 | - | 0.74 | 1.00 | - | 0.73 | 1.00 | - | 0.75 | 1.00 | - | 0.80 | 1.00 | - | 0.68 | 0.80 | - |
| (Ours) hellinger empty | 0.65 | **1.00** | 0.010 | **0.47** | **0.03** | 0.250 | 0.61 | 0.83 | 0.055 | 0.75 | 1.00 | 0.044 | 0.77 | **1.00** | **−0.009** | 0.69 | 0.83 | *−0.012* |
| (Ours) hellinger empty [‡] | 0.71 | **1.00** | −0.014 | 0.65 | 0.93 | 0.056 | *0.75* | **1.00** | **−0.001** | 0.76 | 1.00 | **0.003** | 0.77 | *0.97* | −0.013 | 0.68 | 0.83 | −0.029 |
| (Ours) hellinger empty [‡] [†] | 0.70 | **1.00** | −0.008 | 0.62 | 0.87 | 0.058 | 0.73 | **1.00** | *0.002* | 0.75 | 1.00 | 0.010 | 0.77 | **1.00** | −0.029 | **0.70** | 0.90 | −0.030 |
| (Ours) hellinger near | **0.75** | **1.00** | 0.030 | *0.52* | *0.23* | 0.223 | 0.74 | 0.90 | 0.028 | 0.76 | 1.00 | 0.036 | 0.78 | **1.00** | −0.016 | 0.69 | 0.80 | −0.027 |
| (Ours) hellinger near [‡] | 0.70 | **1.00** | −0.015 | 0.63 | 0.97 | 0.062 | *0.75* | 0.97 | *−0.009* | 0.76 | 1.00 | 0.011 | 0.78 | **1.00** | −0.012 | 0.68 | *0.87* | **−0.011** |
| (Ours) hellinger near [‡] [†] | 0.70 | **1.00** | **−0.002** | 0.61 | 1.00 | 0.064 | 0.73 | 0.93 | *−0.003* | 0.75 | 1.00 | 0.013 | 0.75 | *0.97* | **0.009** | 0.68 | 0.83 | −0.022 |
| (Ours) hellinger superclass | 0.70 | **1.00** | *−0.004* | 0.57 | 0.83 | 0.140 | 0.74 | 0.93 | 0.002 | **0.77** | 1.00 | 0.030 | 0.78 | **1.00** | −0.026 | 0.69 | 0.83 | −0.024 |
| (Ours) hellinger superclass [‡] | 0.70 | **1.00** | −0.012 | 0.68 | 1.00 | 0.041 | 0.73 | 0.97 | −0.013 | 0.75 | 1.00 | *0.009* | 0.78 | **1.00** | −0.011 | 0.68 | 0.80 | −0.019 |
| (Ours) hellinger superclass [‡] [†] | *0.72* | **1.00** | −0.019 | 0.66 | 1.00 | 0.035 | **0.76** | 0.97 | −0.004 | 0.75 | 1.00 | **0.003** | 0.78 | **1.00** | −0.011 | 0.69 | 0.90 | −0.034 |
| Concept Ablation | 0.69 | **1.00** | −0.013 | 0.58 | 0.97 | 0.147 | 0.72 | 0.93 | −0.003 | **0.77** | 1.00 | 0.038 | 0.78 | **1.00** | *−0.026* | 0.69 | 0.90 | −0.025 |

*Erasing Wall-E*

| Method | CS↑ | CA↑ | KID↓ | CS↑ | CA↑ | KID↓ | CS↓ | CA↓ | KID | CS↑ | CA↑ | KID↓ | CS↑ | CA↑ | KID↓ | CS↑ | CA↑ | KID↓ |
|---|---|---|---|---|---|---|---|---|---|---|---|---|---|---|---|---|---|---|
| Original Model | 0.73 | 1.00 | - | 0.74 | 1.00 | - | 0.73 | 1.00 | - | 0.75 | 1.00 | - | 0.80 | 1.00 | - | 0.68 | 0.80 | - |
| (Ours) hellinger empty | 0.66 | **1.00** | 0.023 | 0.72 | 0.97 | −0.011 | **0.55** | **0.77** | 0.089 | 0.71 | 0.90 | 0.092 | 0.77 | **1.00** | −0.021 | 0.68 | 0.70 | −0.025 |
| (Ours) hellinger empty [‡] | 0.68 | **1.00** | *−0.007* | 0.75 | **1.00** | −0.016 | 0.57 | **0.77** | 0.071 | *0.76* | **1.00** | 0.054 | 0.78 | **1.00** | **0.004** | 0.69 | 0.90 | −0.019 |
| (Ours) hellinger empty [‡] [†] | 0.69 | **1.00** | 0.004 | 0.73 | **1.00** | −0.014 | 0.57 | 0.90 | 0.057 | 0.75 | *0.97* | 0.038 | 0.78 | **1.00** | −0.002 | 0.69 | 0.90 | −0.029 |
| (Ours) hellinger near | 0.71 | **1.00** | −0.019 | 0.74 | **1.00** | −0.012 | *0.56* | 0.97 | 0.194 | *0.76* | **1.00** | 0.097 | 0.76 | **1.00** | −0.025 | 0.68 | 0.80 | *−0.016* |
| (Ours) hellinger near [‡] | **0.73** | **1.00** | −0.013 | 0.75 | **1.00** | **−0.008** | 0.59 | 0.97 | 0.153 | *0.76* | **1.00** | 0.039 | 0.78 | **1.00** | *0.006* | 0.69 | 0.83 | −0.022 |
| (Ours) hellinger near [‡] [†] | 0.71 | **1.00** | −0.010 | 0.74 | **1.00** | −0.012 | 0.59 | 1.00 | 0.146 | **0.77** | **1.00** | 0.089 | 0.78 | **1.00** | 0.010 | 0.69 | *0.87* | −0.017 |
| (Ours) hellinger superclass | **0.73** | **1.00** | −0.008 | 0.74 | **1.00** | −0.016 | 0.61 | 0.90 | 0.092 | *0.76* | **1.00** | 0.033 | 0.79 | **1.00** | −0.028 | 0.69 | 0.83 | **−0.015** |
| (Ours) hellinger superclass [‡] | *0.72* | **1.00** | **−0.006** | 0.74 | **1.00** | **−0.008** | 0.68 | 0.90 | 0.072 | *0.76* | **1.00** | 0.039 | 0.79 | *0.97* | **0.004** | 0.69 | 0.90 | −0.021 |
| (Ours) hellinger superclass [‡] [†] | *0.72* | **1.00** | *−0.018* | 0.74 | **1.00** | −0.009 | 0.68 | 0.93 | 0.064 | **0.77** | **1.00** | 0.091 | 0.80 | **1.00** | **0.004** | **0.70** | 0.93 | −0.024 |
| Concept Ablation | **0.73** | **1.00** | −0.017 | 0.73 | **1.00** | −0.009 | 0.60 | *0.83* | 0.094 | *0.76* | **1.00** | 0.065 | 0.78 | **1.00** | −0.021 | 0.68 | 0.80 | −0.021 |

*Erasing R2D2*

| Method | CS↑ | CA↑ | KID↓ | CS↑ | CA↑ | KID↓ | CS↑ | CA↑ | KID↓ | CS↓ | CA↓ | KID | CS↑ | CA↑ | KID↓ | CS↑ | CA↑ | KID↓ |
|---|---|---|---|---|---|---|---|---|---|---|---|---|---|---|---|---|---|---|
| Original Model | 0.73 | 1.00 | - | 0.74 | 1.00 | - | 0.73 | 1.00 | - | 0.75 | 1.00 | - | 0.80 | 1.00 | - | 0.68 | 0.80 | - |
| (Ours) hellinger empty | 0.62 | **1.00** | 0.049 | 0.74 | 0.97 | −0.010 | 0.58 | 0.70 | 0.102 | **0.53** | **0.20** | 0.252 | 0.76 | **1.00** | *−0.008* | 0.68 | 0.70 | **−0.011** |
| (Ours) hellinger empty [‡] | 0.69 | **1.00** | **−0.003** | 0.73 | **1.00** | −0.017 | 0.72 | 0.93 | **0.003** | 0.59 | 0.80 | *0.186* | 0.75 | *0.97* | −0.019 | **0.70** | *0.87* | −0.023 |
| (Ours) hellinger empty [‡] [†] | 0.69 | **1.00** | **−0.004** | 0.71 | **1.00** | −0.006 | 0.74 | 0.93 | 0.011 | 0.58 | *0.77* | **0.154** | 0.77 | *0.97* | −0.009 | **0.70** | 0.83 | −0.032 |
| (Ours) hellinger near | **0.75** | **1.00** | 0.007 | 0.73 | **1.00** | **−0.001** | **0.81** | **1.00** | 0.100 | 0.58 | *0.77* | 0.348 | 0.77 | **1.00** | *−0.008* | **0.70** | 0.90 | *−0.018* |
| (Ours) hellinger near [‡] | 0.73 | **1.00** | −0.008 | 0.74 | **1.00** | −0.012 | 0.78 | 0.97 | 0.008 | 0.64 | 1.00 | 0.235 | 0.79 | **1.00** | −0.015 | **0.70** | 0.90 | −0.023 |
| (Ours) hellinger near [‡] [†] | *0.74* | **1.00** | −0.013 | 0.73 | **1.00** | −0.016 | *0.79* | **1.00** | 0.024 | 0.64 | 1.00 | 0.219 | 0.80 | **1.00** | **−0.006** | 0.69 | 0.83 | −0.029 |
| (Ours) hellinger superclass | *0.74* | **1.00** | −0.014 | 0.72 | **1.00** | −0.016 | 0.76 | 0.93 | 0.027 | **0.54** | 0.93 | 0.225 | 0.79 | **1.00** | −0.031 | **0.70** | 0.90 | −0.029 |
| (Ours) hellinger superclass [‡] | *0.74* | **1.00** | −0.018 | 0.74 | **1.00** | −0.007 | 0.75 | 0.93 | 0.005 | 0.64 | 1.00 | 0.203 | 0.79 | **1.00** | −0.010 | **0.70** | 0.90 | −0.024 |
| (Ours) hellinger superclass [‡] [†] | 0.72 | **1.00** | −0.018 | 0.75 | **1.00** | −0.017 | 0.77 | **1.00** | 0.010 | 0.67 | 1.00 | 0.187 | 0.80 | **1.00** | 0.013 | **0.71** | 0.90 | −0.021 |
| Concept Ablation | *0.74* | **1.00** | −0.009 | 0.73 | **1.00** | **0.001** | 0.76 | *0.97* | 0.030 | 0.55 | 0.93 | 0.244 | 0.78 | **1.00** | −0.039 | **0.70** | 0.90 | −0.031 |

*Erasing Van Gogh*

| Method | CS↑ | CA↑ | KID↓ | CS↑ | CA↑ | KID↓ | CS↑ | CA↑ | KID↓ | CS↑ | CA↑ | KID↓ | CS↓ | CA↓ | KID | CS↑ | CA↑ | KID↓ |
|---|---|---|---|---|---|---|---|---|---|---|---|---|---|---|---|---|---|---|
| Original Model | 0.73 | 1.00 | - | 0.74 | 1.00 | - | 0.73 | 1.00 | - | 0.75 | 1.00 | - | 0.80 | 1.00 | - | 0.68 | 0.80 | - |
| (Ours) hellinger empty | 0.69 | **1.00** | *−0.014* | 0.74 | **1.00** | **−0.007** | 0.67 | **1.00** | 0.012 | **0.77** | **1.00** | 0.029 | **0.55** | 0.50 | 0.151 | 0.62 | 0.70 | 0.035 |
| (Ours) hellinger empty [‡] | *0.74* | **1.00** | **−0.011** | 0.74 | **1.00** | −0.011 | **0.77** | 0.93 | 0.007 | 0.75 | **1.00** | **−0.000** | 0.72 | *0.97* | 0.073 | **0.70** | *0.97* | *−0.004* |
| (Ours) hellinger empty [‡] [†] | 0.73 | **1.00** | −0.020 | 0.75 | **1.00** | −0.016 | *0.76* | *0.97* | 0.007 | 0.75 | **1.00** | *−0.002* | 0.72 | 0.90 | 0.098 | **0.70** | 0.90 | **0.001** |
| (Ours) hellinger near | 0.72 | **1.00** | −0.017 | 0.74 | **1.00** | *−0.010* | 0.74 | 0.97 | −0.012 | **0.77** | **1.00** | 0.008 | *0.56* | 0.90 | 0.186 | **0.70** | *0.97* | 0.049 |
| (Ours) hellinger near [‡] | **0.75** | **1.00** | −0.017 | 0.74 | **1.00** | −0.016 | *0.76* | *0.97* | **0.003** | 0.75 | **1.00** | 0.010 | 0.73 | 0.90 | 0.068 | **0.70** | 0.90 | 0.006 |
| (Ours) hellinger near [‡] [†] | *0.74* | **1.00** | −0.018 | 0.75 | **1.00** | −0.013 | 0.75 | 0.90 | 0.013 | 0.75 | **1.00** | 0.004 | 0.70 | 0.87 | 0.075 | **0.72** | **1.00** | *−0.004* |
| (Ours) hellinger superclass | 0.73 | **1.00** | −0.023 | 0.74 | **1.00** | −0.028 | 0.73 | 0.93 | *−0.003* | **0.76** | **1.00** | −0.007 | 0.58 | *0.17* | 0.132 | 0.64 | 0.57 | **0.001** |
| (Ours) hellinger superclass [‡] | *0.74* | **1.00** | −0.015 | 0.74 | **1.00** | −0.016 | **0.77** | **1.00** | *0.006* | 0.74 | **1.00** | 0.006 | 0.71 | 0.93 | 0.081 | 0.69 | 0.90 | −0.019 |
| (Ours) hellinger superclass [‡] [†] | *0.74* | **1.00** | −0.022 | 0.73 | **1.00** | −0.019 | 0.75 | 0.93 | 0.014 | 0.75 | **1.00** | 0.007 | 0.67 | 0.73 | 0.093 | 0.69 | 0.90 | −0.013 |
| Concept Ablation | *0.74* | **1.00** | −0.023 | *0.74* | **1.00** | −0.019 | 0.74 | 0.93 | *−0.003* | **0.76** | **1.00** | −0.004 | 0.57 | 0.20 | 0.117 | 0.64 | 0.63 | *0.004* |

*Erasing Salvador Dali*

| Method | CS↑ | CA↑ | KID↓ | CS↑ | CA↑ | KID↓ | CS↑ | CA↑ | KID↓ | CS↑ | CA↑ | KID↓ | CS↑ | CA↑ | KID↓ | CS↓ | CA↓ | KID |
|---|---|---|---|---|---|---|---|---|---|---|---|---|---|---|---|---|---|---|
| Original Model | 0.73 | 1.00 | - | 0.74 | 1.00 | - | 0.73 | 1.00 | - | 0.75 | 1.00 | - | 0.80 | 1.00 | - | 0.68 | 0.80 | - |
| (Ours) hellinger empty | 0.70 | **1.00** | −0.015 | *0.74* | **1.00** | −0.012 | 0.69 | 0.90 | 0.011 | **0.78** | **1.00** | 0.024 | 0.71 | *0.97* | 0.055 | *0.59* | 0.73 | 0.091 |
| (Ours) hellinger empty [‡] | *0.74* | **1.00** | −0.014 | *0.74* | **1.00** | −0.013 | 0.76 | 0.93 | 0.007 | 0.75 | **1.00** | 0.005 | 0.76 | *0.97* | 0.051 | 0.67 | 0.77 | 0.008 |
| (Ours) hellinger empty [‡] [†] | 0.73 | **1.00** | −0.018 | 0.75 | **1.00** | **−0.006** | 0.76 | 0.90 | 0.016 | 0.74 | **1.00** | 0.009 | 0.79 | **1.00** | 0.064 | 0.67 | 0.87 | **0.003** |
| (Ours) hellinger near | 0.74 | **1.00** | −0.017 | *0.74* | **1.00** | −0.014 | 0.75 | 0.93 | 0.012 | 0.76 | **1.00** | 0.024 | 0.80 | **1.00** | 0.069 | 0.62 | *0.17* | 0.199 |
| (Ours) hellinger near [‡] | **0.76** | **1.00** | *−0.011* | 0.73 | **1.00** | **−0.007** | 0.75 | 0.90 | *0.006* | 0.75 | **1.00** | **0.000** | 0.77 | *0.97* | 0.047 | 0.67 | 0.73 | 0.028 |
| (Ours) hellinger near [‡] [†] | *0.74* | **1.00** | −0.019 | 0.73 | **1.00** | −0.019 | 0.78 | 0.93 | 0.009 | 0.75 | **1.00** | −0.003 | 0.78 | **1.00** | 0.039 | 0.66 | 0.57 | 0.043 |
| (Ours) hellinger superclass | *0.75* | **1.00** | **−0.007** | 0.75 | **1.00** | −0.008 | 0.75 | 0.90 | 0.009 | 0.74 | **1.00** | −0.009 | 0.75 | **1.00** | 0.050 | 0.60 | 0.30 | 0.054 |
| (Ours) hellinger superclass [‡] | *0.74* | **1.00** | −0.013 | 0.75 | **1.00** | −0.009 | 0.76 | 0.93 | 0.011 | 0.74 | **1.00** | **−0.000** | 0.77 | *0.97* | 0.066 | 0.65 | 0.70 | 0.008 |
| (Ours) hellinger superclass [‡] [†] | 0.73 | **1.00** | −0.017 | *0.74* | **1.00** | −0.018 | 0.77 | 0.93 | 0.002 | 0.75 | **1.00** | −0.004 | 0.78 | **1.00** | 0.046 | 0.63 | 0.43 | 0.009 |
| Concept Ablation | *0.75* | **1.00** | **−0.007** | 0.75 | **1.00** | −0.015 | 0.77 | 0.93 | 0.018 | 0.74 | **1.00** | *−0.001* | 0.75 | **1.00** | 0.100 | **0.58** | **0.00** | 0.056 |

Table 15. Quantitative results with 5000 iterations, reporting CS, CA, and KID. For methods other than the Original Model, the best values are in bold and the second-best in italics. † denotes Importance Sampling and ‡ denotes Prior Preservation and Gradient Surgery. Base model: SD 1.4.

### Erasing Snoopy

| Method | Snoopy CS↓ | CA↓ | KID | Grumpy Cat CS↑ | CA↑ | KID↓ | Wall-E CS↑ | CA↑ | KID↓ | R2D2 CS↑ | CA↑ | KID↓ | Van Gogh CS↑ | CA↑ | KID↓ | Salvador Dali CS↑ | CA↑ | KID↓ |
|---|---|---|---|---|---|---|---|---|---|---|---|---|---|---|---|---|---|---|
| Original Model | 0.76 | 1.00 | - | 0.71 | 0.98 | - | 0.70 | 0.97 | - | 0.73 | 0.98 | - | 0.77 | 0.97 | - | 0.66 | 0.76 | - |
| (Ours) hellinger empty | 0.52 | 0.10 | 0.346 | 0.50 | 0.00 | 0.341 | 0.58 | 0.85 | 0.108 | 0.65 | 1.00 | 0.170 | 0.72 | 0.95 | 0.060 | 0.62 | 0.90 | 0.057 |
| (Ours) hellinger near | 0.56 | 0.70 | 0.320 | 0.74 | 0.97 | 0.302 | 0.70 | 0.97 | 0.047 | 0.78 | 1.00 | 0.072 | 0.73 | 1.00 | 0.060 | 0.66 | 0.60 | 0.029 |
| (Ours) hellinger superclass | 0.54 | 0.50 | 0.141 | 0.61 | 0.70 | 0.129 | 0.61 | 0.57 | 0.092 | 0.71 | 0.87 | 0.174 | 0.71 | 0.87 | 0.053 | 0.66 | 0.83 | 0.038 |
| (Ours) hellinger superclass ‡ | 0.61 | 1.00 | 0.097 | 0.70 | 1.00 | 0.034 | 0.75 | 0.90 | 0.032 | 0.76 | 1.00 | 0.010 | 0.74 | 0.90 | 0.011 | 0.66 | 0.80 | 0.005 |
| (Ours) mse empty | 0.56 | 0.08 | 0.312 | 0.43 | 0.12 | 0.303 | 0.51 | 0.87 | 0.126 | 0.56 | 0.96 | 0.208 | 0.66 | 0.82 | 0.072 | 0.59 | 0.67 | 0.038 |
| (Ours) mse near | 0.65 | 0.32 | 0.376 | 0.56 | 0.48 | 0.415 | 0.59 | 0.82 | 0.151 | 0.71 | 0.90 | 0.104 | 0.65 | 0.81 | 0.057 | 0.59 | 0.38 | 0.076 |
| Concept Ablation | 0.64 | 0.43 | 0.222 | 0.51 | 0.55 | 0.185 | 0.66 | 0.79 | 0.053 | 0.71 | 0.95 | 0.048 | 0.68 | 0.89 | 0.019 | 0.61 | 0.68 | 0.007 |

### Erasing Grumpy Cat

| Method | Snoopy CS↑ | CA↑ | KID↓ | Grumpy Cat CS↓ | CA↓ | KID | Wall-E CS↑ | CA↑ | KID↓ | R2D2 CS↑ | CA↑ | KID↓ | Van Gogh CS↑ | CA↑ | KID↓ | Salvador Dali CS↑ | CA↑ | KID↓ |
|---|---|---|---|---|---|---|---|---|---|---|---|---|---|---|---|---|---|---|
| Original Model | 0.70 | 0.97 | - | 0.77 | 1.00 | - | 0.70 | 0.98 | - | 0.73 | 0.97 | - | 0.77 | 0.97 | - | 0.65 | 0.78 | - |
| (Ours) hellinger empty | 0.58 | 0.70 | 0.105 | 0.50 | 0.10 | 0.409 | 0.69 | 0.95 | 0.139 | 0.75 | 1.00 | 0.096 | 0.69 | 0.95 | 0.058 | 0.69 | 1.00 | 0.014 |
| (Ours) hellinger near | 0.72 | 1.00 | 0.244 | 0.53 | 0.00 | 0.360 | 0.72 | 0.97 | 0.102 | 0.77 | 1.00 | 0.075 | 0.74 | 1.00 | 0.032 | 0.66 | 0.90 | 0.053 |
| (Ours) hellinger superclass | 0.64 | 0.90 | 0.095 | 0.54 | 0.40 | 0.280 | 0.66 | 0.83 | 0.113 | 0.72 | 0.90 | 0.157 | 0.68 | 0.83 | 0.069 | 0.64 | 0.70 | 0.065 |
| (Ours) hellinger superclass ‡ | 0.73 | 1.00 | 0.010 | 0.61 | 1.00 | 0.088 | 0.72 | 0.90 | 0.027 | 0.76 | 1.00 | 0.032 | 0.77 | 1.00 | 0.063 | 0.67 | 0.80 | 0.010 |
| (Ours) mse empty | 0.50 | 0.48 | 0.138 | 0.54 | 0.16 | 0.489 | 0.59 | 0.90 | 0.143 | 0.69 | 0.94 | 0.092 | 0.65 | 0.88 | 0.059 | 0.61 | 0.83 | 0.017 |
| (Ours) mse near | 0.58 | 0.50 | 0.328 | 0.59 | 0.04 | 0.435 | 0.68 | 0.79 | 0.112 | 0.73 | 0.93 | 0.117 | 0.68 | 0.91 | 0.037 | 0.61 | 0.63 | 0.053 |
| Concept Ablation | 0.52 | 0.33 | 0.201 | 0.58 | 0.05 | 0.321 | 0.63 | 0.78 | 0.155 | 0.66 | 0.75 | 0.180 | 0.65 | 0.84 | 0.082 | 0.55 | 0.43 | 0.091 |

### Erasing Wall-E

| Method | Snoopy CS↑ | CA↑ | KID↓ | Grumpy Cat CS↑ | CA↑ | KID↓ | Wall-E CS↓ | CA↓ | KID | R2D2 CS↑ | CA↑ | KID↓ | Van Gogh CS↑ | CA↑ | KID↓ | Salvador Dali CS↑ | CA↑ | KID↓ |
|---|---|---|---|---|---|---|---|---|---|---|---|---|---|---|---|---|---|---|
| Original Model | 0.71 | 0.97 | - | 0.71 | 0.97 | - | 0.76 | 1.00 | - | 0.72 | 0.98 | - | 0.77 | 0.97 | - | 0.66 | 0.77 | - |
| (Ours) hellinger empty | 0.58 | 0.95 | 0.064 | 0.66 | 0.85 | 0.017 | 0.55 | 0.75 | 0.197 | 0.53 | 0.55 | 0.243 | 0.72 | 0.90 | 0.036 | 0.67 | 0.70 | 0.010 |
| (Ours) hellinger near | 0.64 | 0.97 | 0.050 | 0.68 | 0.97 | 0.118 | 0.56 | 0.67 | 0.212 | 0.71 | 0.90 | 0.249 | 0.70 | 0.97 | 0.033 | 0.61 | 0.73 | 0.074 |
| (Ours) hellinger superclass | 0.63 | 0.73 | 0.207 | 0.71 | 0.90 | 0.224 | 0.57 | 0.30 | 0.274 | 0.65 | 0.73 | 0.301 | 0.70 | 0.80 | 0.036 | 0.62 | 0.60 | 0.057 |
| (Ours) hellinger superclass ‡ | 0.72 | 1.00 | -0.007 | 0.71 | 1.00 | -0.003 | 0.70 | 1.00 | 0.075 | 0.76 | 1.00 | 0.062 | 0.76 | 1.00 | 0.031 | 0.68 | 0.60 | -0.003 |
| (Ours) mse empty | 0.49 | 0.67 | 0.047 | 0.62 | 0.89 | 0.021 | 0.59 | 0.85 | 0.167 | 0.48 | 0.50 | 0.302 | 0.67 | 0.90 | 0.048 | 0.59 | 0.54 | 0.033 |
| (Ours) mse near | 0.54 | 0.57 | 0.163 | 0.62 | 0.81 | 0.176 | 0.66 | 0.96 | 0.154 | 0.69 | 0.91 | 0.199 | 0.63 | 0.75 | 0.037 | 0.58 | 0.62 | 0.047 |
| Concept Ablation | 0.54 | 0.42 | 0.228 | 0.56 | 0.56 | 0.282 | 0.64 | 0.30 | 0.241 | 0.56 | 0.30 | 0.328 | 0.59 | 0.55 | 0.083 | 0.53 | 0.43 | 0.122 |

### Erasing R2D2

| Method | Snoopy CS↑ | CA↑ | KID↓ | Grumpy Cat CS↑ | CA↑ | KID↓ | Wall-E CS↑ | CA↑ | KID↓ | R2D2 CS↓ | CA↓ | KID | Van Gogh CS↑ | CA↑ | KID↓ | Salvador Dali CS↑ | CA↑ | KID↓ |
|---|---|---|---|---|---|---|---|---|---|---|---|---|---|---|---|---|---|---|
| Original Model | 0.70 | 0.97 | - | 0.71 | 0.97 | - | 0.70 | 0.97 | - | 0.79 | 1.00 | - | 0.77 | 0.97 | - | 0.65 | 0.78 | - |
| (Ours) hellinger empty | 0.58 | 0.65 | 0.030 | 0.73 | 0.95 | 0.020 | 0.53 | 0.70 | 0.128 | 0.54 | 0.65 | 0.319 | 0.71 | 0.90 | 0.054 | 0.66 | 0.70 | 0.023 |
| (Ours) hellinger near | 0.61 | 0.87 | 0.110 | 0.63 | 0.73 | 0.168 | 0.61 | 0.67 | 0.271 | 0.53 | 0.10 | 0.403 | 0.66 | 1.00 | 0.150 | 0.61 | 0.83 | 0.145 |
| (Ours) hellinger superclass | 0.62 | 0.57 | 0.192 | 0.60 | 0.47 | 0.176 | 0.63 | 0.57 | 0.212 | 0.55 | 0.20 | 0.382 | 0.64 | 0.73 | 0.110 | 0.61 | 0.67 | 0.063 |
| (Ours) hellinger superclass ‡ | 0.73 | 1.00 | 0.001 | 0.73 | 1.00 | 0.019 | 0.72 | 1.00 | 0.043 | 0.63 | 1.00 | 0.222 | 0.76 | 1.00 | 0.052 | 0.68 | 0.80 | 0.002 |
| (Ours) mse empty | 0.50 | 0.83 | 0.049 | 0.59 | 0.76 | 0.037 | 0.49 | 0.64 | 0.142 | 0.59 | 0.77 | 0.385 | 0.67 | 0.93 | 0.062 | 0.60 | 0.54 | 0.017 |
| (Ours) mse near | 0.57 | 0.52 | 0.108 | 0.57 | 0.67 | 0.146 | 0.51 | 0.56 | 0.184 | 0.59 | 0.46 | 0.321 | 0.61 | 0.93 | 0.085 | 0.56 | 0.81 | 0.055 |
| Concept Ablation | 0.58 | 0.67 | 0.199 | 0.60 | 0.60 | 0.180 | 0.57 | 0.73 | 0.182 | 0.65 | 0.36 | 0.341 | 0.66 | 0.82 | 0.050 | 0.57 | 0.71 | 0.062 |

### Erasing Van Gogh

| Method | Snoopy CS↑ | CA↑ | KID↓ | Grumpy Cat CS↑ | CA↑ | KID↓ | Wall-E CS↑ | CA↑ | KID↓ | R2D2 CS↑ | CA↑ | KID↓ | Van Gogh CS↓ | CA↓ | KID | Salvador Dali CS↑ | CA↑ | KID↓ |
|---|---|---|---|---|---|---|---|---|---|---|---|---|---|---|---|---|---|---|
| Original Model | 0.70 | 0.97 | - | 0.71 | 0.97 | - | 0.70 | 0.97 | - | 0.72 | 0.97 | - | 0.83 | 1.00 | - | 0.65 | 0.77 | - |
| (Ours) hellinger empty | 0.69 | 0.95 | 0.060 | 0.67 | 0.95 | 0.037 | 0.68 | 1.00 | 0.118 | 0.71 | 1.00 | 0.087 | 0.52 | 0.05 | 0.499 | 0.59 | 0.70 | 0.135 |
| (Ours) hellinger near | 0.71 | 1.00 | 0.031 | 0.68 | 0.90 | 0.075 | 0.75 | 0.97 | 0.059 | 0.78 | 1.00 | 0.088 | 0.64 | 0.53 | 0.275 | 0.71 | 1.00 | 0.188 |
| (Ours) hellinger superclass | 0.69 | 0.97 | 0.059 | 0.74 | 1.00 | 0.090 | 0.71 | 0.97 | 0.062 | 0.73 | 1.00 | 0.139 | 0.58 | 0.10 | 0.145 | 0.61 | 0.40 | 0.082 |
| (Ours) hellinger superclass ‡ | 0.75 | 1.00 | 0.016 | 0.74 | 1.00 | 0.023 | 0.76 | 0.90 | 0.022 | 0.75 | 1.00 | 0.036 | 0.66 | 0.60 | 0.122 | 0.70 | 0.90 | -0.003 |
| (Ours) mse empty | 0.59 | 0.90 | 0.022 | 0.60 | 0.79 | 0.031 | 0.58 | 0.93 | 0.111 | 0.55 | 0.94 | 0.144 | 0.59 | 0.19 | 0.447 | 0.52 | 0.33 | 0.167 |
| (Ours) mse near | 0.67 | 0.95 | 0.056 | 0.63 | 0.94 | 0.099 | 0.70 | 0.93 | 0.055 | 0.73 | 0.95 | 0.080 | 0.65 | 0.37 | 0.292 | 0.61 | 0.93 | 0.209 |
| Concept Ablation | 0.65 | 0.93 | 0.068 | 0.68 | 0.91 | 0.040 | 0.68 | 0.86 | 0.057 | 0.73 | 0.94 | 0.080 | 0.65 | 0.24 | 0.129 | 0.58 | 0.60 | 0.069 |

### Erasing Salvador Dali

| Method | Snoopy CS↑ | CA↑ | KID↓ | Grumpy Cat CS↑ | CA↑ | KID↓ | Wall-E CS↑ | CA↑ | KID↓ | R2D2 CS↑ | CA↑ | KID↓ | Van Gogh CS↑ | CA↑ | KID↓ | Salvador Dali CS↓ | CA↓ | KID |
|---|---|---|---|---|---|---|---|---|---|---|---|---|---|---|---|---|---|---|
| Original Model | 0.70 | 0.97 | - | 0.71 | 0.97 | - | 0.70 | 0.97 | - | 0.73 | 0.96 | - | 0.77 | 0.97 | - | 0.71 | 0.84 | - |
| (Ours) hellinger empty | 0.62 | 0.90 | 0.042 | 0.65 | 0.95 | 0.049 | 0.63 | 0.80 | 0.062 | 0.66 | 0.95 | 0.136 | 0.55 | 0.60 | 0.138 | 0.60 | 0.35 | 0.296 |
| (Ours) hellinger near | 0.73 | 1.00 | 0.015 | 0.72 | 0.97 | 0.005 | 0.69 | 0.97 | 0.046 | 0.78 | 1.00 | 0.101 | 0.70 | 1.00 | 0.081 | 0.57 | 0.07 | 0.179 |
| (Ours) hellinger superclass | 0.66 | 0.87 | 0.073 | 0.63 | 0.73 | 0.147 | 0.67 | 0.97 | 0.104 | 0.74 | 0.97 | 0.187 | 0.61 | 0.47 | 0.265 | 0.57 | 0.27 | 0.172 |
| (Ours) hellinger superclass ‡ | 0.76 | 1.00 | 0.004 | 0.70 | 0.90 | 0.015 | 0.78 | 0.90 | 0.015 | 0.76 | 1.00 | 0.085 | 0.75 | 1.00 | 0.064 | 0.63 | 0.50 | 0.022 |
| (Ours) mse empty | 0.58 | 0.89 | 0.032 | 0.58 | 0.94 | 0.052 | 0.55 | 0.93 | 0.077 | 0.60 | 0.95 | 0.114 | 0.47 | 0.39 | 0.132 | 0.63 | 0.45 | 0.280 |
| (Ours) mse near | 0.62 | 0.95 | 0.008 | 0.66 | 0.90 | 0.017 | 0.62 | 0.87 | 0.065 | 0.73 | 0.96 | 0.086 | 0.65 | 0.94 | 0.086 | 0.64 | 0.08 | 0.175 |
| Concept Ablation | 0.58 | 0.64 | 0.131 | 0.67 | 0.79 | 0.104 | 0.60 | 0.60 | 0.243 | 0.70 | 0.94 | 0.174 | 0.56 | 0.43 | 0.349 | 0.62 | 0.17 | 0.238 |

*Table 16.* Quantitative results with 150 iterations for variational methods, reporting CS, CA, and KID. For methods other than the Original Model, the best values are in bold and the second-best in italics. † denotes Importance Sampling and ‡ denotes Prior Preservation and Gradient Surgery. Base model: SD 1.4.

### Erasing Snoopy

| Method | Snoopy CS↓ | CA↓ | KID | Grumpy Cat CS↑ | CA↑ | KID↓ | Wall-E CS↑ | CA↑ | KID↓ | R2D2 CS↑ | CA↑ | KID↓ | Van Gogh CS↑ | CA↑ | KID↓ | Salvador Dali CS↑ | CA↑ | KID↓ |
|---|---|---|---|---|---|---|---|---|---|---|---|---|---|---|---|---|---|---|
| Original Model | 0.73 | 1.00 | - | 0.74 | 1.00 | - | 0.73 | 1.00 | - | 0.75 | 1.00 | - | 0.80 | 1.00 | - | 0.68 | 0.80 | - |
| (Ours) hellinger empty | 0.52 | 0.00 | 0.269 | 0.67 | 0.67 | 0.177 | 0.71 | 1.00 | 0.098 | 0.71 | 1.00 | 0.128 | 0.76 | 0.97 | 0.083 | 0.66 | 0.70 | 0.023 |
| (Ours) hellinger empty ‡ | 0.55 | 0.10 | 0.355 | 0.74 | 1.00 | 0.186 | 0.74 | 1.00 | 0.091 | 0.76 | 1.00 | 0.154 | 0.76 | 1.00 | 0.027 | 0.64 | 0.73 | 0.027 |
| (Ours) hellinger empty ‡ † | 0.57 | 0.00 | 0.368 | 0.71 | 1.00 | 0.137 | 0.72 | 0.93 | 0.069 | 0.78 | 1.00 | 0.144 | 0.72 | 1.00 | 0.045 | 0.65 | 0.83 | 0.023 |
| (Ours) hellinger near | 0.54 | 0.17 | 0.276 | 0.79 | 1.00 | 0.127 | 0.78 | 1.00 | 0.079 | 0.78 | 1.00 | 0.078 | 0.80 | 1.00 | 0.037 | 0.68 | 0.87 | 0.001 |
| (Ours) hellinger near ‡ | 0.54 | 0.00 | 0.331 | 0.75 | 1.00 | 0.144 | 0.72 | 1.00 | 0.084 | 0.75 | 1.00 | 0.119 | 0.75 | 0.97 | 0.051 | 0.66 | 0.77 | 0.029 |
| (Ours) hellinger near ‡ † | 0.54 | 0.03 | 0.414 | 0.76 | 1.00 | 0.173 | 0.70 | 0.97 | 0.116 | 0.76 | 1.00 | 0.194 | 0.73 | 0.93 | 0.050 | 0.63 | 0.77 | 0.058 |
| (Ours) hellinger superclass | 0.55 | 0.00 | 0.295 | 0.76 | 1.00 | 0.144 | 0.77 | 0.93 | 0.076 | 0.78 | 1.00 | 0.084 | 0.79 | 1.00 | 0.055 | 0.68 | 0.90 | 0.012 |
| (Ours) hellinger superclass ‡ | 0.56 | 0.03 | 0.293 | 0.74 | 1.00 | 0.183 | 0.71 | 0.97 | 0.100 | 0.79 | 1.00 | 0.124 | 0.74 | 1.00 | 0.043 | 0.65 | 0.77 | 0.027 |
| (Ours) hellinger superclass ‡ † | 0.57 | 0.03 | 0.271 | 0.76 | 1.00 | 0.159 | 0.69 | 1.00 | 0.095 | 0.78 | 1.00 | 0.097 | 0.74 | 1.00 | 0.023 | 0.64 | 0.80 | 0.042 |

### Erasing Grumpy Cat

| Method | Snoopy CS↑ | CA↑ | KID↓ | Grumpy Cat CS↓ | CA↓ | KID | Wall-E CS↑ | CA↑ | KID↓ | R2D2 CS↑ | CA↑ | KID↓ | Van Gogh CS↑ | CA↑ | KID↓ | Salvador Dali CS↑ | CA↑ | KID↓ |
|---|---|---|---|---|---|---|---|---|---|---|---|---|---|---|---|---|---|---|
| Original Model | 0.73 | 1.00 | - | 0.74 | 1.00 | - | 0.73 | 1.00 | - | 0.75 | 1.00 | - | 0.80 | 1.00 | - | 0.68 | 0.80 | - |
| (Ours) hellinger empty | 0.70 | 0.90 | 0.096 | 0.52 | 0.07 | 0.412 | 0.75 | 0.97 | 0.106 | 0.77 | 1.00 | 0.080 | 0.76 | 0.97 | 0.060 | 0.68 | 0.97 | 0.029 |
| (Ours) hellinger empty ‡ | 0.67 | 0.97 | 0.113 | 0.53 | 0.03 | 0.488 | 0.69 | 0.93 | 0.084 | 0.78 | 1.00 | 0.102 | 0.76 | 1.00 | 0.049 | 0.67 | 0.80 | 0.021 |
| (Ours) hellinger empty ‡ † | 0.69 | 1.00 | 0.091 | 0.52 | 0.00 | 0.482 | 0.69 | 0.93 | 0.094 | 0.78 | 1.00 | 0.103 | 0.73 | 1.00 | 0.064 | 0.66 | 0.93 | 0.023 |
| (Ours) hellinger near | 0.76 | 1.00 | 0.068 | 0.55 | 0.00 | 0.335 | 0.76 | 0.93 | 0.099 | 0.79 | 1.00 | 0.093 | 0.80 | 1.00 | 0.055 | 0.68 | 0.90 | 0.015 |
| (Ours) hellinger near ‡ | 0.70 | 1.00 | 0.121 | 0.51 | 0.00 | 0.489 | 0.74 | 1.00 | 0.077 | 0.78 | 1.00 | 0.128 | 0.75 | 1.00 | 0.038 | 0.67 | 0.80 | 0.014 |
| (Ours) hellinger near ‡ † | 0.69 | 1.00 | 0.093 | 0.51 | 0.00 | 0.553 | 0.74 | 0.97 | 0.082 | 0.79 | 1.00 | 0.080 | 0.77 | 1.00 | 0.042 | 0.69 | 0.90 | 0.032 |
| (Ours) hellinger superclass | 0.77 | 1.00 | 0.076 | 0.55 | 0.00 | 0.399 | 0.76 | 0.97 | 0.084 | 0.79 | 1.00 | 0.082 | 0.77 | 1.00 | 0.069 | 0.67 | 0.97 | 0.022 |
| (Ours) hellinger superclass ‡ | 0.71 | 1.00 | 0.061 | 0.54 | 0.00 | 0.508 | 0.72 | 0.97 | 0.070 | 0.78 | 1.00 | 0.064 | 0.74 | 1.00 | 0.059 | 0.66 | 0.87 | 0.023 |
| (Ours) hellinger superclass ‡ † | 0.64 | 1.00 | 0.150 | 0.51 | 0.00 | 0.541 | 0.71 | 0.90 | 0.100 | 0.78 | 1.00 | 0.088 | 0.74 | 0.97 | 0.043 | 0.67 | 0.77 | 0.026 |

### Erasing Wall-E

| Method | Snoopy CS↑ | CA↑ | KID↓ | Grumpy Cat CS↑ | CA↑ | KID↓ | Wall-E CS↓ | CA↓ | KID | R2D2 CS↑ | CA↑ | KID↓ | Van Gogh CS↑ | CA↑ | KID↓ | Salvador Dali CS↑ | CA↑ | KID↓ |
|---|---|---|---|---|---|---|---|---|---|---|---|---|---|---|---|---|---|---|
| Original Model | 0.73 | 1.00 | - | 0.74 | 1.00 | - | 0.73 | 1.00 | - | 0.75 | 1.00 | - | 0.80 | 1.00 | - | 0.68 | 0.80 | - |
| (Ours) hellinger empty | 0.70 | 0.90 | 0.057 | 0.73 | 1.00 | 0.033 | 0.55 | 0.23 | 0.256 | 0.69 | 0.83 | 0.172 | 0.78 | 1.00 | 0.038 | 0.69 | 0.90 | −0.007 |
| (Ours) hellinger empty ‡ | 0.70 | 1.00 | 0.097 | 0.75 | 1.00 | 0.087 | 0.60 | 0.87 | 0.256 | 0.61 | 0.93 | 0.292 | 0.76 | 1.00 | 0.044 | 0.66 | 0.87 | 0.010 |
| (Ours) hellinger empty ‡ † | 0.69 | 1.00 | 0.112 | 0.75 | 1.00 | 0.046 | 0.59 | 0.37 | 0.349 | 0.64 | 1.00 | 0.394 | 0.78 | 1.00 | 0.040 | 0.67 | 0.87 | 0.027 |
| (Ours) hellinger near | 0.76 | 1.00 | 0.066 | 0.75 | 1.00 | 0.047 | 0.60 | 0.53 | 0.251 | 0.78 | 1.00 | 0.112 | 0.80 | 1.00 | 0.050 | 0.69 | 0.90 | −0.013 |
| (Ours) hellinger near ‡ | 0.71 | 1.00 | 0.129 | 0.75 | 1.00 | 0.076 | 0.56 | 0.27 | 0.294 | 0.68 | 1.00 | 0.283 | 0.79 | 1.00 | 0.039 | 0.66 | 0.83 | 0.006 |
| (Ours) hellinger near ‡ † | 0.68 | 1.00 | 0.110 | 0.74 | 1.00 | 0.064 | 0.60 | 0.53 | 0.305 | 0.65 | 0.90 | 0.309 | 0.77 | 1.00 | 0.049 | 0.68 | 0.93 | 0.012 |
| (Ours) hellinger superclass | 0.78 | 1.00 | 0.064 | 0.75 | 1.00 | 0.014 | 0.59 | 0.50 | 0.250 | 0.77 | 1.00 | 0.091 | 0.80 | 1.00 | 0.026 | 0.70 | 0.93 | −0.012 |
| (Ours) hellinger superclass ‡ | 0.72 | 1.00 | 0.094 | 0.76 | 1.00 | 0.060 | 0.61 | 0.67 | 0.316 | 0.69 | 1.00 | 0.259 | 0.78 | 1.00 | 0.048 | 0.66 | 0.83 | 0.003 |
| (Ours) hellinger superclass ‡ † | 0.72 | 1.00 | 0.112 | 0.75 | 1.00 | 0.040 | 0.58 | 0.33 | 0.341 | 0.69 | 0.97 | 0.277 | 0.78 | 1.00 | 0.053 | 0.68 | 0.93 | 0.004 |

### Erasing R2D2

| Method | Snoopy CS↑ | CA↑ | KID↓ | Grumpy Cat CS↑ | CA↑ | KID↓ | Wall-E CS↑ | CA↑ | KID↓ | R2D2 CS↓ | CA↓ | KID | Van Gogh CS↑ | CA↑ | KID↓ | Salvador Dali CS↑ | CA↑ | KID↓ |
|---|---|---|---|---|---|---|---|---|---|---|---|---|---|---|---|---|---|---|
| Original Model | 0.73 | 1.00 | - | 0.74 | 1.00 | - | 0.73 | 1.00 | - | 0.75 | 1.00 | - | 0.80 | 1.00 | - | 0.68 | 0.80 | - |
| (Ours) hellinger empty | 0.70 | 0.90 | 0.082 | 0.74 | 0.97 | 0.017 | 0.63 | 0.87 | 0.134 | 0.57 | 0.40 | 0.394 | 0.77 | 1.00 | 0.052 | 0.68 | 0.80 | 0.032 |
| (Ours) hellinger empty ‡ | 0.65 | 0.97 | 0.148 | 0.74 | 1.00 | 0.094 | 0.60 | 0.83 | 0.215 | 0.56 | 0.03 | 0.549 | 0.77 | 1.00 | 0.030 | 0.65 | 0.90 | 0.042 |
| (Ours) hellinger empty ‡ † | 0.70 | 0.97 | 0.127 | 0.76 | 1.00 | 0.066 | 0.61 | 0.97 | 0.207 | 0.56 | 0.10 | 0.516 | 0.79 | 1.00 | 0.062 | 0.69 | 0.97 | 0.006 |
| (Ours) hellinger near | 0.78 | 1.00 | 0.138 | 0.74 | 1.00 | 0.070 | 0.67 | 1.00 | 0.135 | 0.55 | 0.23 | 0.431 | 0.79 | 1.00 | 0.049 | 0.70 | 0.90 | 0.004 |
| (Ours) hellinger near ‡ | 0.68 | 1.00 | 0.227 | 0.73 | 1.00 | 0.123 | 0.65 | 1.00 | 0.144 | 0.57 | 0.13 | 0.545 | 0.79 | 1.00 | 0.057 | 0.67 | 0.83 | 0.019 |
| (Ours) hellinger near ‡ † | 0.67 | 1.00 | 0.147 | 0.75 | 1.00 | 0.071 | 0.62 | 0.93 | 0.192 | 0.55 | 0.00 | 0.561 | 0.76 | 1.00 | 0.058 | 0.70 | 0.83 | 0.018 |
| (Ours) hellinger superclass | 0.75 | 1.00 | 0.118 | 0.73 | 1.00 | 0.057 | 0.67 | 0.97 | 0.105 | 0.57 | 0.37 | 0.407 | 0.79 | 1.00 | 0.039 | 0.69 | 0.87 | 0.007 |
| (Ours) hellinger superclass ‡ | 0.71 | 0.97 | 0.142 | 0.76 | 1.00 | 0.041 | 0.59 | 0.83 | 0.186 | 0.55 | 0.03 | 0.450 | 0.78 | 1.00 | 0.034 | 0.67 | 0.90 | −0.002 |
| (Ours) hellinger superclass ‡ † | 0.67 | 1.00 | 0.141 | 0.72 | 1.00 | 0.084 | 0.59 | 0.87 | 0.205 | 0.58 | 0.13 | 0.535 | 0.78 | 1.00 | 0.064 | 0.68 | 0.93 | 0.027 |

### Erasing Van Gogh

| Method | Snoopy CS↑ | CA↑ | KID↓ | Grumpy Cat CS↑ | CA↑ | KID↓ | Wall-E CS↑ | CA↑ | KID↓ | R2D2 CS↑ | CA↑ | KID↓ | Van Gogh CS↓ | CA↓ | KID | Salvador Dali CS↑ | CA↑ | KID↓ |
|---|---|---|---|---|---|---|---|---|---|---|---|---|---|---|---|---|---|---|
| Original Model | 0.73 | 1.00 | - | 0.74 | 1.00 | - | 0.73 | 1.00 | - | 0.75 | 1.00 | - | 0.80 | 1.00 | - | 0.68 | 0.80 | - |
| (Ours) hellinger empty | 0.73 | 1.00 | 0.049 | 0.71 | 0.97 | 0.061 | 0.72 | 1.00 | 0.067 | 0.74 | 1.00 | 0.043 | 0.53 | 0.17 | 0.322 | 0.64 | 0.90 | 0.081 |
| (Ours) hellinger empty ‡ | 0.76 | 1.00 | 0.089 | 0.76 | 1.00 | 0.045 | 0.75 | 0.93 | 0.067 | 0.79 | 1.00 | 0.079 | 0.59 | 0.57 | 0.344 | 0.57 | 0.03 | 0.316 |
| (Ours) hellinger empty ‡ † | 0.77 | 1.00 | 0.141 | 0.72 | 1.00 | 0.038 | 0.75 | 1.00 | 0.074 | 0.79 | 1.00 | 0.066 | 0.61 | 0.40 | 0.341 | 0.56 | 0.23 | 0.283 |
| (Ours) hellinger near | 0.76 | 1.00 | 0.048 | 0.74 | 0.97 | 0.016 | 0.76 | 0.93 | 0.041 | 0.77 | 1.00 | 0.030 | 0.55 | 0.00 | 0.311 | 0.63 | 0.73 | 0.061 |
| (Ours) hellinger near ‡ | 0.78 | 1.00 | 0.129 | 0.74 | 1.00 | 0.053 | 0.72 | 0.97 | 0.088 | 0.78 | 1.00 | 0.086 | 0.58 | 0.07 | 0.296 | 0.59 | 0.37 | 0.143 |
| (Ours) hellinger near ‡ † | 0.75 | 0.97 | 0.150 | 0.71 | 1.00 | 0.050 | 0.73 | 0.87 | 0.068 | 0.80 | 1.00 | 0.106 | 0.61 | 0.53 | 0.305 | 0.56 | 0.10 | 0.313 |
| (Ours) hellinger superclass | 0.76 | 1.00 | 0.036 | 0.75 | 1.00 | 0.035 | 0.77 | 0.97 | 0.056 | 0.76 | 1.00 | 0.040 | 0.55 | 0.10 | 0.294 | 0.69 | 0.83 | 0.048 |
| (Ours) hellinger superclass ‡ | 0.76 | 1.00 | 0.120 | 0.75 | 1.00 | 0.056 | 0.75 | 0.93 | 0.082 | 0.78 | 1.00 | 0.057 | 0.58 | 0.20 | 0.365 | 0.58 | 0.23 | 0.272 |
| (Ours) hellinger superclass ‡ † | 0.74 | 1.00 | 0.114 | 0.74 | 1.00 | 0.050 | 0.73 | 0.97 | 0.069 | 0.79 | 1.00 | 0.073 | 0.61 | 0.73 | 0.360 | 0.55 | 0.13 | 0.298 |

### Erasing Salvador Dali

| Method | Snoopy CS↑ | CA↑ | KID↓ | Grumpy Cat CS↑ | CA↑ | KID↓ | Wall-E CS↑ | CA↑ | KID↓ | R2D2 CS↑ | CA↑ | KID↓ | Van Gogh CS↑ | CA↑ | KID↓ | Salvador Dali CS↓ | CA↓ | KID |
|---|---|---|---|---|---|---|---|---|---|---|---|---|---|---|---|---|---|---|
| Original Model | 0.73 | 1.00 | - | 0.74 | 1.00 | - | 0.73 | 1.00 | - | 0.75 | 1.00 | - | 0.80 | 1.00 | - | 0.68 | 0.80 | - |
| (Ours) hellinger empty | 0.73 | 1.00 | 0.062 | 0.71 | 1.00 | 0.010 | 0.72 | 0.93 | 0.077 | 0.71 | 0.97 | 0.066 | 0.60 | 0.53 | 0.138 | 0.58 | 0.23 | 0.315 |
| (Ours) hellinger empty ‡ | 0.72 | 0.90 | 0.112 | 0.75 | 1.00 | 0.018 | 0.71 | 0.93 | 0.109 | 0.78 | 1.00 | 0.065 | 0.58 | 0.30 | 0.293 | 0.58 | 0.10 | 0.246 |
| (Ours) hellinger empty ‡ † | 0.73 | 0.93 | 0.118 | 0.71 | 1.00 | 0.042 | 0.75 | 1.00 | 0.084 | 0.80 | 1.00 | 0.056 | 0.56 | 0.10 | 0.295 | 0.59 | 0.03 | 0.281 |
| (Ours) hellinger near | 0.78 | 1.00 | 0.076 | 0.75 | 1.00 | −0.008 | 0.76 | 1.00 | 0.060 | 0.76 | 1.00 | 0.032 | 0.61 | 0.50 | 0.190 | 0.55 | 0.13 | 0.313 |
| (Ours) hellinger near ‡ | 0.72 | 0.90 | 0.092 | 0.74 | 1.00 | 0.018 | 0.76 | 0.93 | 0.093 | 0.78 | 1.00 | 0.071 | 0.60 | 0.30 | 0.284 | 0.55 | 0.00 | 0.410 |
| (Ours) hellinger near ‡ † | 0.73 | 0.97 | 0.116 | 0.73 | 1.00 | 0.019 | 0.76 | 0.97 | 0.048 | 0.79 | 1.00 | 0.058 | 0.56 | 0.20 | 0.251 | 0.57 | 0.17 | 0.281 |
| (Ours) hellinger superclass | 0.77 | 1.00 | 0.032 | 0.74 | 1.00 | −0.001 | 0.77 | 0.97 | 0.073 | 0.76 | 1.00 | 0.023 | 0.69 | 0.97 | 0.122 | 0.56 | 0.13 | 0.291 |
| (Ours) hellinger superclass ‡ | 0.75 | 0.90 | 0.120 | 0.72 | 0.97 | 0.011 | 0.75 | 0.90 | 0.072 | 0.78 | 1.00 | 0.050 | 0.57 | 0.37 | 0.260 | 0.58 | 0.17 | 0.273 |
| (Ours) hellinger superclass ‡ † | 0.74 | 0.93 | 0.122 | 0.73 | 1.00 | 0.016 | 0.76 | 0.93 | 0.070 | 0.79 | 1.00 | 0.055 | 0.58 | 0.23 | 0.247 | 0.57 | 0.03 | 0.357 |

