# OpenReview forum: "A Unified Framework for Diffusion Model Unlearning with f-Divergence"
_ICML.cc/2026/Conference — ICML 2026 regular_

### Official Review · Reviewer_nGKy · 2026-03-11

**Soundness:** 3
**Presentation:** 3
**Significance:** 3
**Originality:** 3
**Overall Recommendation:** 4
**Confidence:** 1

**Summary:**

This paper proposes a unified framework for concept erasure in text-to-image diffusion models that generalises existing MSE-based unlearning losses with f-divergences. For a subset of f-divergences (Jeffreys, squared Hellinger, Pearson), the paper derives closed-form loss functions. The paper provides a theoretical analysis of gradient magnitudes. Experiments across style erasure, object ablation, and nudity removal on SD 1.4, 1.5, 2.1, and XL demonstrate results.

**Compliance With Llm Reviewing Policy:**

Affirmed.

**Final Justification:**

Rebuttal mostly addressed the issues, and I kept the positive rating.

**Key Questions For Authors:**

See weaknesses

**Limitations:**

See weaknesses

**Strengths And Weaknesses:**

Strengths:

A unified framework is proposed to generalise the MSE. A comprehensive theoretical analysis is provided and preliminarily validated through real-data experiments. Results show the effectiveness of the proposed method, and the method is compared to other unlearning methods. Experiments show the effectiveness across multiple types of concepts to unlearn.

Weaknesses:

1. It would be better to test on more recent models, such as FLUX.
2. It would be helpful to add a short summary of the findings in table captions (e.g., Table 1)
3. It would be better to give some guidance for the choice of the loss for future works.

---

> ### Author Rebuttal · Authors · 2026-03-30
>
> We thank the Reviewer nGKy for the positive assessment and for highlighting the “comprehensive theoretical analysis” and “effectiveness across multiple types of concepts” as strengths. We have addressed the constructive suggestions as follows:
>
> **[W1: Evaluation on Recent Models]**:
> We agree that testing the framework on the latest architectures is important for demonstrating its longevity and its practical impact. Therefore, we have performed additional experiments on Stable Diffusion 3 (SD3), a Diffusion Transformer-based architecture. In detail:
> - **Architectural Generalization**: To address this, we have included new experiments on SD3. Like FLUX, SD3 utilizes the state-of-the-art Diffusion Transformer (DiT) architecture, not relying anymore on a UNet (which was instead used for SD1.4, SD1.5, SD2.1, and SDXL). The results are reported in Table A (erasure of artistic styles) and Table B (erasure of nudity) in our reply to W1 of Reviewer cAg6, showing that the benefits of using $f$-DMU transfer to different backbones. Crucially, the empirical trends reported in our original experiments, such as the superior preservation of H-DMU, hold for this new architecture. In the revised paper, we will also include examples with FLUX, as we will have enough time to run all experiments.
> - **Model Agnosticism**: We want to highlight that $f$-DMU is architecturally agnostic (for T2I DMs, which is the scope of the paper) because it formulates the loss on the output distributions rather than targeting specific internal layers (as cross-attention, used by UCE or RECE). Thus, it is natively ready for current and future diffusion-based (and flow-based, such as SD3) architectures.
>
> **[W2: Summary of the Findings in Tables]**:
> This is an excellent suggestion for improving the readability of the paper. We have revised the captions for all main tables to include a brief summary of the key takeaway. For instance, for Table 1: “Comparison of unlearning methods on SD1.4. Key finding: Our Hellinger-based loss achieves significant improvement in the preservation of non-target concepts compared to standard MSE.”
>
> **[W3: Guidance for the Choice of the Loss Function]**:
> We thank the Reviewer for this valuable suggestion. We agree that providing clear suggestions for practitioners is essential for the long-term impact of the $f$-DMU framework. In response, we have updated the manuscript to provide explicit guidance:
> - **Highlighting Existing Comparison**: While Table 6 in Appendix B.3 already detailed the theoretical trade-offs, we realized this information should be more prominent. We have therefore inserted the core of this discussion into the main text of the paper.
> - **Revised Conclusion and Practitioner's Guide**: We have added a dedicated “Guidance for Practitioners” paragraph in our Conclusions (Sec. 6): “Our framework allows users to select a loss based on their specific unlearning requirements: i) H-DMU (closed-form) is recommended (due to its bounded gradients) as the default for most tasks, where preserving non-target concepts and generative quality is crucial. ii) $\chi^2$-DMU (closed-form) is recommended when the user targets stronger erasure than MSE and some slight fidelity loss is acceptable. iii) Our variational framework, and in particular its Hellinger-based loss, is recommended when the goal is the reduction of the number of training iterations and aggressive concept removal. We emphasize that our $f$-divergence-based losses provide a new library of theoretically-grounded objectives that can serve as the foundation for future DM unlearning algorithms.”
>
> **Closing Statement**: If the Reviewer finds our response satisfactory, we would be grateful if they would consider increasing their score.

---

> > ### Author Rebuttal · Reviewer_nGKy · 2026-04-03
> >
> > Mostly resolved

---

> > > ### Author Response · Authors · 2026-04-06
> > >
> > > We thank the Reviewer for the feedback and for the positive evaluation of our work. We have incorporated all suggested revisions into the manuscript and appreciate your support for the paper.

---

### Official Review · Reviewer_v3Rf · 2026-03-12

**Soundness:** 3
**Presentation:** 4
**Significance:** 2
**Originality:** 3
**Overall Recommendation:** 4
**Confidence:** 2

**Summary:**

The paper studies machine unlearning in text-to-image diffusion models and proposes $f$-DMU, a framework based on f-divergence. The authors derive unlearning objectives from the closed-form expression of the $f$-divergence between Gaussian distributions and further extend the formulation to a min–max optimization that allows the use of arbitrary f-divergences. Experiments on several unlearning scenarios demonstrate improvements over MSE-based objectives.

**Compliance With Llm Reviewing Policy:**

Affirmed.

**Final Justification:**

My main concerns has mostly been resolved.

**Key Questions For Authors:**

Q1: The paper presents a unified framework based on f-divergence, but most theoretical analysis and empirical results focus on the closed-form losses. Can you provide additional empirical evidence for the variational formulation, or under what settings the varational objective becomes advantageous?

Q2. Can the authors generalize the method to other diffusion backbones or generative architectures?

**Limitations:**

yes.

**Strengths And Weaknesses:**

Strengths:
1. The paper derives the objective from the closed-form f-divergence between Gaussian distributions and provides analysis for the resulting min–max formulation.
2. The framework connects several commonly used losses under the f-divergence perspective and offers a systematic way to study different divergence choices.
3. The method is evaluated on several Stable Diffusion variants and multiple unlearning scenarios, showing consistent improvements over MSE-based baselines.

Weaknesses:
1. Emphasis on closed-form objectives: Although the paper presents a unified framework for arbitrary f-divergence (Sec.3.), the theoretical analysis largely focuses on closed-form losses derived from Gaussian assumptions. In Sec 4.1, the authors restrict the main analysis to these closed-form analysis, while the variational formulation appears to be less stable and leads to lower image fidelity (Appendix B.3.). This somewhat limits the strength of the "unified f-divergence framework" claim.

2. Relation to MSE in the closed-form case: For the closed-form case, the proposed objectives is closed related to weighted MSE objective, where the differences mainly arise from divergence-dependent weighting of the squared error term (Eqs.6-8). So it remains uncluear whether the improvements com from the divergence itself or from the improved dynamics induced by weighting.

3. Claim on model-agnostic is overstated: The framework is claimed to be model-agnostic, yet the empirical evaluation is limited to the Stable Diffusion family. While these models differ in scale, they share similar architectures and training pipelines. It would strengthen the claim to evaluate the method on more diverse generative architectures.

---

> ### Author Rebuttal · Authors · 2026-03-30
>
> We thank Reviewer v3Rf for the high evaluation of our paper’s presentation and for the constructive comments. We address the specific points below to clarify the impact of our framework and the breadth of our theoretical analysis.
>
> **[W1 & Q1: Theoretical Focus and Variational Loss Advantages]**:
> Our theoretical treatment provides a balanced analysis of both frameworks: it covers both closed-form (Sec. 4.1) and variational (Sec. 4.2) losses. Furthermore, the variational framework has a crucial theoretical and practical role. We provide more details below:
> - **Theoretical Coverage**: While Sec. 4.1 analyzes the gradients of the proposed closed-form losses, *Sec. 4.2 and Appendices B.1.3 and B.2 study the local convergence properties of variational losses*, mathematically relating the choice of $f$-divergence with the speed of convergence.
> - **Importance of our Variational Framework**: Our variational framework offers distinct advantages:
>   1. **Efficiency**: As shown by comparing Tables 12-14, the variational approach requires significantly fewer iterations (150 vs 500) to achieve target unlearning effectiveness.
>   2. **Empirical Performance**: In Table 2, our variational Hellinger-based loss outperforms established baselines like DoCo and UCE. Although the main part of the paper focuses on the closed-form Hellinger loss, the Appendix comprises multiple results of different variational losses (including Tables 9,10,11,14), demonstrating their practical relevance.
>   3. **Universality**: It enables the use of any $f$-divergence, including those that lack closed-form Gaussian expressions (e.g., Total Variation), providing a generalized and elegant tool.
>   4. **Explanatory**: It highlights advantages and disadvantages of the existing (i.e., DoCo) and possible future variational unlearning methods.
>
> If the paper is accepted, thanks to the additional page allowed, we will move some of these empirical results from the Appendix to the main part, providing more evidence about the variational framework.
>
> **[W2: Divergence Choice vs. Gradient Weighting]**:
> We explain below that *the $f$-divergence and the resulting gradient weighting are mathematically inseparable*: the objective determines the gradient modulation and thus the improvement in learning dynamics. In detail:
> - **Preliminary Note About Weighted MSE**: We would like to clarify that our closed-form loss functions are not weighted versions of MSE. A standard weighted MSE loss is $\sum_i w_i MSE_i$. In contrast, our closed-form losses are a transformation of MSE: $\sum_i g(MSE_i)$, where $g(\cdot)$ is a non-linear function (i.e., exponential). Instead, the gradients of our closed-form losses are related to weighted gradients of MSE (7).
> - **Weighting as Mathematical Consequence**: The $f$-divergence defines the geometry of the optimization landscape, and the weighting in (7) (or, more in general for any $\alpha$-divergence in (54)) is the direct mathematical consequence of that geometry. The improvement in convergence comes from selecting an $f$-divergence (thus, a loss) that induces better optimization dynamics.
> - **Principled Adaptive Dynamics**: Unlike heuristic weighting or manual LR scheduling, the weighting terms in (7) derive from a principled theoretical formulation and provide sample-specific gradient modulation. For example, the  scaling term $e^{-MSE(\Phi, \hat{\Phi})}$ in H-DMU is not a heuristic term we added to mitigate the outliers effect, it is the inherent gradient scaling of H-DMU. We invite the Reviewer, if curious, to read a related discussion in our reply to Q1 of Reviewer cAg6.
>
> **[W3 & Q2: Architecture Generalization]**:
> We agree that expanding the evaluation beyond the SD U-Net family strengthens our contributions. Therefore, we have performed additional experiments. In detail:
> - **SD3 Validation**: We have now included new results for SD3, which is based on a fundamentally different architecture (i.e., DiT). The results are reported in Tabs. A-B in our reply to W1 of Reviewer cAg6, and have been included in the revised version of the manuscript. The empirical trends reported in our original experiments, such as the superior preservation of H-DMU, hold for this new architecture.
> - **Architectural Independence**: $f$-DMU is architecturally agnostic because it formulates the loss on the output distribution rather than targeting specific internal layers (like cross-attention, used for UCE or RECE). This confirms that, as long as a model is a T2I DM, $f$-DMU is applicable regardless of the internal backbone, thus being suitable also for future diffusion-based architectures.
>
> **Closing Statement**: We hope that our responses and additional results have addressed the Reviewer’s concerns. If so, please consider re-evaluating our submission.

---

> > ### Author Rebuttal · Reviewer_v3Rf · 2026-04-03
> >
> > Thank you for the rebuttal. My concerns have been adequately addressed. In particular, the response clarified the role of the variational formulation, provided a more precise explanation of how the proposed objectives differ from standard MSE-based formulations, and added evidence that the framework can extend beyond the original Stable Diffusion settings. I therefore maintain my positive assessment of the paper and my current overall recommendation, while encouraging the authors to make the practical guidance on when to prefer closed-form versus variational losses as explicit as possible in the final version.

---

> > > ### Author Response · Authors · 2026-04-06
> > >
> > > We thank the Reviewer for the positive assessment and for the constructive feedback. We have updated the manuscript to include explicit practical guidance on choosing between closed-form and variational losses, as suggested. We have also integrated the detailed clarifications regarding the variational framework and the additional experimental results.
> > > We appreciate your recommendation for acceptance.

---

### Official Review · Reviewer_9beX · 2026-03-12

**Soundness:** 2
**Presentation:** 3
**Significance:** 2
**Originality:** 2
**Overall Recommendation:** 4
**Confidence:** 2

**Summary:**

This paper proposes f-DMU, a unified framework for diffusion model unlearning based on f-divergence. The method generalizes existing KL/MSE-based concept erasure objectives by formulating unlearning as minimizing an f-divergence between the output distributions of a target concept and an anchor concept. For special cases such as squared Hellinger distance, Pearson χ² divergence, and Jeffreys divergence, the paper derives closed-form losses; for general f-divergences, it further introduces a variational min-max formulation. The paper also provides theoretical analysis on gradient behavior and local convergence, and empirically studies different divergences on multiple text-to-image diffusion models and unlearning settings. Results suggest that Hellinger-based losses provide a better trade-off between erasure effectiveness and prior preservation than the standard MSE objective, while variational formulations are more flexible but also more disruptive.

**Compliance With Llm Reviewing Policy:**

Affirmed.

**Final Justification:**

The rebuttal addressed my main concerns to a large extent.

**Key Questions For Authors:**

Please refer to the weakness section.

**Limitations:**

Yes.

**Strengths And Weaknesses:**

Strengths

1.[Meaningful Theoretical Generalization]
The paper addresses a meaningful problem in diffusion model unlearning. Generalizing the commonly used KL/MSE-based objective into a unified f-divergence framework is conceptually clean and gives a more principled view of several existing methods.

2.[Strong Theoretical Analysis]
A major strength of the paper is that it does not stop at proposing a generalized objective, but also analyzes how different divergences lead to different gradient scaling behaviors and convergence properties.

3.[Clear Presentation]
The paper is generally well written and easy to follow.

Weaknesses

1.[Practical Value Beyond Hellinger Remains Unclear]
While the paper presents a unified f-divergence framework, Table 1 suggests that the main practical conclusion is simply that the Hellinger-based closed-form loss works best in most settings. This makes the broader practical value of the unified framework somewhat unclear.

2.[Comparison is insufficient]
The paper would be stronger with comparisons against newer and more representative baselines in diffusion unlearning and concept erasure, such as ESD, SPM, RECE, and MACE. Without these comparisons, it is still difficult to fully position the proposed framework against recent methods.

---

> ### Author Rebuttal · Authors · 2026-03-30
>
> We thank Reviewer 9beX for the positive remarks regarding our “meaningful theoretical generalization” and “strong theoretical analysis”. We address the concerns regarding practical value and baselines below.
>
> **[W1: Practical Value Beyond Hellinger-Based Losses]**:
> Beyond the superior empirical performance of our Hellinger-based closed-form loss, the practical relevance of f-DMU resides in *systematically matching the divergence to the deployment scenario*, while providing a theoretical lens to understand and control unlearning dynamics. In detail:
> - **Beyond Empirical Search**: Our framework replaces the standard reliance on a trial-and-error selection of the objective function with a principled method (i.e., a “map”) based on gradient and convergence analysis. This “map” guides a practitioner in the choice of the loss depending on their specific goal (e.g., H$^2$ for stability and prior preservation, $\chi^2$ for aggressive unlearning). Our analysis of the class of $\alpha$-divergences reveals a continuous spectrum of gradient amplitudes (Fig. 1), allowing for a *fine-grained control over the scaling*. $f$-DMU helps in moving the loss function design for DM unlearning from heuristic-driven engineering to principled design.
> - **Unification of Existing Methods**: By unifying existing methods (e.g., CAbl, DoCo) into a single $f$-divergence-based framework, we provide a theoretical explanation for their empirical successes and failures. Our framework provides the community with a way to analyze why certain methods fail, such as MSE-based objectives leading to degenerate generation (Fig. 2) and how they can be improved by changing the loss.
> - **Framework Flexibility and Practical Effectiveness**: Our variational framework is flexible for specialized needs. For instance, our variational framework is useful when the user requires aggressive concept removal performed in a low number of iterations (specified from line 325), as it can be observed from Tabs.12-14. Moreover, in Table 2, our variational H-DMU performs better than famous existing baselines like UCE and DoCo, better than other baselines introduced in Table C below (which extends Table 2), and it achieves *superior preservation also compared to our closed-form H-DMU*.
>
> **[W2: Extensive Comparison with SOTA]**:
> We thank the Reviewer for the suggestion to include more recent baselines. To strengthen our empirical evaluation, we have significantly expanded our comparisons in the revised manuscript. In particular:
> - **Artistic Style Erasure**: We have added comparisons against ESD (ICCV 2023), SPM (CVPR 2024), RECE (ECCV 2024) [1], SAFREE (ICLR 2025) [2], and SPEED (ICLR 2026) [3] in Table C (MACE will also be included in the revised paper). These results show the effectiveness of H-DMU, and of our theoretically grounded framework, even against the latest SOTA methods. In fact, for a similar performance on preservation of non-target concepts, H-DMU achieves better erasure.
> - **Nudity Unlearning**: We have integrated SPEED into our benchmarks for nudity erasure (Table D). This complements our existing extensive comparison against SOTA methods in the original submission (see Tab. 4, Fig. 7, Fig. 13, Tab. 7, Tab. 8). Notably, H-DMU significantly outperforms also very recent methods like SPEED.
> - **Experiments on Diffusion Transformer-Based Architecture**: To prove the framework is not limited to U-Net architectures, we successfully applied f-DMU to Stable Diffusion 3 (SD3). As shown in Tables A-B (in our reply to W1 of Reviewer cAg6), our theoretical insights hold for modern Diffusion Transformer (DiT) architectures. In fact, the empirical trends reported in our original experiments, such as the superior preservation of H-DMU, hold for SD3.
>
> **Table C. Erasing Van Gogh (extends Tab. 2 in the original paper).**
> Method|CS$_e$|CA$_e$|CS$_r$|CA$_r$|KID$_r$
> -|-|-|-|-|-
> ESD|0.657|0.6|0.668|0.74|0.027
> SPM|0.714|1.0|0.733|0.92|0.021
> RECE|0.631|0.3|0.656|0.75|0.029
> SAFREE|0.661|0.7|0.677|0.78|0.025
> SPEED|0.642|0.4|0.684|0.80|0.029
> H-DMU|0.624|0.2|0.672|0.78|0.027
>
> **Table D. Erasing nudity and robustness to adversarial attacks (extends Tab. 4 in the original paper). For each value in the table, the lower the better.**
> Method|I2P|RAB|MMA-t|MMA-a|MMA-s.a.
> -|-|-|-|-|-
> SPEED|0.479|0.547|0.368|0.570|0.464
> H-DMU|0.063|0.157|0.035|0.049|0.042
>
> [1] Gong, Chao, et al. "Reliable and efficient concept erasure of text-to-image diffusion models." European Conference on Computer Vision. Cham: Springer Nature Switzerland, 2024.
> [2] Yoon, Jaehong, et al. "Safree: Training-free and adaptive guard for safe text-to-image and video generation." ICLR 2025.
> [3] Li, Ouxiang, et al. "Speed: Scalable, precise, and efficient concept erasure for diffusion models." ICLR 2026.
>
> **Closing Statement**: We believe we have addressed the concerns about the practical value of $f$-DMU and the comparison with state-of-the-art methods. If so, please consider increasing the score.

---

> > ### Author Rebuttal · Reviewer_9beX · 2026-04-04
> >
> > I carefully reviewed all the reviewers’ comments, and most of my concerns have been adequately addressed. And i'd adjusted my score accordingly.

---

> > > ### Author Response · Authors · 2026-04-06
> > >
> > > We thank the Reviewer for the feedback and for the positive assessment of our work. We have incorporated all suggested revisions into the manuscript and appreciate your support for the paper.

---

### Official Review · Reviewer_cAg6 · 2026-03-13

**Soundness:** 3
**Presentation:** 3
**Significance:** 2
**Originality:** 3
**Overall Recommendation:** 4
**Confidence:** 3

**Summary:**

This paper proposes a unified framework for diffusion model unlearning. More specifically, the authors demonstrate that previous methods (e.g., DoCo and CAbl) can be generalized as minimizing the f-divergence between the target-concept-conditioned distribution and the anchor-concept-conditioned distribution.

The authors provide two theoretical analyses: (1) a gradient analysis showing that the gradients induced by the $H^2$ divergence and the $\chi^2$ divergence correspond to scaled versions of the KL-divergence-based objective, and (2) a local convergence analysis suggesting faster convergence for the $H^2$ divergence. Experimental results empirically validate the theoretical analysis and demonstrate the superior performance of the proposed method compared to existing baselines.

**Compliance With Llm Reviewing Policy:**

Affirmed.

**Final Justification:**

The rebuttal addressed most of my initial concerns. Therefore I remain positive about this paper.

**Key Questions For Authors:**

- If the scaled gradient behavior induced by different divergences is a key factor in convergence, could similar effects be achieved through **learning-rate scheduling or gradient scaling techniques** when using the standard KL/MSE objective?
- (minor) How are anchor concepts chosen?

**Limitations:**

yes

**Strengths And Weaknesses:**

**Strengths**
- This manuscript investigates a relevant question in diffusion model unlearning. The authors claim to explore a central concept: that existing methods can be unified under an **f-divergence framework**. This perspective helps deepen the understanding of existing approaches and may benefit future research on concept erasure methods.
- The method is **theoretically grounded**, and the theoretical insights (gradient analysis and convergence analysis) are supported by empirical results. The alignment between theory and experiments is a strength of the paper.

**Weaknesses**
- **Lack of experiments on latest models.** Most experiments are conducted on Stable Diffusion models earlier than SD3. While these models are still widely used, the field is rapidly moving toward Transformer-based diffusion models (e.g., Flux, SD3, Qwen-Image). Evaluating the method on more recent architectures would strengthen the practical relevance of the work.
- **Failure cases.** This is a minor issue, but fine-tuning-based unlearning methods are known to be imperfect. In some cases, they fail to remove the concept completely or unintentionally erase related concepts. Providing qualitative examples of such failure cases would help readers better understand the limitations of the proposed method.

---

> ### Author Rebuttal · Authors · 2026-03-30
>
> We thank Reviewer cAg6 for the detailed feedback and the stimulating comments. We address the specific comments and questions below.
>
> **[W1: Experiments on Latest Models]**:
> We have performed additional experiments on SD3, a Diffusion Transformer-based architecture. In detail:
> - **New Experiments on SD3**: As shown in Tables A-B below, $f$-DMU effectively erases artistic styles and nudity in SD3. The empirical trends reported in our original experiments, such as the superior preservation of H-DMU, hold for this new architecture. P-DMU outperforms MSE for nudity erasure. We added Tables A-B and corresponding qualitative examples in the revised paper. In the revised paper, we will also include examples with FLUX and Qwen-Image, as we will have enough time to run all experiments.
> - **Architectural Agnosticism**: $f$-DMU can be used with any DM-based T2I model:  unlike architecture-specific methods (e.g., UCE), $f$-DMU formulates the objective as a divergence minimization between output distributions, making it compatible with any diffusion or flow-based architecture.
>
> **Table A. Erasing Van Gogh on SD3 (same columns as Tab. 2 of the original paper).**
> Method|CS$_e$|CA$_e$|CS$_r$|CA$_r$|KID$_r$
> -|-|-|-|-|-
> MSE|0.771|0.4|0.753|0.6|0.031
> H-DMU|0.773|0.4|**0.761**|**0.7**|**0.028**
> P-DMU|**0.766**|0.4|0.752|0.56|0.032
>
> **Table B. Erasing nudity on SD3 (same columns as Tab. 7 of the original paper).**
> Method|B.(F)|G.(F)|B.(M)|G.(M)|B.|F.|B.|A.|T.
> -|-|-|-|-|-|-|-|-|-
> SD3|11|0|7|0|0|10|34|31|59
> MSE|8|0|9|0|0|8|37|23|50
> H-DMU|**7**|0|9|0|0|9|36|23|50
> P-DMU|9|0|9|0|0|**6**|**33**|**20**|**48**
>
>  **[W2: Advantages & Disadvantages of Fine-Tuning Methods]**:
> We agree with the Reviewer. However, fine-tuning methods permanently modify the model weights, thus they cannot be circumvented when the user has access to the model. This is a fundamental feature in many practical scenarios.
> - **f-DMU Mitigates Failure Cases**: Our framework can mitigate possible failure cases of erasure of non-target concepts: our analysis demonstrates that H-DMU leads to less degradation of non-target concepts compared to MSE (e.g., Tab. 1).
> - **New Subsection**: As suggested by the Reviewer, we created a new subsection in the Appendix reporting qualitative examples of the erasure-preservation trade-off between different $f$-divergences.
>
> **[Q1: Gradient Scaling of f-DMU]**:
> We deeply thank the Reviewer for the interesting question. We argue that standard learning rate (LR) scheduling or gradient scaling (GS) cannot replicate the *sample-wise scaling* inherent in $f$-DMU, and that its behavior is a principled mathematical property rather than an external heuristic. In detail:
> - **Sample-Wise Scaling**: Standard LR schedulers and GS techniques (e.g., step decay, cosine annealing, gradient clipping, or even adaptive methods like ReduceLROnPlateau, Hypergradient Descent [1]) apply a scalar to the entire batch, based on the training or gradients history. In contrast, $f$-DMU performs *per-sample weighting* based on the specific dissimilarity of each image pair, reacting immediately to each sample’s alignment error (see (50)). For instance, H-DMU scales the gradient of each sample by $e^{-MSE(\Phi, \hat{\Phi})}$, mitigating the effect of outliers in a way that standard scheduling cannot.
> - **Theoretically Principled Scaling**: LR schedulers and GS techniques are often hand-crafted methods. While it is possible to hand-craft a scheduler to mimic $f$-DMU gradient behavior, our theoretical analysis shows that this adaptive behavior is an inherent property of $f$-DMU. It requires *no manual tuning* and introduces *no additional computational overhead*. Furthermore, the scaling factor of $f$-DMU changes smoothly with the $f$-divergence (Fig. 1), leading to a continuous spectrum of gradient dynamics: for any $\alpha$-divergence, we show in (54) that the scaling factor becomes $e^{- \frac{\alpha(1-\alpha)}{2\sigma^2} \text{MSE}(\Phi, \hat{\Phi})}$.
>
> *We have added this discussion in Section 4.1 of the revised paper*, and we have explicitly specified that (7) reports the gradient expressions for the i-th sample.
>
> **[Q2: Choice of Anchor Concepts]**:
> The anchor choice depends on the user goal (lines 140-155). Our experimental results on the anchor concept choice are reported in Appendix C.5 (from line 1811), highlighting that, even for different anchors, H-DMU remains superior to MSE. For the experiments in the main part of the paper, we use the standard hyperclass anchor, like the original CAbl paper, for a fair comparison. However, as detailed in Sec. 3, $f$-DMU is anchor-agnostic, allowing users to select the strategy that best fits their specific unlearning scenario.
>
> [1] Baydin, Atilim Gunes, et al. "Online learning rate adaptation with hypergradient descent." ICLR 2018.
>
> **Closing Statement**: We hope that our responses and additional results have addressed the Reviewer’s concerns. If so, please consider re-evaluating our submission.

---

> > ### Author Rebuttal · Reviewer_cAg6 · 2026-04-03
> >
> > I checked the rebuttal by the authors and most of my concerns are resolved. I remain positive about this paper.

---

> > > ### Author Response · Authors · 2026-04-06
> > >
> > > We thank the Reviewer for the feedback and for the positive assessment of our work. We have incorporated all suggested revisions into the manuscript and appreciate your recommendation for acceptance.

---

### Decision · Program_Chairs · 2026-04-30

**Decision:**

Accept (regular)

**Comment:**

This paper proposes a f-divergence based loss for diffusion model unlearning, which serves as a generalized framework for previously used l2 loss based methods. One strength of the paper which is unanimously acknowledged by reviewers is its theoretical analysis on the gradient and convergence under the f-divergence loss. The experiments also adequately support the claim which is also appreciated by all reviewers. Reviewers also brought up certain weaknesses such as the lack of evaluation on the state of the art models, which the AC deems to be a valid but minor issue. Based on these considerations, the AC believes this is a valuable contribution and recommend accept.